# Effective Dimension in Bandit Problems under Censorship

**Gauthier Guinet** [*]
AWS AI Labs
guinetgg@amazon.com

**Saurabh Amin** [†]
MIT
amins@mit.edu

**Patrick Jaillet** [‡]
MIT
jaillet@mit.edu

## Abstract

In this paper, we study both multi-armed and contextual bandit problems in censored environments. Our goal is to estimate the performance loss due to censorship in the context of classical algorithms designed for uncensored environments. Our main contributions include the introduction of a broad class of censorship models and their analysis in terms of the *effective dimension* of the problem – a natural measure of its underlying statistical complexity and main driver of the regret bound. In particular, the effective dimension allows us to maintain the structure of the original problem at first order, while embedding it in a bigger space, and thus naturally leads to results analogous to uncensored settings. Our analysis involves a continuous generalization of the Elliptical Potential Inequality, which we believe is of independent interest. We also discover an interesting property of decision-making under censorship: a transient phase during which initial misspecification of censorship is self-corrected at an extra cost; followed by a stationary phase that reflects the inherent slowdown of learning governed by the effective dimension. Our results are useful for applications of sequential decision-making models where the feedback received depends on strategic uncertainty (e.g., agents' willingness to follow a recommendation) and/or random uncertainty (e.g., loss or delay in arrival of information).

## 1 Introduction

Bandit problems are prototypical models of sequential decision-making under uncertainty. They are widely studied due to their applications in recommender systems, online advertising, medical treatment assignment, revenue management, network routing and control [26, 39]. Our work is motivated by settings in which the feedback received by the decision-maker in each round of decision is censored by a stochastic process that depends on the current action as well as past history of feedbacks and actions. For instance, in typical missing data problems, the decision-maker needs to deal with frequent losses of information (or delays in arrival of information) due to exogenous failures such as faulty and/or unreliable communication. Missing observations in dynamical interactions with the environment are a common concern in diverse fields ranging from operations management to health sciences to physical sciences [45, 20, 29]. In other settings, such as AI-driven platforms for health alerts, route guidance, and product recommendations [7, 46], the reception of feedback depends on whether or not the decision (or recommendation) is adopted by strategic agents (e.g. patients, customers or drivers) with private valuations. Thus, from the platform's viewpoint, the adoption behavior of heterogeneous agents can be regarded as a *stochastic censorship process*.

---

[*]Work done prior to joining Amazon.

[†]Dept. of Civil and Environmental Engineering & Laboratory for Information and Decision Systems.

[‡]Dept. of Electrical Engineering and Computer Science & Laboratory for Information and Decision Systems.

36th Conference on Neural Information Processing Systems (NeurIPS 2022).

In static environments, the bias induced by the presence of randomly missing information has been thoroughly studied [29, 33]. However, in online settings, the dynamics of learning and acting are inherently coupled: since censorship mediates current information of the environment, it impacts the outcome of data-driven decision process; this in turn conditions the future decisions and future censored feedback, creating a complex and endogenous joint temporal dependency. Our work contributes to the analysis of such phenomena for a broad classes of decision and censorship models. Importantly, it is the first *normative inquiry* of how censorship impacts the statistical complexity of bandit problems. We develop an analysis approach that is useful for both estimating the performance loss due to censorship and refining the classical algorithms designed for uncensored environments.

## 1.1 Related Work

Within the extensive bandits literature, well-surveyed in [26, 39], our work is most closely related to stochastic delayed bandits. Initially, this line of work focused on the joint evolution of actions and information in settings where the reception of the latter is delayed [15]. Of particular interest is the packet loss model recently introduced in [24], which provides the regret bound $\mathcal{O}(\frac{1}{p}R_T)$ where $R_T$ is the uncensored regret and $p$ the censorship probability. Analogous results have been shown in the context of Combinatorial Multi-Armed Bandits with probabilistically triggered arms; see for example, [11] and [43]. Our work provides a systematic approach to study more general censorship models, and sheds light on how the impact of coupled feedback and censorship realizations on the expected regret can be evaluated in terms of the *effective dimension* of the problem.

Importantly, we also tackle the contextual bandit problems, where relatively few results are available on the regret under missing or censored feedback. A notable exception is the work of [42], who focus on a different information structure and obtain a scaling of $1/p$ (see Remark 4.4). A related contribution by [2] provides both a potential-based analysis of the Upper Confidence Bound algorithm (UCB) for multi-armed bandits and an algorithmic variant leveraging the Kaplan-Meier estimator, although their censorship setting is different than ours. In particular, our results are applicable to settings when delay is significantly large (possibly infinite). This is in contrast to prior results on bandits with delayed information structure which assume either that the delay is *constant*, *upper bounded*, has a *finite mean*, or simply provide regret guarantees that are *linear in the cumulative delays* up to time $T$ [15, 22, 30, 47, 34]. Under such assumptions on delay, one usually gets a second order additive dependency of the regret in terms of delay parameters, which practically says that delay is benign for bandits. On the other hand, we show that censorship leads to a first order multiplicative dependency on regret and we provide a complete characterization of this dependency for a wide range of bandits and censorship models.

Moreover, the abovementioned works primarily focus on modifying well-known bandit algorithms to account for delays, or propose new delay-robust algorithms which may be difficult to implement in practice; a notable exception includes [44] but it focuses on Thompson Sampling. In our work, we instead focus on estimating the performance loss due to censorship and derive insights on the behavior of well-known UCB class of algorithms [27, 13, 1]. These algorithms are widely used in practice; moreover, their theoretical study has been shown to be useful for analysis of broader class of algorithms (notably Thompson Sampling [3, 36] and Information-Directed Sampling [37, 23]).

On a somewhat related note, the literature on non-stochastic multi-armed bandit problems with delays [32, 9, 21] also tackles multiplicative dependency, although in a different setting than ours. Another related line of work is Partial Monitoring [6, 25] which deals with generic categorization of learnability, rather than a fine-grained analysis of dimensionality in relation to censorship, which is our current focus.

Our work contributes to the Generalized Linear Contextual Bandits literature [16, 28] in two ways: firstly, through the use of these models in a sequential decision-making framework on which the impact of censorship is assessed in Sec. 4. Secondly, by showing that our multi-threshold censorship model $\mathcal{MT}$ induces, at first order, a non-linear structure that closely mirrors such models. Our results provide new tools to study this structure. It is useful to note that the notion of *effective dimension* has been well-studied in the statistical learning and kernels literature [40, 41] (where it is defined for a Gram matrix $K_n$ and regularization $\lambda$ as $d_{\text{eff}}^n(\lambda) = \text{tr}(K_n(K_n + \lambda\mathbb{I}_d)^{-1})$). Our work shows that an analogous quantity governs the regret bound of bandit problems in censored settings.

Finally, there is a rich literature on classical missing and censored data problems [29, 33]. Although conditional on the choice of a given action the missing data/censorship process we study is an instance of missing-completely-at-random (MCAR), the online action generating process adds a significant difficulty to the problem: whereas MCAR is typically studied under a well-defined distributional assumption (e.g. i.i.d. generation of action), our problem needs to deal with adaptive (hence non i.i.d.) data generation process with respect to the filtration of past information. In particular, the structure of missing data set results from strong endogenous dependencies with past realization of the censorship (see Sec. 2).

## 1.2 Summary of Results

In Sec. 3, we consider Multi-Armed Bandit (MAB) models and prove that the regret scales as $\tilde{\mathcal{O}}(d_{eff}\sqrt{T})$ (Thm. 3.1), where $d_{eff}$ is the effective dimension with value $\sum_{a \in [d]} \frac{1}{p_a}$. In doing so, we recover and generalize related results from [24, 11] to more complex regularized settings and noise models. In particular, we prove that the effective dimension results from characterizing the so-called censored cumulative potential $\mathbb{V}_\alpha$. Interestingly, we also show that the adaptive nature of censorship on $\mathbb{V}_\alpha$ plays only a second order role (Prop. 3.7), that is, impact of censorship can be treated in an *offline* manner at first order.

Importantly, our study of MAB under censorship instantiates an analysis framework which extends to Linear Contextual Bandits (LCB) (Sec. 4). Our main result provides that regret is still governed by the effective dimension, but now with a dependency of $\tilde{\mathcal{O}}(\sigma\sqrt{d \cdot d_{eff}}\sqrt{T})$ (Thm. 4.1). To the best of our knowledge, these regret bounds provide the first theoretical characterization in LCB with censorship, and contribute to the literature by evaluating the impact of censorship on the performance of UCB-type algorithms. Our second main contribution is identifying the effective dimension for a broad class of multi-threshold models $\mathcal{MT}$ as well as a precise understanding of the dynamic behavior induced by these models (Thm. 4.6). In particular, we find that censorship introduces a two-phase behavior: a transient phase during which the initial censoring misspecification is self-corrected at an additional cost; followed by a stationary phase that reflects the inherent slowdown of learning governed by the effective dimension. In extending our analysis from MAB to LCB, we also develop a continuous generalization of the widely used Elliptical Potential Inequality (Prop. 4.3), which we believe is also of independent interest. Finally, our results (Thm. 3.1 and Prop. 3.2 for MAB and Thm. 4.1 for LCB) suggest that the UCB class of algorithms is indeed a reliable method for stochastic bandits problems under censorship.

## 2 Problem Setup and Background

**Bandit Model:**  We successively consider stochastic multi-armed bandits (Sec. 3) and Linear Contextual Bandits (LCB) (Sec. 4) in censored environments. In both settings, at each round $t \leq T$, the agent observes an action set $\mathcal{A}_t \subset \mathcal{A}$. She then selects an action $a_t \in \mathcal{A}_t$ (i.e. an *arm*) to which a noisy feedback $r(a_t) + \epsilon_t$ is associated, where $r(a_t)$ is a bounded reward and $\epsilon_t$ is an i.i.d. sub-Gaussian noise of pseudo-variance $\sigma^2$. For action $a$, the sub-optimality gap at time $t$ is denoted $\Delta_t(a) \triangleq \max_{\tilde{a} \in \mathcal{A}_t} r(\tilde{a}) - r(a)$, and the maximal gap $\Delta_{max} \triangleq \max_{a,t} \Delta_t(a)$. We now recall the specifics of each model:

- **MAB**: There is a finite number of actions $d$, enumerated as $\mathcal{A} \triangleq [d]$, each having a scalar reward $\theta_a^\star$. Arms are *independent*: playing one arm gives no information about the others.

- **LCB**: The action set $\mathcal{A}_t$ is a subset of the unit ball $\mathbb{B}_d$, possibly infinite. Unless explicitly mentioned, the reward is assumed to be linear with respect to a latent unknown vector $\theta^\star \in \mathbb{R}^d$, i.e. $r(a) = \langle a, \theta^\star \rangle$. Non-stochastic contexts are modeled by the fact that $\mathcal{A}_t$ is drawn by an oblivious adversary. Here one does not need to rely on the typical i.i.d assumption on their generating process [47, 16].

**Information Structure:**  In the classical uncensored setting, the noisy feedback is immediately observed post-decision and utilized to make decisions in the next round. We introduce the following **censorship** model: an independent Bernoulli random variable of parameter $p(a_t)$ denoted as $x_{a_t}$ is drawn after each decision $a_t$ and the feedback is observed, i.e., *realized*, if and only if $x_{a_t} = 1$; else the feedback is said to be *censored*. We recover the uncensored setting when $p(a) \equiv 1$. Henceforth,

in both finite and linear settings, the Bernoulli parameter corresponding to the censorship probability depends on the action chosen i.e. our model allows the censorship to be heterogeneous across actions. Given that the action chosen at time $t$ is a random variable, $p(a_t)$ refers to a random variable as well.

---

**Algorithm 1:** Generic UCB

---

**Input:** Total time $T$, Regularization $\lambda$, Precision $\delta$
**for** $t = 1, \ldots, T$ **do**
    Provide reward estimator $\tilde{r}_t^\lambda$ verifying w.p.
    $1 - \delta$:
      $\forall a \in \mathcal{A}_t, r(a) \leq \tilde{r}_t^\lambda(a)$;
    Play action $a_t = \mathrm{argmax}_{a \in \mathcal{A}_t} \tilde{r}_t^\lambda(a)$ ;
    **if** $(a_t, r(a_t) + \epsilon_t)$ *is realized i.e.* $x_{a_t} = 1$ **then**
      | Update $\tilde{r}_t^\lambda$;
    **end**
**end**

---

**Algorithms:** To study the impact of censorship on bandit problems, we consider the class of high-probability index algorithms based on the *optimism under uncertainty* principle, commonly referred as **UCB**-algorithms. Following [23], Algorithm 1 summarizes the generic UCB design framework. We detail in App. A the specific instances of UCB for MAB (resp. LCB) used in Sec. 3 (resp. Sec. 4). Moreover, this family of algorithms strongly relies on regularized reward estimators $\tilde{r}_t^\lambda$, where the regularizer is mostly used to prevent an artificial cold-start exploratory phase.

**Performance Criterion:** The frequentist performance of the agent is measured by the notion of *pseudo regret*, i.e., the difference between the algorithm's cumulative reward and the best total reward. More formally, we introduce for any policy $\pi \in \Pi$:

$$R(T, \pi) = \sum_{t=1}^{T} \max_{a \in \mathcal{A}_t} r(a) - \sum_{t=1}^{T} r(a_t) = \sum_{t=1}^{T} \Delta_t(a_t).$$

We aim to provide guarantees on $\mathbb{E}[R(T, \pi)]$ with respect to the number of rounds $T$ and quantities that govern the *complexity* of the problem (for example number of arms, ambient dimension $d$, parameters of censorship model or smoothness properties of the reward $r$). Here, the expectation is with respect to the noise induced by the feedback, the censorship and a possibly randomized policy.

### 2.1 Notations

Transpose of a vector $u$ is denoted by $u^\top$, classical Euclidean inner product by $\langle ., . \rangle$ and trace operator by $\mathrm{Tr}$. For positive semi-definite matrix $\Sigma \in \mathbb{R}^{d \times d}$ and for any vector $u \in \mathbb{R}^d$, notation $\|u\|_\Sigma$ refers to $\sqrt{u^\top \Sigma u}$. We use notation $\mathbb{I}_d$ to denote the $d \times d$ identity matrix. $\mathbb{B}_d$ is the unit ball in $\mathbb{R}^d$. $[n]$ is the set of integers $\{1, 2, \cdots, n\}$. For a given function $f$, we note $f^{(i)}$ the $i^{th}$ derivative of $f$. To avoid confusion with the dimension $d$, we use $\partial x$ instead of $dx$ to denote an infinitesimal increase of $x$. We use the asymptotic notations $\sim, \mathcal{O}, \Theta$ and $\tilde{\mathcal{O}}$ ($\mathcal{O}$ when $\log$ factors are removed). Finally, for an event $\mathcal{H}$, we use $\neg\mathcal{H}$ to denote its complement.

## 3 Multi-Armed Bandits

### 3.1 Effective Dimension and Regret Bounds

The main result of this section is that censorship effectively enlarges the dimension of the problem. We define the effective dimension as $d_{eff} \triangleq \sum_{a \in [d]} \frac{1}{p_a}$ and our result (Thm. 3.1) shows that, at first order, the regret is guaranteed to be the same as the uncensored problem with $d_{eff}$ arms instead of $d$.

**Theorem 3.1.** *Under censorship, the UCB algorithm with regularization $\lambda$ has an instance-independent expected regret of:*

$$\mathbb{E}[R(T, \pi_{UCB})] \leq \tilde{\mathcal{O}}(\sigma \sqrt{d_{eff} T}).$$

Furthermore, we obtain analogous regret guarantees for instance-dependent cases where, at first order, the uncensored dimension $\sum_{a \neq a^\star} \frac{\sigma^2}{\Delta_a}$ enlarges to $\sum_{a \neq a^\star} \frac{\sigma^2}{p_a \Delta_a}$:

**Proposition 3.2.** *For a fixed action set $\mathcal{A}_t \equiv [d]$ and for a-priori known action gap $\Delta_a \triangleq \max_{\tilde{a}} \theta_{\tilde{a}}^\star - \theta_a^\star$, the UCB algorithm with regularization $\lambda$ has the instance-dependent expected regret:*

$$\mathbb{E}[R(T, \pi_{UCB})] \leq \mathcal{O}\Big( \log(T) \sum_{a \neq a^\star} \frac{1}{p_a} \max(\frac{\sigma^2}{\Delta_a}, \Delta_a) \Big).$$

On one hand, a preliminary understanding of censorship posits an increase of the average "*regret per information gain*" [23] (as it takes longer on average to get the same amount of information) but does not change the underlying complexity of the problem. One the other hand, our results (Thm. 3.1 and Prop. 3.2) postulate that the censored problem is equivalent at first order to a higher dimensional problem but explored with the same *regret per information gain*.

The abovementioned results extends to a-priori known heteroskedasticity (see Rem. 3 and 4 in App. B). For this general setting, the effective dimension for instance-independent (resp. dependent) case is given by $\sum_a \frac{\sigma_a^2}{p_a}$ (resp. $\sum_{a \neq a^\star} \frac{\sigma_a^2}{p_a \Delta_a}$), where $\sigma_a^2$ is the variance proxy of arm $a$. Although the scaling in $\sum_a \frac{1}{\Delta_a p_a}$ was already mentioned in [24] for unregularized setting with homogeneous variance $\sigma^2$ and proven to be optimal, our results generalize these findings.

## 3.2 Cumulative Censored Potential

We now provide a proof sketch of Thm. 3.1, and in doing so, we instantiate an analysis framework that will be extended in Sec. 4. This proof consists in the successive elimination of the noise induced by the feedback and censorship. This leads to regret guarantees on a resulting deterministic quantity by characterizing worst-case learning conditions. The first step of the proof is a variant of the classical reduction of the UCB regret to another quantity we refer to as the *expected cumulative censored potential*. Before stating it, we define at the end of a round $t \in [T]$, the random number of times an arm $a$ has been *pulled* as $\tau_a(t) \triangleq \sum_{l=1}^t \mathbf{1}\{a_l = a\}$. Similarly, the number of times an action $a$ has been *realized* at the end of round $t$ is denoted $N_a(t) \triangleq \sum_{l=1}^t \mathbf{1}\{a_l = a, x_{a_l} = 1\}$. We then have:

**Lemma 3.3.** *Given an uniform regularization of $\lambda > 0$, the UCB algorithm verifies:*

$$\mathbb{E}[R(T, \pi_{UCB})] \leq 2\sqrt{6\sigma^2 \log(T)} \mathbb{E}[\mathbb{V}_{\frac{1}{2}}(T, \pi_{UCB})] + 2\lambda \|\theta^\star\|_\infty \mathbb{E}[\mathbb{V}_1(T, \pi_{UCB})] + \frac{2d\Delta_{max}}{T}$$

*where, for any $\alpha > 0$ and $\pi \in \Pi$, the cumulative potential under censorship is given by:*

$$\mathbb{V}_\alpha(T, \pi) = \sum_{t=1}^T (N_{a_t}(t-1) + \lambda)^{-\alpha}.$$

Without censorship, the cumulative potential translates the average rate of decay of uncertainty on the reward of different arms and is closely linked to the divergence between the true reward distribution and the empirical distribution of observed rewards [38]. Introducing censorship transforms the classical deterministic decay rate into a stochastic one. For a typical reward distribution, the rate of decay is proportional to a term in $n^{-\alpha}$ or can be upper bounded by such a term (see for e.g. [38]), where $n$ is the number of *observed* rewards. Therefore, a higher $\alpha$ corresponds to faster learning.

In contrast to the classical non-regularized analysis or to the LCB case of Sec. 4, we observe two different orders of $\alpha$ (1/2 and 1) coming from the use of the $L_\infty$-norm instead of the $L_2$-norm. Taken independently, they lead to respective contributions of $\mathcal{O}(d_{eff} \log(T))$ and $\mathcal{O}(\sqrt{d_{eff} T})$. Note that by working with a general $\alpha$, our analysis naturally extends beyond sub-Gaussian noise to more general assumptions about the Laplace transform of noise (e.g., lighter or heavier tails), as discussed in Rem. 2. To further study $\mathbb{V}_\alpha$, we introduce the following property:

**Proposition 3.4.** *For all $\alpha > 0$, $\delta \in ]0, 1]$ and given $\psi_\alpha$ a primitive of $x \mapsto x^{-\alpha}$, we have:*

$$\max_{\pi \in \Pi} \mathbb{E}[\mathbb{V}_\alpha(T, \pi)] \leq \frac{d_{eff}}{(1-\delta)^\alpha} \left[ \psi_\alpha \left( \frac{T}{d_{eff}} + \frac{\lambda}{1-\delta} \right) - \psi_\alpha \left( \frac{\lambda}{1-\delta} \right) \right] + \frac{24 d_{eff} \log(T) + d}{\lambda^\alpha} + \frac{4 d_{eff}}{\lambda^\alpha \delta^2 T^{12\delta^2}}.$$

The proof of this proposition involves two steps: firstly, we remove the stochastic dependence induced by the censorship through concentration properties (See App. B), and we then solve the resulting policy maximization problem (Lemma 3.5). In the first step, we consider for a given $\delta \in ]0, 1]$ the event:

$$\mathcal{H}_{CEN}(\delta) = \{\exists a \in [d], t \in [T], N_a(t) < (1-\delta)p_a \tau_a(t) \quad \text{and} \quad \tau_a(t) \geq T_0(a)\},$$

where $T_0(a) \triangleq 24 \log(T)/p_a + 1$ and claim that $\mathbb{P}(\mathcal{H}_{CEN}(\delta)) \leq \frac{4d_{eff}}{\delta^2} T^{-12\delta^2}$, improving a result of [24]. Here $\mathcal{H}_{CEN}$ denotes the event where there is a significant gap between the realized and expected

number of observed rewards. We consider its complement in our analysis of the principal order of regret. This allows us to lower bound for each action, the realized number of reward observations by a multiple of the number of times that action was selected, thus eliminating the randomness induced by censoring.

Our second step makes use of the following lemma (also known as a *water-filling process* in information theory [14]):

**Lemma 3.5.** *For $\psi_\alpha$ a primitive of $x \mapsto x^{-\alpha}$ where $\alpha \in ]0,1]$, regularization $(\lambda_a)_{a \in [d]} \in (\mathbb{R}_{>0})^d$ and censorship vector $(p_a)_{a \in [d]}$, the solution of the optimization problem:*

$$\max_{\tau_1 \dots, \tau_d \geq 0} \sum_{a \in [d]} \frac{1}{p_a} \Big( \psi_\alpha(p_a \tau_a + \lambda_a) - \psi_\alpha(\lambda_a) \Big) \quad s.t. \quad \sum_{a \in [d]} \tau_a = T$$

*is given by $\tau_a^\star = \frac{1}{p_a}[C - \lambda_a]^+$, where $C$ ensures the total budget constraint $\sum_{a \in [d]} \tau_a^\star = T$. In particular, with $\lambda_{eff} \triangleq \frac{1}{d_{eff}} \sum_{a \in [d]} \frac{\lambda_a}{p_a}$ and $\lambda_a^0 \triangleq d_{eff}(\lambda_a - \lambda_{eff})$, the optimal solution is given by $\tau_a^\star \triangleq \frac{1}{p_a d_{eff}}(T - \lambda_a^0)$ for $T \geq \max_a \lambda_a^0$ and the optimal value is $d_{eff} \psi_\alpha(\frac{T}{d_{eff}} + \lambda_{eff}) - \sum_{a \in [d]} \frac{1}{p_a} \psi_\alpha(\lambda_a)$.*

For unregularized algorithms, this framework can be easily applied to provide instances-dependent guarantees by adding constraints of type $\tau_a \leq f(\Delta_a)$ within Lemma 3.5. Optimal guarantees under regularization such as the ones given in Prop. 3.2 require however to consider both orders of $\mathbb{V}_\alpha$ ($1/2$ and 1) simultaneously and not independently, leading to slight variations as shown in the proof of Prop. 3.2. Next, we further discuss the properties of $\mathbb{V}_\alpha$ given its importance in our analysis.

### 3.3 Evaluating Adaptivity Gain

It is well known that adaptivity is a key feature of sequential decision problems: optimal policies use feedback from previous decisions to decide the next action to take based on the data, and in comparison non-adaptive policies can be quite suboptimal. Somewhat interestingly, the main result of this section is that adaptivity in the context of censoring does not provide a significant advantage to the decision maker. More precisely, being able to observe which decisions have been censored and adapting to this information does not bring more than a second order gain. In proving this result, we quantify and gain insight into the expected performance of policies that are adaptive to the realization of the censorship process, in comparison to a class of non-adaptive (i.e., offline) policies.

In fact, through the introduction of $\mathcal{H}_{CEN}(\delta)$ and for any $\alpha \in [0,1]$, $\delta \in ]0,1]$, we showed in Prop. 3.4 the upper bound $\frac{d_{eff}}{(1-\delta)^\alpha} \psi_\alpha(\frac{T}{d_{eff}} + \frac{\lambda}{1-\delta})$ for the learning complexity $\max \mathbb{E}[\mathbb{V}_\alpha(T, \pi)]$ where the maximum is taken over the class of adaptive policies $\Pi_{adapt}$, i.e., measurable with respect to the censorship. Note that the exact value of such maximum is notoriously difficult to study due to the adaptive nature of censorship induced by the decision-making process. Next, we introduce $\Pi_{off}$, the class of policies that are not adaptive with respect to the censorship and we prove that :

**Lemma 3.6.** *For $\alpha \in ]0,1]$ and $\lambda > 0$, we have $\max\limits_{\pi \in \Pi_{off}} \mathbb{E}[\mathbb{V}_\alpha(T, \pi)] \sim d_{eff} \psi_\alpha(\frac{T}{d_{eff}} + \lambda)$.*

In other words, restricting attention to offline policies is sufficient to obtain the correct scaling. The next step to complete our claim is the asymptotic expansion:

**Proposition 3.7.** *For $\alpha \in ]0,1]$, by denoting $\gamma_\alpha(\mathbf{p}) \triangleq \frac{\alpha}{2 d_{eff}^{1-\alpha}} \sum_{a \in [d]} \frac{1}{p_a} \Big( \sum_{\tilde{a} \neq a} \frac{1 - p_{\tilde{a}}}{p_{\tilde{a}}} \Big)$, we have:*

$$\max_{\pi \in \Pi_{adapt}} \mathbb{E}[\mathbb{V}_\alpha(T, \pi)] - \max_{\pi \in \Pi_{off}} \mathbb{E}[\mathbb{V}_\alpha(T, \pi)] = \gamma_\alpha(\mathbf{p}) \frac{1}{T^\alpha} + o(\frac{1}{T^\alpha}). \qquad (\star)$$

*Moreover, if for a given $\beta \in ]0,1[$, we introduce $\Pi_{single}(\beta T)$ the policy class whose censorship information set has a single updating at time $\lfloor \beta T \rfloor$, we have:*

$$\max_{\pi \in \Pi_{single}(\beta T)} \mathbb{E}[\mathbb{V}_\alpha(T, \pi)] - \max_{\pi \in \Pi_{off}} \mathbb{E}[\mathbb{V}_\alpha(T, \pi)] = \gamma_\alpha(\mathbf{p}) \frac{\beta}{T^\alpha} + o(\frac{1}{T^\alpha}). \qquad (\star\star)$$

Thus, $\gamma_\alpha(\mathbf{p})$ can be viewed as an adaptivity gain resulting from the continuous correction of the cumulative variance induced by the action selection process. Essentially, it is closely related to

the Jensen Gap of an appropriate random variable and the proof involves the study of the Taylor expansion of the potential function $\psi_\alpha$. ($\star\star$) tells us that a single observation of the censorship realization is sufficient to obtain a near-optimal *gain in adaptivity*. We present a proof sketch of Prop. 3.7 in App. B. This shows that censorship in MAB can be treated in an *offline* manner at first order.

## 4 Contextual Bandit

In this section, we study Linear Contextual Bandits (LCBs) under censorship. The regret analysis for the generic censorship model in Sec. 2 is significantly more complex for LCB than for MAB. This is due to the fact that different actions contribute differently to the information acquisition, leading to a non-linear phenomenon governing the trade-off between reward and information gain (see Sec. 4.4).

### 4.1 Multi-threshold Models and Regret Bounds

To address the abovementioned challenge, we now introduce a simple *multi-threshold* censorship model, which enables a precise regret analysis. In particular, we consider that feedback is censored according to the following action-dependant probability:

$$p : a \in \mathbb{B}_d \mapsto \sum_{j=0}^{k} \mathbf{1}\{\sin(\phi_j) \leq \langle a, u \rangle < \sin(\phi_{j+1})\}p_j, \qquad (\mathcal{MT})$$

where $(\phi_j)_{j \leq k+1}$ is an increasing sequence verifying $\phi_0 = -\frac{\pi}{2}$, $\phi_{k+1} = \frac{\pi}{2}$ and $u \in \mathbb{R}^d$ is a unit vector. We assume that $(p_j)_{j \leq k}$ is decreasing, i.e. the censorship is increasing with $j$ in direction $u$. Henceforth, we refer to the interval $[\sin(\phi_j), \sin(\phi_{j+1})[$ as *region j*. Note that simple models such as uniform censorship are subsumed by this family (for $k$ equals 0).

The two main features of the multi-threshold model are: the *radial* aspect (the censorship probability depends on the action through a scalar product with a given vector) and the *monotonicity* (the censorship is monotone in the value of this scalar product). Note that $\mathcal{MT}$ can be seen as a piecewise constant approximation of any Generalized Linear Model (GLM) [31]. Thus, the simplicity of this censorship model is not an inherently limiting factor on the generality of our subsequent results.

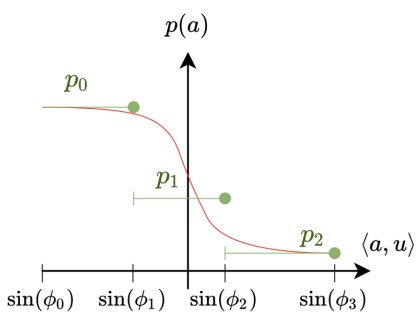

Figure 1: Example of a multi-threshold model for $k = 2$ (Green). Logistic censorship model (Red)

Moreover, $\mathcal{MT}$ admits a natural behavioral interpretation: Such a distribution can be seen as induced by a population model of heterogeneous random-utility maximizing agents. A single threshold model (i.e. $k$ equals 1) corresponds to a given agent type, and the multi-threshold model naturally results from aggregate responses of heterogeneous population [4].

We now state the main result of this section:

**Theorem 4.1.** *For a given multi-threshold censorship model $\mathcal{MT}$, there exits $d_{\text{eff}}$ such that the UCB algorithm with regularization $\lambda$ has an instance-independent expected regret of:*

$$\mathbb{E}[R(T, \pi_{UCB})] \leq \tilde{\mathcal{O}}(\sigma \sqrt{d \cdot d_{\text{eff}}} \sqrt{T}).$$

Importantly, note the mapping from the original dimension $d$ to the enlarged $\sqrt{d \cdot d_{\text{eff}}}$, in contrast to the previous dilation $d \mapsto d_{\text{eff}}$ for the case of MAB problems. An extension to Generalized Linear Contextual Bandits is provided in App. C.6 where we show that the dimension is governed by $\sqrt{d \cdot d_{\text{eff}}}/\kappa$, with $\kappa$ corresponding to a minimum of the derivative of the link function (encompassing the smoothness of the GLM at its maximum) [28, 16]. We conjecture that this result still holds if we relax the monotonicity property of $\mathcal{MT}$ although it will require some modifications in the proofs of section D. On the other hand, we believe that the radial property is necessary, considering the related literature on GLMs (further discussed in App. C.6) where it appears prominently.

## 4.2 Generalized Cumulative Censored Potential

Analogous to the MAB case, we now introduce for LCB the random matrices corresponding to the effective realization $\mathbb{W}_t^C \triangleq \lambda \mathbb{I}_d + \sum_{n=1}^t x_{a_t} a_t a_t^\top$ and the expected realization $\mathbb{W}_t \triangleq \lambda \mathbb{I}_d + \sum_{n=1}^t p(a_t) a_t a_t^\top$. We also introduce the continuous counterpart of $\mathbb{W}_t$ defined as $\mathbb{W}(t) \triangleq \lambda \mathbb{I}_d + \int_{u=0}^t p(a(u)) a(u) a(u)^\top \partial u$, where $(a(u))_{u \leq T}$ is an integrable deterministic path.[4] We emphasise that the use of continuous counterpart is key in enabling our next results. As in the MAB case, we bound the regret although now using a generalization of $\mathbb{V}_\alpha$:

**Lemma 4.2.** *For all $\delta \in ]0, 1]$, there exists a constant $\tilde{\beta}_\delta(T) = \Theta(\sqrt{d \log(T)})$ such that*

$$\mathbb{E}[R(T, \pi_{UCB})] \leq 2\tilde{\beta}_\delta(T)\sqrt{T \mathbb{E}[\mathbb{V}_1(T, \pi_{UCB})]} + \delta T \Delta_{max},$$

*where, for $\alpha > 0$ and $\pi \in \Pi$, the linear extension of the cumulative censored potential is given by:*

$$\mathbb{V}_\alpha(T, \pi) \triangleq \sum_{t=1}^T \|a_t\|^2_{(\mathbb{W}_{t-1}^C)^{-\alpha}} = \sum_{t=1}^T \operatorname{Tr}((\mathbb{W}_{t-1}^C)^{-\alpha} a_t a_t^\top).$$

The proof idea is analogous (albeit more complex) than in the finite action case (see App. C). In order to get a handle on $\mathbb{V}_\alpha$, we again leverage a two-step approach: first we eliminate the randomness due to censorship (here, we utilize matrix martingale inequalities) and then optimize the resulting deterministic quantity seen through a continuous lens. The first step requires the following result:

**Proposition 4.3.** *For any $\delta \in ]0, 1]$, $\lambda > 0$, $\alpha > 0$ and policy $\pi \in \Pi$, we have:*

$$\mathbb{E}[V_\alpha(T, \pi)] \leq \frac{\delta}{\lambda^\alpha} + C(\delta)^\alpha \operatorname{Tr}\left(\int_0^T \mathbb{W}(t)^{-\alpha} a(t) a(t)^\top \partial t\right),$$

*where $C(\delta) \triangleq 8(\lambda + 1) \max(\log(d/\delta))/\lambda, 1)/\lambda$.*

The key idea of this result is to observe that the telescopic sum on which the classical Elliptical Potential lemma [1, 35, 8] heavily relies on is, in fact, the discrete approximation of an integral over a matrix path. This critical methodological contribution is further discussed in Rem. 1 and 5.

**Remark 1.** *One way to fully appreciate the generality of this result is to consider the simpler case of classical uncensored environment for which we obtain for $\alpha > 0, \alpha \neq 1$:*

$$\sum_{t=1}^T \|a_t\|^2_{\mathbb{W}_{t-1}^{-\alpha}} \leq \left(\frac{\lambda + 1}{\lambda}\right)^\alpha \frac{\operatorname{Tr}\left(\int_0^T \partial \mathbb{W}(t)^{1-\alpha}\right)}{1 - \alpha} = \left(\frac{\lambda + 1}{\lambda}\right)^\alpha \frac{\operatorname{Tr}(\mathbb{W}_T^{1-\alpha} - \mathbb{W}_0^{1-\alpha})}{1 - \alpha}.$$

*For $\alpha = 1$, a similar reasoning is applied using the formula $\operatorname{Tr}(\log(A)) = \log(\det A)$:*

$$\sum_{t=1}^T \|a_t\|^2_{\mathbb{W}_{t-1}^{-1}} \leq \frac{\lambda + 1}{\lambda} \int_0^T \frac{\partial \log \det(\mathbb{W}(t))}{\partial t} \partial t = \frac{\lambda + 1}{\lambda} \operatorname{Tr}(\log \mathbb{W}_T - \log \mathbb{W}_0) = \frac{\lambda + 1}{\lambda} \log \frac{\det \mathbb{W}_T}{\det \mathbb{W}_0}.$$

*A deeper study of the eigenvalues of $\mathbb{W}_T^{1-\alpha}$ then yields the worst-case upper bound $d^\alpha(d\lambda + T)^{1-\alpha}/(1 - \alpha)$ for $\alpha < 1$ and $d\lambda^{1-\alpha}/(\alpha - 1)$ for $\alpha > 1$, recovering more naturally and extending the results of [8]. Thus, analogous to the water filling process highlighted in the MAB case in Lemma 3.5, we now consider a spectral water-filling process [14] optimizing over the eigenvalues of $\psi_\alpha(\mathbb{W}_T)$ with a slight abuse of notations ($\mathbb{W}_T^{1-\alpha}$ and $\log \mathbb{W}_T$ in this discussion).*

Following Rem.1, for the general censored case the challenge now becomes to identify a suitable matrix operator on which the aforementioned spectral maximization can be performed. By applying Lemma 4.2, we henceforth focus on the case of $\alpha = 1$ for which Prop. 4.3 implies that for any policy:

$$\operatorname{Tr}\left(\int_0^T \mathbb{W}(t)^{-1} a(t) a(t)^\top \partial t\right) = \int_0^T \frac{1}{p(a(t))} \frac{\partial \log \det(\mathbb{W}(t))}{\partial t} \partial t.$$

Next, we focus on maximizing this integral over the policy class $\Pi$ and again recover the notion of effective dimension.

---

[4]In this section, the generic notation $X(t)$ is used for continuous time quantities and $X_t$ for discrete time.

## 4.3 Effective Dimension in Linear Settings

We now highlight immediate properties of the effective dimension, and then present its general study for the multi-threshold model $\mathcal{MT}$.

**Lemma 4.4.** *Let us consider an uniform censorship model $p : a \mapsto \bar{p}$. By leveraging the case of equality in the Arithmetic-Geometric inequality applied to the eigenvalues of $\mathbb{W}_T$, we then simply deduce the associated effective dimension $d_{eff} \triangleq d/\bar{p}$:*

$$\max_{\pi \in \Pi} \int_0^T \frac{1}{\bar{p}} \frac{\partial \log \det(\mathbb{W}(t))}{\partial t} \partial t = d_{eff} \log(1 + \frac{T}{\lambda d_{eff}}).$$

In fact, the logarithmic scaling of this quantity persists while moving beyond the uniform censorship assumption. This also highlights the importance of the leading dimension factor, crudely upper bounded by $d/p_{min}$ in the next lemma:

**Lemma 4.5.** *For any censorship function $p$, by introducing lower and upper bounds $(p_{min}, p_{max})$ of $p$, we have:*

$$\frac{d}{p_{max}} \log(1 + \frac{p_{min}T}{d\lambda}) \leq \max_{\pi \in \Pi} \int_0^T \frac{1}{p(a(t))} \frac{\partial \log \det(\mathbb{W}(t))}{\partial t} \partial t \leq \frac{d}{p_{min}} \log(1 + \frac{p_{max}T}{d\lambda}).$$

Related problems in the Generalized Linear Models literature [47, 28, 16] are implicitly solved in the spirit of Lemma 4.5, where a minimum of the derivative of the link function plays the role of $p_{min}$ above. However, when the function $p$ varies with action $a$, a more careful analysis is required to derive useful dimensional bounds. Our next major result addresses this gap in the literature by improving the bounds provided in Lemma 4.5:

**Theorem 4.6.** *For a multi-threshold censorship model $\mathcal{MT}$, we have:*

$$\max_{\pi \in \Pi} \int_0^T \frac{1}{p(a(t))} \frac{\partial \log \det(\mathbb{W}(t))}{\partial t} \partial t = d_{eff} \log(T) + o(\log(T)), \tag{$\mathcal{P}$}$$

*where $d_{eff}$ is the effective dimension. Furthermore, $d_{eff}$ is characterized by two cases:*

- **Case 1:** *Single region $j$ effective dimension $d_{eff} = \frac{d}{p_j}$.*

- **Case 2:** *Bi-region $(i, j)$ effective dimension, with $i < j$:*

$$d_{eff} = \frac{1}{p_j} \left[ (d-1)\frac{1 - l(i,j)}{\frac{p_i}{p_j} - l(i,j)} + \frac{u(i,j) - 1}{u(i,j) - \frac{p_i}{p_j}} \right] < \frac{d}{p_j}. \tag{$\mathcal{D}$}$$

*where $l(i,j) \triangleq \frac{\sin^2(\phi_i)}{\sin^2(\phi_j)}$ and $u(i,j) \triangleq \frac{\cos^2(\phi_i)}{\cos^2(\phi_j)}$.*

The implications of these cases are further discussed in Fig. 2 in App. D. Notice that a necessary condition for the bi-region $(i, j)$ effective dimension to arise is the constraint on $\frac{p_i}{p_j}$:

$$\max(1, \underbrace{\frac{dl(i,j)u(i,j)}{u(i,j) + (d-1)l(i,j)}}_{\triangleq s^\star(i,j)}) < \frac{p_i}{p_j} < \underbrace{\frac{(d-1)u(i,j) + l(i,j)}{d}}_{\triangleq r^\star(i,j)}$$

In the limit $\frac{p_i}{p_j} \to r^\star(i,j)$, $d_{eff}$ goes again to $d/p_j$. We interpret this limiting case as *locally hard* in the sense that censorship in region $j$ is sufficiently important in comparison to all other regions to impose a maximal effective dimension to the problem, irrespective of the values of $p_i$, matching Lemma 4.5. On the other hand, for the other limiting case (under additional mild assumptions), we find that $d_{eff}$ also goes to $d/p_j$, but now for a *uniformly hard* reason: that is, censorship is approximately constant and equal to $p_j$, recovering the Lemma 4.4. Finally, in between these two extremes lies the *minimum effective dimension* for a given value of $\frac{p_i}{p_j}$.

## 4.4 Temporal dynamics of $\mathbb{W}(t)$

The proof of Thm. 4.6 requires the characterization of the dynamics of the optimal policy of ($\mathcal{P}$). Importantly, we discover that the evolution of $\mathbb{W}(t)$ is described by two qualitatively different regimes as outlined next. It turns out that our continuous approach to analyzing cumulative censored potential is an important tool to obtaining this result.

**Transient Regime:** There exists a decreasing sequence of censorship regions $\{i_1 = k, \ldots, i_l\}$ of length $l \in [k+1]$ and associated time sequence $\{t_0 \triangleq 0, t_1, \ldots, t_l\}$ such that whenever $t_j \leq t \leq t_{j+1}$ for a given index $j \leq l-1$, the evolution of $\mathbb{W}(t)$ is given by:

$$\mathbb{W}(t) = p_{i_{j+1}}(t - t_j)\mathbb{W}_{i_{j+1}} + \mathbb{W}(t_j) = p_{i_{j+1}}(t - t_j)\mathbb{W}_{i_{j+1}} + \sum_{n=1}^{j} p_{i_n}(t_n - t_{n-1})\mathbb{W}_{i_n} + \lambda\mathbb{I}_d,$$

where $\mathbb{W}_i$ denotes the $d \times d$ diagonal matrix $\mathrm{diag}(\frac{\cos^2(\phi_i)}{d-1}, \ldots, \frac{\cos^2(\phi_i)}{d-1}, \sin^2(\phi_i))$. Interestingly, the initial misspecification of censorship is self-corrected during this transient step but at an extra cost. This characterization of transient regime highlights an important consequence of using classical algorithms in censored environments.

**Steady State Regime:** Post-transient regime, the dynamics of $\mathbb{W}(t)$ enter a steady state regime, where one of the two cases necessarily arise:[5].

- **Case 1: Single region $i_l$.** This case arises when the last element of the time sequence $t_l$ is equal to $+\infty$ and we have the single region evolution for all $t \geq t_{l-1}$:

$$\mathbb{W}(t) = p_{i_l}(t - t_{l-1})\mathbb{W}_{i_l} + \mathbb{W}(t_{l-1}) = p_{i_l}(t - t_{l-1})\mathbb{W}_{i_l} + \sum_{n=1}^{l-1} p_{i_n}(t_n - t_{n-1})\mathbb{W}_{i_n} + \lambda\mathbb{I}_d.$$

  The effective dimension corresponding to this dynamics is $d/p_{i_l}$, with the following equality for $T \geq t_{l-1}$:

$$\int_0^T \frac{1}{p(a(t))} \frac{\partial \log \det(\mathbb{W}(t))}{\partial t} \partial t = \frac{1}{p_{i_l}} \log \det(\mathbb{W}(T)) + \sum_{n=1}^{l-1} \left(\frac{1}{p_{i_n}} - \frac{1}{p_{i_{n+1}}}\right) \log \det \mathbb{W}(t_n).$$

- **Case 2: Bi-region $(i_{l+1}, i_l)$.** This case arises when the steady-state dynamics of $\mathbb{W}(t)$ span the two regions $(i_{l+1}, i_l)$ with $i_{l+1} < i_l$. For all $t \geq t_l$, we have the evolution:

$$\mathbb{W}(t) \propto p_{i_{l+1}}(t + \lambda^\star) \begin{pmatrix} \cos^2(\phi_{i_l})(u(i_{l+1}, i_l) - \frac{p_{i_{l+1}}}{p_{i_l}})\mathbb{I}_{d-1} & (0) \\ (0) & \sin^2(\phi_{i_l})(\frac{p_{i_{l+1}}}{p_j} - l(i_{l+1}, i_l)) \end{pmatrix}.$$

  where $\lambda^\star$ and the proportionality factor are specified in SI. The corresponding effective dimension is given by ($D$) and the following equality holds for all $T \geq t_l$:

$$\int_0^T \frac{1}{p(a(t))} \frac{\partial \log \det(\mathbb{W}(t))}{\partial t} \partial t = d_{eff} \log(1 + \frac{T - t_l}{t_l + \lambda^\star}) + \sum_{n=1}^{l} \left(\frac{1}{p_{i_n}} - \frac{1}{p_{i_{n+1}}}\right) \log \det \mathbb{W}(t_n).$$

For further discussions on transient and steady state regimes, we refer to Fig.3, 4 and 5, in App. D.

## 5 Concluding Remarks

In this work, we demonstrate that the complexity of bandit learning under censorship is governed by the notion of effective dimension. To do so, we developed a novel analysis framework which enables us to precisely estimate this quantity for a broad class of multi-threshold censorship models. An important future work would be to extend our model and approach to Bayesian settings, which will likely provide us with useful insights on the cumulative censored potential $\mathbb{V}_\alpha$, as initiated by [18]. Future work also includes relaxing the Missing Completely at Random (MCAR) property in favor of time-dependent censorship models such as Markov Decision Processes (MDPs). We believe that tools similar to those developed in our potential-based analysis can be applied in this case. Finally, the contributions of our work may be of interest to the recent value alignment literature, where the question of learnability under humain-AI interactions is central. [10, 17, 12].

We do not envision any negative societal impacts of our work other than that of bandits algorithms deployed in AI-driven platforms.

---

[5] These cases are fully characterized in terms of parameters of censorship model in Lemmas D.1, D.2, D.3 and Cor. D.1.1.

## Acknowledgments and Disclosure of Funding

This research project is supported by the AFOSR FA9550-19-1-0263 "Building attack resilience into complex networks" Grant. The authors would like to thank Prem Talwai and the anonymous reviewers for providing insightful comments and suggestions.

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
