# A  Preliminaries

In this section, we first provide the instances of the UCB algorithm used in Sec. 3 and Sec. 4. We also indicate in Tab. 1 the notations used throughout the paper to help the reader.

## A.1  UCB algorithms

- **UCB-MAB:** Following [26], the UCB algorithms for the MAB case with homogeneous regularization $\lambda > 0$ uses the following optimistic reward estimator at time $t$:

$$\tilde{r}_t^\lambda(a) \triangleq \hat{\theta}_t^\lambda(a) + \sqrt{\frac{6\sigma^2 \log(T)}{\lambda + N_a(t-1)}} + \frac{\lambda \|\theta^\star\|_\infty}{\lambda + N_a(t-1)}.$$

It is based on the use of the regularized empirical mean to estimate the reward of action $a$ at the end of round $t$:

$$\hat{\theta}_t^\lambda(a) \triangleq \frac{1}{N_a(t) + \lambda} \sum_{\tau=1}^t (r(a_\tau) + \tau)\mathbf{1}\{a_\tau = a, x_{a_\tau} = 1\}$$

$$= \frac{N_a(t)}{N_a(t) + \lambda}\theta_a^\star + \frac{1}{N_a(t) + \lambda} \sum_{\tau=1}^t \epsilon_{a_\tau}\mathbf{1}\{a_\tau = a, x_{a_\tau} = 1\}.$$

The high-confidence property of this algorithm is proven in Lemma B.1.[6] Under a-priori known heteroskedasticity, the reward estimator can be expressed as:

$$\tilde{r}_t^\lambda(a) \triangleq \hat{\theta}_t^\lambda(a) + \sqrt{\frac{6\sigma_a^2 \log(T)}{\lambda + N_a(t-1)}} + \frac{\lambda \|\theta^\star\|_\infty}{\lambda + N_a(t-1)}.$$

- **UCB for LCB** Following [1, 26], the UCB algorithms for the LCB case with homogeneous regularization $\lambda > 0$ uses the following optimistic reward estimator at time $t$:

$$\tilde{r}_t^\lambda(a) \triangleq \langle a, \hat{\theta}_{t-1}^\lambda \rangle + \beta_{t-1}(\delta)\|a\|_{\mathbb{W}_{t-1}^C},$$

where we introduced the random quantity:

$$\beta_{t-1}(\delta) \triangleq \sqrt{\sigma^2 \log\left(\frac{\det(\mathbb{W}_{t-1}^C)}{\det(\lambda\mathbb{I}_d)}\right) + 2\sigma^2 \log(\frac{1}{\delta})} + \sqrt{\lambda}\|\theta^\star\|_2$$

It is based on the use of the regularized least square estimator to estimate the vector $\theta^\star$ at the end of round $t$:

$$\hat{\theta}_t^\lambda = (\mathbb{W}_t^C)^{-1} \sum_{\tau=1}^t (\epsilon_\tau + \langle a_\tau, \theta^\star \rangle)x_{a_\tau} a_\tau$$

The high-confidence property of this estimator is proven in Lemma C.1.

# B  Proof of Sec. 3 - Multi-Armed Bandits

In this section, we prove the results in Sec. 3 on the MAB case. We start by proving Lemmas 3.3, B.1, B.2, 3.5 and Prop. 3.4. Thanks to those results, we then tackle Thm. 3.1 and Prop. 3.2. To conclude the section, we further study the properties of the adaptivity gain, by proving Lemma 3.6 and Prop. 3.7. Recall that effective dimension $d_{eff}$ is referring to $\sum_{a\in[d]} \frac{1}{p_a}$ in this section.

---

[6]Typically, an upper bound on $\|\theta^\star\|_\infty$ for MAB (resp. $\|\theta^\star\|_2$ for LCB) is used instead of this unknown quantity. We keep $\|\theta^\star\|_\infty$ (resp. $\|\theta^\star\|_2$) not to overload notations but our results immediately extends to the use of the latter.

Table 1: Summary of Notations

### Bandit Problem Variables

| | | |
|---|---|---|
| $T$ | $\triangleq$ | Total number of rounds of the sequential decision-making problem. |
| $d$ | $\triangleq$ | Number of arms in Sec.3, Dimension of action feature vector in Sec.4. |
| $(\mathcal{A}_t, \mathcal{A})$ | $\triangleq$ | Action set at time $t$; Union of all action sets $\mathcal{A}_t$. |
| $a_t$ | $\triangleq$ | Action picked at time $t$; selected by policy $\pi$, seen as a function of previous history. |
| $(\epsilon_t, \sigma^2)$ | $\triangleq$ | Stochastic feedback noise a time $t$. Sub-Gaussian with pseudo-variance parameter $\sigma^2$. If $\sigma^2$ depends on the action selected (heteroskedasticity), we use $\sigma_a^2$ instead. |
| $(r, \theta^\star)$ | $\triangleq$ | Unknown reward function, maps action to scalar reward. Parameterized by unknown latent state $\theta^\star$. |
| $\Delta_t(a)$ | $\triangleq$ | Sub-optimality gap of action $a$ at time $t$, reward difference with optimal decision of clairvoyant policy |
| $(\Delta_a, \Delta_{\max})$ | $\triangleq$ | If $\Delta_t(a)$ is independent of $t$, we use $\Delta_a \equiv \Delta_t(a)$. $\Delta_{\max}$ is an upper bound of $\Delta_t(a)$ for all actions $a$ and time $t$. |
| $R(T, \pi)$ | $\triangleq$ | Pseudo regret of policy $\pi$ over $T$ rounds. |

### Censorship Variables

| | | |
|---|---|---|
| $p_a$ | $\triangleq$ | Probability that action $a$ is censored if selected, used in Sec. 3. Notation $p(a)$ is used in Sec.4 to emphasize the dependency of $p$ on action $a$. |
| $(\phi_j, u, p_j)$ | $\triangleq$ | Parameters of the multi-threshold censorship model. Vector $u$ defines the direction of censorship, $(\phi_j)_{j \leq k+1}$ define the censorship regions with fixed censorship probability and $(p_j)_{j \leq k}$ define the probability of being censored for each region $j$. |
| $x_{a_t}$ | $\triangleq$ | Random variable indicating if feedback is censored as round $t$. Follows i.i.d Bernoulli distribution of parameter $p(a_t)$. |

### Algorithmic and Analysis Variables

| | | |
|---|---|---|
| $\lambda$ | $\triangleq$ | Regularization tuning parameter. $\lambda_a$ is used if heterogeneous action-based regularization. |
| $\tilde{\Delta}_t^\lambda(a)$ | $\triangleq$ | High-probability upper bound on the sub-optimality gap, used in UCB algorithms. |
| $\mathbb{V}_\alpha(T, \pi)$ | $\triangleq$ | Random cumulative censored potential, seen as a function of policy $\pi$ and number of rounds $T$. First introduced in Sec.3 and extended in Sec.4. |
| $\psi_\alpha$ | $\triangleq$ | Primitive of the function $x \mapsto x^{-\alpha}$, for a given $\alpha > 0$. |
| $N_a(t)$ | $\triangleq$ | Total number of time action $a$ is *realized* at the end of round $t$ by policy $\pi$. Used in Sec.3. |
| $\tau_a(t)$ | $\triangleq$ | Total number of time action $a$ is *played* at the end of round $t$ by policy $\pi$. Used in Sec.3. |
| $\mathbb{W}_t^C$ | $\triangleq$ | Censored Design Matrix. Linear generalization of $(N_a(t))_{a \in [d]}$. Used in Sec.4. |
| $\mathbb{W}_t$ | $\triangleq$ | Expected Design Matrix. Linear generalization of $(p_a \tau_a(t))_{a \in [d]}$. Used in Sec.4. |
| $\mathbb{W}(t)$ | $\triangleq$ | Continuous generalization of the expected design matrix $\mathbb{W}_t$. |

## B.1 Proof of Lemma 3.3

**Lemma 3.3.** *Given an uniform regularization of $\lambda > 0$, the UCB algorithm verifies:*

$$\mathbb{E}[R(T, \pi_{UCB})] \le 2\sqrt{6\sigma^2 \log(T)}\mathbb{E}[\mathbb{V}_{\frac{1}{2}}(T, \pi_{UCB})] + 2\lambda\|\theta^\star\|_\infty\mathbb{E}[\mathbb{V}_1(T, \pi_{UCB})] + \frac{2d\Delta_{max}}{T}$$

*where, for any $\alpha > 0$ and $\pi \in \Pi$, the cumulative potential under censorship is given by:*

$$\mathbb{V}_\alpha(T, \pi) = \sum_{t=1}^{T}(N_{a_t}(t-1) + \lambda)^{-\alpha}.$$

*Proof.* At a given round $t \in [T]$, we have under the event $\neg\mathcal{H}_{\text{UCB}}^\lambda$ introduced in Lemma B.1:

$$\Delta_t(a_t) = \max_{a \in \mathcal{A}_t}\theta_a^\star - \theta_{a_t}^\star \le 2\sqrt{6\sigma^2\frac{\log(T)}{N_{a_t}(t-1) + \lambda}} + 2\frac{\lambda\|\theta^\star\|_\infty}{\lambda + N_{a_t}(t-1)},$$

where the inequality comes from the definition of the UCB algorithm and the conditioning on $\neg\mathcal{H}_{\text{UCB}}^\lambda$. We find there the origin of the two different orders of $N_a$ ($1/2$ and $1$). Taken independently, those lead to a contribution of respectively $\mathcal{O}(d_{eff}\log(T))$ and $\mathcal{O}(\sqrt{d_{eff}T})$. More precisely, we have:

$$R(T, \pi_{\text{UCB}}|\neg\mathcal{H}_{\text{UCB}}^\lambda) \le 2\sqrt{6\sigma^2\log(T)}\sum_{t=1}^{T}\sqrt{\frac{1}{N_{a_t}(t-1) + \lambda}} + 2\lambda\|\theta^\star\|_\infty\sum_{t=1}^{T}\frac{1}{N_{a_t}(t-1) + \lambda}$$

$$= 2\sqrt{6\sigma^2\log(T)}\mathbb{V}_{\frac{1}{2}}(T, \pi_{\text{UCB}}) + 2\lambda\|\theta^\star\|_\infty\mathbb{V}_1(T, \pi_{\text{UCB}}).$$

Therefore, thanks to Lemma B.1, we deduce that:

$$R(T, \pi_{\text{UCB}}) \le (1 - \mathbb{P}(\mathcal{H}_{\text{UCB}}^\lambda))R(T, \pi_{\text{UCB}}|\neg\mathcal{H}_{\text{UCB}}^\lambda) + \mathbb{P}(\mathcal{H}_{\text{UCB}}^\lambda)\Delta_{max}T$$

$$\le 2\sqrt{6\sigma^2\log(T)}\mathbb{V}_{\frac{1}{2}}(T, \pi_{\text{UCB}}) + 2\lambda\|\theta^\star\|_\infty\mathbb{V}_1(T, \pi_{\text{UCB}}) + \frac{2d\Delta_{max}}{T}.$$

Finally, we conclude that:

$$\mathbb{E}[R(T, \pi_{\text{UCB}})] \le 2\sqrt{6\sigma^2\log(T)}\mathbb{E}[\mathbb{V}_{\frac{1}{2}}(T, \pi_{\text{UCB}})] + 2\lambda\|\theta^\star\|_\infty\mathbb{E}[\mathbb{V}_1(T, \pi_{\text{UCB}})] + \frac{2d\Delta_{max}}{T}.$$

$\square$

## B.2 Statement and Proof of Lemma B.1

The main step in this reduction from regret to cumulative censored potential is the study of the *failure of optimism* event thanks to the following result:

**Lemma B.1.** *For a regularization $\lambda > 0$ and $\delta \in ]0, 1]$, we introduce the event:*

$$\mathcal{H}_{UCB}^\lambda = \left\{\exists a \in [d], t \in [T], |\hat\theta_t^\lambda(a) - \theta_a^\star| > \sqrt{\frac{6\sigma^2\log(T)}{\lambda + N_a(t)}} + \frac{\lambda\|\theta^\star\|_\infty}{\lambda + N_a(t)}\right\}.$$

*We then have $\mathbb{P}(\mathcal{H}_{UCB}^\lambda) \le \frac{2d}{T^2}$.*

*Proof.* Although this event is similar to the one introduced in the classical UCB proof idea, the subtlety comes from the randomness induced by the censorship as well as the impact of regularization. The main idea is adopt a worst-case agnostic approach. First, let's note that for a given $t \in [T], a \in [d]$, we have:

$$|\hat\theta_t^\lambda(a) - \theta_a^\star| = |\frac{1}{N_a(t) + \lambda}\sum_{\tau=1}^{t}\epsilon_\tau\mathbf{1}\{a_\tau = a, x_{a_\tau} = 1\} - \frac{\lambda}{N_a(t) + \lambda}\theta_a^\star|$$

$$\le |\frac{1}{N_a(t) + \lambda}\sum_{\tau=1}^{t}\epsilon_\tau\mathbf{1}\{a_\tau = a, x_{a_\tau} = 1\}| + \frac{\lambda}{N_a(t) + \lambda}\|\theta^\star\|_\infty.$$

Therefore, for a given $a \in [d], t \in [T]$, by introducing the event $\mathcal{B}_{(t,a)} \triangleq \left\{ |\hat{\theta}_t^\lambda(a) - \theta_a^\star| > \sqrt{\frac{6\sigma^2 \log(T)}{\lambda + N_a(t)}} + \frac{\lambda \|\theta^\star\|_\infty}{\lambda + N_a(t)} \right\}$, we deduce:

$$\mathcal{B}_{(t,a)} \subset \left\{ |\frac{1}{N_a(t) + \lambda} \sum_{\tau=1}^{t} \epsilon_\tau \mathbf{1}\{a_\tau = a, x_{a_\tau} = 1\}| + \frac{\lambda}{N_a(t) + \lambda} \|\theta^\star\|_\infty > \sqrt{\frac{6\sigma^2 \log(T)}{\lambda + N_a(t)}} + \frac{\lambda \|\theta^\star\|_\infty}{\lambda + N_a(t)} \right\}$$

$$\subset \left\{ |\frac{1}{N_a(t) + \lambda} \sum_{\tau=1}^{t} \epsilon_\tau \mathbf{1}\{a_\tau = a, x_{a_\tau} = 1\}| > \sqrt{\frac{6\sigma^2 \log(T)}{\lambda + N_a(t)}} \right\}.$$

Then, we have:

$$\mathbb{P}(\mathcal{H}_{\text{UCB}}^\lambda) = \mathbb{P}\Big( \bigcup_{a \in [d]} \bigcup_{t \in [T]} \mathcal{B}_{(t,a)} \Big)$$

$$\leq \mathbb{P}\Big( \bigcup_{a \in [d]} \bigcup_{t \in [T]} \Big\{ |\frac{1}{N_a(t) + \lambda} \sum_{\tau=1}^{t} \epsilon_\tau \mathbf{1}\{a_\tau = a, x_{a_\tau} = 1\}| > \sqrt{\frac{6\sigma^2 \log(T)}{\lambda + N_a(t)}} \Big\} \Big)$$

$$\leq \sum_{a \in [d]} \mathbb{P}\Big( \bigcup_{t \in [T]} \Big\{ |\frac{1}{N_a(t) + \lambda} \sum_{\tau=1}^{t} \epsilon_\tau \mathbf{1}\{a_\tau = a, x_{a_\tau} = 1\}| > \sqrt{\frac{6\sigma^2 \log(T)}{\lambda + N_a(t)}} \Big\} \Big)$$

$$= \sum_{a \in [d]} \mathbb{P}\Big( \bigcup_{k \in [T]} \bigcup_{t \in [T]} \Big\{ |\frac{1}{N_a(t) + \lambda} \sum_{\tau=1}^{t} \epsilon_\tau \mathbf{1}\{a_\tau = a, x_{a_\tau} = 1\}|^2 > \frac{6\sigma^2 \log(T)}{\lambda + N_a(t)}; N_a(t) = k \Big\} \Big)$$

$$= \sum_{a \in [d]} \sum_{k \in [T]} \mathbb{P}(N_a(t) = k) \mathbb{P}\Big( \bigcup_{t \in [T]} \Big\{ |\frac{1}{k + \lambda} \sum_{\tau=1}^{t} \epsilon_\tau \mathbf{1}\{a_\tau = a, x_{a_\tau} = 1\}|^2 > \frac{6\sigma^2 \log(T)}{k} \Big| N_a(t) = k \Big\} \Big)$$

$$\leq \sum_{a \in [d]} \sum_{k \in [T]} \mathbb{P}\Big( \bigcup_{t \in [T]} \Big\{ |\frac{1}{k + \lambda} \sum_{\tau=1}^{t} \epsilon_\tau \mathbf{1}\{a_\tau = a, x_{a_\tau} = 1\}|^2 > \frac{6\sigma^2 \log(T)}{\lambda + k} \Big| N_a(t) = k \Big\} \Big)$$

$$= \sum_{a \in [d]} \sum_{k \in [T]} \mathbb{P}\Big( |\frac{\sum_{l=1}^{k} \epsilon_l}{k + \lambda}|^2 > \frac{6\sigma^2 \log(T)}{\lambda + k} \Big),$$

where we successively used union bounds over the action set and number of realizations and conditioned over number of realizations $k$. We re-indexed the random sub-Gaussian variables $(\epsilon_t)$ for last expression thanks to the i.i.d property. Then, for a given $k$, using Hoeffding inequality for sub-Gaussian variables, we have:

$$\mathbb{P}\Big( |\frac{\sum_{l=1}^{k} \epsilon_l}{k + \lambda}|^2 > \frac{6\sigma^2 \log(T)}{k + \lambda} \Big) = \mathbb{P}\Big( |\sum_{l=1}^{k} \epsilon_l| > \sqrt{6\sigma^2 (k + \lambda) \log(T)} \Big) \leq 2 \exp\{ -\frac{6\sigma^2 (k + \lambda) \log(T)}{2k\sigma^2} \}$$

$$\leq \frac{2}{T^3}$$

where the used that fact that $\sum_{l=1}^{k} \epsilon_l$ is sub-Gaussian of pseudo-variance parameter $k\sigma^2$ Therefore, this yields:

$$\sum_{a \in [d]} \sum_{k \in [T]} \mathbb{P}\Big( |\frac{\sum_{l=1}^{k} \epsilon_l}{k + \lambda}|^2 > \frac{6\sigma^2 \log(T)}{k + \lambda} \Big) \leq \frac{2d}{T^2}.$$

Finally, we conclude that $\mathbb{P}(\mathcal{H}_{\text{UCB}}^\lambda) \leq \frac{2d}{T^2}$. $\qquad \square$

**Remark 2.** *We note that assuming tails distribution for the reward noise $\epsilon$ of the form:*

$$\mathbb{P}(\epsilon \geq x) \leq \exp\left\{ \frac{-x^{1+q}}{2\sigma^2} \right\}$$

*for a given $q > 0$, as suggested for instance in [47], would lead the use of the confidence interval:*

$$\mathcal{H}_{UCB}^{\lambda,q} = \left\{ \exists a \in [d], t \in [T], |\hat{\theta}_t^\lambda(a) - \theta_a^\star| > \left(6\sigma^2 \log(T)\right)^{\frac{1}{1+q}} \left(\lambda + N_a(t)\right)^{-\frac{q}{1+q}} + \frac{\lambda\|\theta^\star\|_\infty}{\lambda + N_a(t)} \right\}.$$

*Indeed, the same reasoning as above would then yield:*

$$\mathbb{P}\left(\left|\frac{\sum_{l=1}^k \epsilon_l}{k+\lambda}\right| > (6\sigma^2 \log(T))^{\frac{1}{1+q}}(k+\lambda)^{-\frac{a}{1+q}}\right) = \mathbb{P}\left(\left|\sum_{l=1}^k \epsilon_l\right| > (6\sigma^2(k+\lambda)\log(T))^{\frac{1}{1+q}}\right)$$

$$\leq 2\exp\{-\frac{6\sigma^2(k+\lambda)\log(T)}{2k\sigma^2})\} \leq \frac{2}{T^3}$$

*and therefore $\mathbb{P}(\mathcal{H}_{UCB}^{\lambda,q}) \leq \frac{2d}{T^2}$. For $q = 1$, we recover the sub-Gaussian case, which in turns lead to the study of $\mathbb{V}_{1/2}$, as done in Lemma 3.3. For general $q > 0$, we would would then consider $\mathbb{V}_{q/(1+q)}$, which lead to the upper bound $\mathcal{O}(d_{eff}^{q/(1+q)} T^{1/(1+q)})$ through the use of Prop. 3.4.*

### B.3 Statement and Proof of Lemma B.2

**Lemma B.2.** *For any $\delta \in ]0,1]$, $\lambda > 0$ and censorship model, let's introduce the event:*

$$\mathcal{H}_{CEN}^I(\delta) = \left\{ \exists a \in [d], t \in [T], N_a(t) < (1-\delta)p_a\tau_a(t) \quad and \quad \tau_a(t) \geq T_0(a) \right\},$$

*where $T_0(a) \triangleq 24\log(T)/p_a + 1$. We then have $\mathbb{P}(\mathcal{H}_{CEN}^I(\delta)) \leq \frac{4d_{eff}}{\delta^2} T^{-12\delta^2}$.*

*Proof.* First, we apply successively two unions bounds over the action set and the number of realizations, mirroring the analysis of [24]:

$$\mathbb{P}(\mathcal{H}_{CEN}^I(\delta)) \leq \sum_{a\in[d]} \mathbb{P}\left(\left\{\exists t \in [T], \tau_a(t) \geq T_0(a), N_a(t) < (1-\delta)p_a\tau_a(t)\right\}\right)$$

$$= \sum_{a\in[d]} \mathbb{P}\left(\bigcup_{k_a\in[T_0(a),T]} \bigcup_{t\in[T]} \left\{\tau_a(t) \geq T_0(a), N_a(t) < (1-\delta)p_a\tau_a(t), \tau_a(t) = k_a\right\}\right)$$

$$\leq \sum_{a\in[d]} \sum_{k_a\geq T_0(a)} \mathbb{P}\left(\bigcup_{t\in[T]} \left\{N_a(t) < (1-\delta)p_a\tau_a(t)\Big|\tau_a(t) = k_a\right\}\right).$$

We then use a multiplicative Chernoff inequality for Binomial Distribution to deduce:

$$\sum_{a\in[d]} \sum_{k_a\geq T_0(a)} \mathbb{P}\left(N_a(t) < (1-\delta)p_a\tau_a(t)\Big|\tau_a(t) = k_a\right) \leq \sum_{a\in[d]} \sum_{k_a\geq T_0(a)} \exp\{-\frac{\delta^2 k_a p_a}{2}\}.$$

The novelty of our proof is to leverage a integral comparison to deduce the improved control:

$$\sum_{a\in[d]} \sum_{k_a\geq T_0(a)} \exp\{-\frac{\delta^2 k_a p_a}{2}\} \leq 2\sum_{a\in[d]} \left[-\frac{2}{\delta^2 p_a}\exp\{-\frac{\delta^2 k_a p_a}{2}\}\right]_{T_0(a)-1}^{\tau_a(t)}$$

$$\leq \frac{4}{\delta^2}d_{eff}\frac{1}{T^{12\delta^2}} - \frac{4}{\delta^2}\sum_{a\in[d]}\frac{1}{p_a}\exp\{-\frac{\delta^2 \tau_a(t) p_a}{2}\} \leq \frac{4}{\delta^2}d_{eff}\frac{1}{T^{12\delta^2}}.$$

Picking for instance $\delta = \frac{1}{2}$ yields $\mathbb{P}(\mathcal{H}_{CEN}^I(\frac{1}{2})) \leq \frac{16d_{eff}}{T^3}$. $\qquad\square$

### B.4 Proof of Lemma 3.5

**Lemma 3.5.** *For $\psi_\alpha$ a primitive of $x \mapsto x^{-\alpha}$ where $\alpha \in ]0,1]$, regularization $(\lambda_a)_{a\in[d]} \in (\mathbb{R}_{>0})^d$ and censorship vector $(p_a)_{a\in[d]}$, the solution of the optimization problem:*

$$\max_{\tau_1...,\tau_d\geq 0} \sum_{a\in[d]} \frac{1}{p_a}\left(\psi_\alpha(p_a\tau_a + \lambda_a) - \psi_\alpha(\lambda_a)\right) \quad s.t. \quad \sum_{a\in[d]}\tau_a = T$$

*is given by $\tau_a^\star = \frac{1}{p_a}[C - \lambda_a]^+$, where $C$ ensures the total budget constraint $\sum_{a\in[d]}\tau_a^\star = T$. In particular, with $\lambda_{eff} \triangleq \frac{1}{d_{eff}}\sum_{a\in[d]}\frac{\lambda_a}{p_a}$ and $\lambda_a^0 \triangleq d_{eff}(\lambda_a - \lambda_{eff})$, the optimal solution is given by $\tau_a^\star \triangleq \frac{1}{p_a d_{eff}}(T - \lambda_a^0)$ for $T \geq \max_a \lambda_a^0$ and the optimal value is $d_{eff}\psi_\alpha(\frac{T}{d_{eff}} + \lambda_{eff}) - \sum_{a\in[d]}\frac{1}{p_a}\psi_\alpha(\lambda_a)$.*

*Proof.* We first introduce the Lagrangian of the problem $\mathcal{L}(\tau_1, \ldots, \tau_d, \mu) := \sum_{a \in [d]} \frac{1}{p_a} \left( \psi_\alpha(p_a \tau_a + \lambda_a) - \psi_\alpha(\lambda_a) \right) + \mu(T - \sum_{a \in [d]} \tau_a)$. Differentiating with respect to $\tau_a$ for all $a \in [d]$ yields the equations:

$$\frac{1}{(p_a \tau_a + \lambda_a)^\alpha} - \mu = 0.$$

We then write it equivalently as:

$$\tau_a = \frac{1}{p_a} [\mu^{-1/\alpha} - \lambda_a].$$

However, since $(\tau_a)$ must be nonnegative, it may not always be possible to find a solution of this form. We then verify using KKT conditions that the solution:

$$\tau_a = \frac{1}{p_a} [C - \lambda_a]^+,$$

where $C$ ensures the total budget constraint $\sum_{a \in [d]} \tau_a^\star = T$, is optimal. In particular, whenever $T \geq \max_a \lambda_a^0$, we recover the solution provided in the second part the Lemma. $\square$

## B.5 Proof of Prop. 3.4

**Proposition 3.4.** *For all $\alpha > 0$, $\delta \in ]0, 1]$ and given $\psi_\alpha$ a primitive of $x \mapsto x^{-\alpha}$, we have:*

$$\max_{\pi \in \Pi} \mathbb{E}[\mathbb{V}_\alpha(T, \pi)] \leq \frac{d_{\textit{eff}}}{(1 - \delta)^\alpha} \left[ \psi_\alpha\left(\frac{T}{d_{\textit{eff}}} + \frac{\lambda}{1 - \delta}\right) - \psi_\alpha\left(\frac{\lambda}{1 - \delta}\right) \right] + \frac{24 d_{\textit{eff}} \log(T) + d}{\lambda^\alpha} + \frac{4 d_{\textit{eff}}}{\lambda^\alpha \delta^2 T^{12\delta^2}}.$$

*Proof.* For a given $\alpha \in ]0, 1]$, we condition on the event $\mathcal{H}_{\text{CEN}}^I(\delta)$ introduced in Lemma B.2 and consider the cases $\tau_a(t) \geq T_0(a)$ and $\tau_a(t) < T_0(a)$. This yields for any policy $\pi \in \Pi$:

$$\mathbb{V}_\alpha(T, \pi | \mathcal{H}_{\text{CEN}}^I(\delta)) \leq \frac{\sum_{a \in [d]} T_0(a)}{\lambda^\alpha} + \sum_{t=1}^T ((1 - \delta) p_{a_t} \tau_{a_t}(t - 1) + \lambda)^{-\alpha}$$

$$\leq \frac{24 d_{\textit{eff}} \log(T) + d}{\lambda^\alpha} + \frac{1}{(1 - \delta)^\alpha} \sum_{t=1}^T \left( p_{a_t} \tau_{a_t}(t - 1) + \frac{\lambda}{1 - \delta} \right)^{-\alpha}$$

$$\leq \frac{24 d_{\textit{eff}} \log(T) + d}{\lambda^\alpha} + \frac{1}{(1 - \delta)^\alpha} \sum_{a \in [d]} \int_0^{\tau_a(T)} \left( p_a u + \frac{\lambda}{1 - \delta} \right)^{-\alpha} \partial u$$

$$= \frac{24 d_{\textit{eff}} \log(T) + d}{\lambda^\alpha} + \frac{1}{(1 - \delta)^\alpha} \sum_{a \in [d]} \frac{1}{p_a} [\psi_\alpha(p_a \tau_a(T) + \frac{\lambda}{1 - \delta}) - \psi_\alpha(\frac{\lambda}{1 - \delta})].$$

We then apply the Lemma 3.5 with constant $\tilde{\lambda} \triangleq \lambda/(1 - \delta)$ to deduce:

$$\max_{\pi \in \Pi} \mathbb{V}_\alpha(T, \pi | \neg \mathcal{H}_{\text{CEN}}^I(\delta)) \leq \frac{24 d_{\textit{eff}} \log(T) + d}{\lambda^\alpha} + \frac{d_{\textit{eff}}}{(1 - \delta)^\alpha} \left[ \psi_\alpha\left(\frac{T}{d_{\textit{eff}}} + \frac{\lambda}{1 - \delta}\right) - \psi_\alpha\left(\frac{\lambda}{1 - \delta}\right) \right].$$

Then, we conclude thanks to Lemma B.2 that:

$$\max_{\pi \in \Pi} \mathbb{E}[\mathbb{V}_\alpha(T, \pi)] \leq \mathbb{P}(\neg \mathcal{H}_{\text{CEN}}^I(\delta)) \max_{\pi \in \Pi} \mathbb{V}_\alpha(T, \pi | \neg \mathcal{H}_{\text{CEN}}^I(\delta)) + (1 - \mathbb{P}(\neg \mathcal{H}_{\text{CEN}}^I(\delta))) \frac{1}{\lambda^\alpha}$$

$$\leq \frac{1}{(1 - \delta)^\alpha} d_{\textit{eff}} \left[ \psi_\alpha\left(\frac{T}{d_{\textit{eff}}} + \frac{\lambda}{1 - \delta}\right) - \psi_\alpha\left(\frac{\lambda}{1 - \delta}\right) \right] + \frac{24 d_{\textit{eff}} \log(T) + d}{\lambda^\alpha}$$

$$+ \frac{4}{\delta^2} d_{\textit{eff}} \frac{1}{\lambda^\alpha T^{12\delta^2}}.$$

In particular, for $\alpha = 1$ and $\delta = \frac{1}{2}$, this involves:

$$\max_{\pi \in \Pi} \mathbb{E}[\mathbb{V}_1(T, \pi)] \leq 2 d_{\textit{eff}} \log\left(\frac{T}{2\lambda} + 1\right) + \frac{24 d_{\textit{eff}} \log(T) + d}{\lambda} + 16 d_{\textit{eff}} \frac{1}{\lambda T^2},$$

and for $\alpha = \frac{1}{2}$ and $\delta = \frac{1}{2}$, this yields:

$$\max_{\pi \in \Pi} \mathbb{E}[\mathbb{V}_{\frac{1}{2}}(T, \pi)] \le \sqrt{2} d_{eff} \left[ \sqrt{\frac{T}{d_{eff}} + 2\lambda} - \sqrt{2\lambda} \right] + \frac{24 d_{eff} \log(T) + d}{\sqrt{\lambda}} + 16 d_{eff} \frac{1}{\sqrt{\lambda T^2}}.$$

$\square$

## B.6 Proof of Thm. 3.1

**Theorem 3.1.** *Under censorship, the UCB algorithm with regularization $\lambda$ has an instance-independent expected regret of:*

$$\mathbb{E}[R(T, \pi_{UCB})] \le \tilde{\mathcal{O}}(\sigma \sqrt{d_{eff} T}).$$

*Proof.* We first apply Lemma 3.3 to deduce:

$$\mathbb{E}[R(T, \pi_{\text{UCB}})] \le 2\sqrt{6\sigma^2 \log(T)} \mathbb{E}[\mathbb{V}_{\frac{1}{2}}(T, \pi_{\text{UCB}})] + 2\lambda \|\theta^\star\|_\infty \mathbb{E}[\mathbb{V}_1(T, \pi_{\text{UCB}})] + \frac{2d\Delta_{max}}{T}$$

$$\le 2\sqrt{6\sigma^2 \log(T)} \max_{\pi \in \Pi} \mathbb{E}[\mathbb{V}_{\frac{1}{2}}(T, \pi)] + 2\lambda \|\theta^\star\|_\infty \max_{\pi \in \Pi} \mathbb{E}[\mathbb{V}_1(T, \pi)] + \frac{2d\Delta_{max}}{T}.$$

We then apply proposition 3.4, with $\delta = 1/2$ in order to deduce:

$$\mathbb{E}[R(T, \pi_{\text{UCB}})] \le 2\sqrt{6\sigma^2 \log(T)} \Big( \sqrt{2} d_{eff} \Big[ \sqrt{\frac{T}{d_{eff}} + 2\lambda} - \sqrt{2\lambda} \Big] + \frac{24 d_{eff} \log(T) + d}{\sqrt{\lambda}} + 16 d_{eff} \frac{1}{\sqrt{\lambda T^2}} \Big)$$

$$+ 2\lambda \|\theta^\star\|_\infty \Big( 2 d_{eff} \log \Big( \frac{T}{2\lambda} + 1 \Big) + \frac{24 d_{eff} \log(T) + d}{\lambda^\alpha} + 16 d_{eff} \frac{1}{\lambda T^2} \Big) + \frac{2d\Delta_{max}}{T}.$$

By taking $\lambda = o(\log(T))$ and considering only the leading order, we conclude that:

$$\mathbb{E}[R(T, \pi_{\text{UCB}})] \le \tilde{\mathcal{O}}(\sigma \sqrt{d_{eff} T}).$$

Note that our proof easily allows to get high-probability bounds on regret instead of bounds on its expected value. $\square$

**Remark 3.** *We now extend Thm. 3.1 to heteroskedastic MAB. In this model, the pseudo-variance of the sub-Gaussian noisy reward is arm-dependent and denoted $\sigma_a$. Moreover, the value of $\sigma_a$ is known to the designer of the algorithm, that is, it can be used as a parameter for the UCB algorithm. We first apply a slightly modified version of Lemma 3.3 to deduce:*

$$\mathbb{E}[R(T, \pi_{UCB})] \le 2\sqrt{6\log(T)} \mathbb{E}[\bar{\mathbb{V}}_{\frac{1}{2}}(T, \pi_{UCB})] + 2\lambda \|\theta^\star\|_\infty \mathbb{E}[\mathbb{V}_1(T, \pi_{UCB})] + \frac{2d\Delta_{max}}{T},$$

*where for $\alpha > 0$ and $\pi \in \Pi$, we introduced the variance-based cumulative potential:*

$$\bar{\mathbb{V}}_\alpha(T, \pi) = \sum_{t=1}^{T} \Big( \frac{N_{a_t}(t-1)}{\sigma_{a_t}^{1/\alpha}} + \frac{\lambda}{\sigma_{a_t}^{1/\alpha}} \Big)^{-\alpha}.$$

*Thus, heteroskedasticity induces the mapping $\breve{p}_a \equiv p_a / \sigma_a^{1/\alpha}$ and $\breve{\lambda}_a \equiv \lambda / \sigma_a^{1/\alpha}$. Following the proof of Prop. 3.4, we deduce for any $\alpha > 0$ and time allocation $(\tau_a(T))_{a \in [d]}$:*

$$\mathbb{V}_\alpha(T, \pi | \mathcal{H}_{CEN}^I(\delta)) \le \frac{24 d_{eff} \log(T) + d}{\lambda^\alpha} + \frac{1}{(1-\delta)^\alpha} \sum_{a \in [d]} \frac{1}{\breve{p}_a} [\psi_\alpha(\breve{p}_a \tau_a(T) + \frac{\breve{\lambda}_a}{1-\delta}) - \psi_\alpha(\frac{\breve{\lambda}_a}{1-\delta})].$$

*In order to apply Lemma 3.5, we introduce the notation:*

$$\breve{d}_{eff} = \sum_{a \in [d]} \frac{\sigma_a^{1/\alpha}}{p_a}, \quad \breve{\lambda}_{eff} = \frac{\lambda}{1-\delta} \frac{d_{eff}}{\breve{d}_{eff}} \quad and \quad \breve{\lambda}_a^0 = \frac{\lambda \breve{d}_{eff}}{1-\delta} \Big( \frac{1}{\sigma_a^{1/\alpha}} - \frac{d_{eff}}{\breve{d}_{eff}} \Big).$$

and we deduce that whenever $T \geq \max_a \breve{\lambda}_a^0$, we have:

$$\mathbb{V}_\alpha(T, \pi | \mathcal{H}_{CEN}^I(\delta)) \leq \frac{24 d_{eff} \log(T) + d}{\lambda^\alpha} + \frac{1}{(1-\delta)^\alpha}\Big[\breve{d}_{eff}\psi_\alpha(\frac{T}{\breve{d}_{eff}} + \breve{\lambda}_{eff}) - \sum_{a \in [d]} \frac{\sigma_a^{1/\alpha}}{p_a}\psi_\alpha(\frac{\lambda}{(1-\delta)\sigma_a^{1/\alpha}})\Big].$$

In particular, by considering the case $\alpha = 1/2$ and only the leading order, we deduce that:

$$\mathbb{E}[R(T, \pi_{UCB})] \leq \tilde{\mathcal{O}}\Big(\sqrt{\breve{d}_{eff} T}\Big),$$

where as affirmed $\breve{d}_{eff} = \sum_{a \in [d]} \frac{\sigma_a^2}{p_a}$.

## B.7 Proof of Prop. 3.2

**Proposition 3.2.** *For a fixed action set $\mathcal{A}_t \equiv [d]$ and for a-priori known action gap $\Delta_a \triangleq \max_{\tilde{a}} \theta_{\tilde{a}}^\star - \theta_a^\star$, the UCB algorithm with regularization $\lambda$ has the instance-dependent expected regret:*

$$\mathbb{E}[R(T, \pi_{UCB})] \leq \mathcal{O}\Big(\log(T) \sum_{a \neq a^\star} \frac{1}{p_a} \max(\frac{\sigma^2}{\Delta_a}, \Delta_a)\Big).$$

*Proof.* As in the proof of Lemma 3.4, for a given round $t \in [T]$, we have under the event $\neg\mathcal{H}_{UCB}^\lambda$

$$\Delta_a = \max_{\tilde{a} \in \mathcal{A}} \theta_{\tilde{a}}^\star - \theta_a^\star \leq 2\sqrt{6\sigma^2 \frac{\log(T)}{N_{a_t}(t-1) + \lambda}} + 2\frac{\lambda\|\theta^\star\|_\infty}{\lambda + N_{a_t}(t-1)}.$$

It is as an inequality of the second degree and thus for any $t \in [T], a \in [d]$:

$$x_1 \left(\sqrt{\frac{1}{\lambda + N_a(t)}}\right)^2 + x_2\sqrt{\frac{1}{\lambda + N_a(t)}} - \Delta_a \geq 0,$$

where $x_1 = 2\lambda\|\theta^\star\|_\infty$ and $x_2 = 2\sqrt{6\sigma^2\log(T)}$. Solving it yields:

$$\sqrt{\frac{1}{\lambda + N_a(t)}} \geq \frac{1}{2x_1}(-x_2 + \sqrt{x_2^2 + 4\Delta_a x_1}),$$

or equivalently:

$$N_a(T) \leq \left(\frac{4\lambda\|\theta^\star\|_\infty}{\sqrt{24\sigma^2\log(T) + 8\Delta_a\lambda\|\theta^\star\|_\infty} - \sqrt{24\sigma^2\log(T)}}\right)^2 - \lambda \triangleq \Theta(T),$$

where we used the notation $\Theta(T)$ to simplify the presentation. Therefore, under $\neg\mathcal{H}_{CEN}^I(\frac{1}{2})$, we have:

$$\tau_a(t) \leq \max(T_0(a), \frac{2}{p_a}\Theta(T)).$$

This yields a conditional regret of:

$$R(T|\neg(\mathcal{H}_{CEN}^I(\frac{1}{2}) \cup \mathcal{H}_{UCB}^\lambda)) \leq \sum_{a \in [d], a \neq a^\star} \Delta_a \tau_a(T) = \sum_{a \in [d], a \neq a^\star} \frac{2\Delta_a}{p_a}\max(12\log(T) + \frac{p_a}{2}, \Theta(T)),$$

where $a^\star \triangleq \text{argmax}_{\tilde{a} \in \mathcal{A}} \theta_{\tilde{a}}^\star$ and an expected regret of:

$$\mathbb{E}[R(T, \pi_{UCB})] \leq \sum_{a \in [d], a \neq a^\star} \frac{2\Delta_a}{p_a}\max(12\log(T) + \frac{p_a}{2}, \Theta(T)) + \frac{d\Delta_{max}}{T} + \frac{16 d_{eff}\Delta_{max}}{T^2}.$$

In particular, for the regularization $\lambda = o(\log(T))$, we have the asymptotic:

$$\Theta(T) = \left(\frac{4\lambda\|\theta^\star\|_\infty}{\sqrt{24\sigma^2\log(T) + 8\Delta_a\lambda\|\theta^\star\|_\infty} - \sqrt{24\sigma^2\log(T)}}\right)^2 = \frac{24\sigma^2\log(T)}{\Delta_a^2} + \frac{8\lambda\|\theta^\star\|_\infty}{2\Delta_a} + o(1).$$

And thus, we conclude that:

$$\mathbb{E}[R(T, \pi_{UCB})] \leq \mathcal{O}\Big(\log(T) \sum_{a \in [d], a \neq a^\star} \frac{1}{p_a}\max(\frac{\sigma^2}{\Delta_a}, \Delta_a)\Big).$$

Again, note that our proof easily allows to get high-probability bounds on regret instead of bounds on its expected value. $\qquad\square$

**Remark 4.** *As in the instance-independent case, previous reasoning immediately extends to a-priori known heteroskedasticity and yields the upper bound:*

$$\mathbb{E}[R(T, \pi_{UCB})] \leq \mathcal{O}\Big( \log(T) \sum_{a \in [d], a \neq a^\star} \frac{1}{p_a} \max(\frac{\sigma_a^2}{\Delta_a}, \Delta_a) \Big).$$

Next, we provide additional insights to the main result of this section. In particular, we seek to gain intuition about how the policies that are adaptive to the realization of censorship process would perform in expectation against a class of non-adaptive (i.e. offline ) policies. In order to precisely derive asymptotic behavior of such policies, we introduce and study a continuous counterpart of the discrete original policy maximization problem $\max_{\pi \in \Pi} \mathbb{E}[\mathbb{V}_\alpha(T, \pi)]$.

Lemma 3.6 provides the basis for continuous approach in the case of offline policies by leveraging concentration inequalities for inverse Binomial distribution. We then extend this approach in the proof of Prop. 3.7. This extension enables us to provide an exact expression for the asymptotic gain of a policy class that monitors the censorship at a single point in time, as well as estimate the gain from fully adaptive policies.

### B.8 Proof of Lemma 3.6

**Lemma 3.6.** *For $\alpha \in ]0, 1]$ and $\lambda > 0$, we have $\max_{\pi \in \Pi_{\text{off}}} \mathbb{E}[\mathbb{V}_\alpha(T, \pi)] \sim d_{\text{eff}} \psi_\alpha(\frac{T}{d_{\text{eff}}} + \lambda)$.*

*Proof.* Given the offline nature of the policy class, we have:

$$\max_{\pi \in \Pi_{\text{off}}} \mathbb{E}[\mathbb{V}_\alpha(T, \pi)] = \max_{(\tau_a)_{a \in [d]}} \sum_{a \in [d]} \mathbb{E}\Big[\sum_{n=1}^{\tau_a} \frac{1}{(X_{n-1}^a + \lambda)^\alpha}\Big)\Big] = \max_{(\tau_a)_{a \in [d]}} \sum_{a \in [d]} \sum_{n=1}^{\tau_a} \mathbb{E}\Big[\frac{1}{(X_{n-1}^a + \lambda)^\alpha}\Big)\Big]$$

where we have re-indexed $(N_a(t))$ by actions, where $(\tau_a)_{a \in [d]}$ is a time allocation such that $\sum_{a \in [d]} \tau_a = T$ and where for a given action $a$, $(X_n^a)_{n \leq \tau_a}$ are dependent random variables verifying $X_{n+1} = X_n^a + \mathcal{B}(p_a)$ and $X_n^a \sim \mathcal{B}(n, p_a)$.

To lower bound this quantity, we fix a time allocation $(\tau_a)_{a \in [d]}$ and use the fact that $x \mapsto x^{-\alpha}$ is convex with Jensen's inequality to deduce:

$$\sum_{n=1}^{\tau_a} \mathbb{E}\Big[\frac{1}{(X_{n-1}^a + \lambda)^\alpha}\Big)\Big] \geq \sum_{n=1}^{\tau_a} \frac{1}{(\mathbb{E}[X_{n-1}^a] + \lambda)^\alpha} = \sum_{n=1}^{\tau_a} \frac{1}{(p_a(n-1) + \lambda)^\alpha} = \frac{1}{\lambda^\alpha} + \sum_{n=1}^{\tau_a - 1} \frac{1}{(p_a n + \lambda)^\alpha}$$

We then leverage the fact that $(px + \lambda)^{-\alpha} \geq \int_{x-1}^x (pu + \lambda)^{-\alpha} \partial u$ to deduce:

$$\sum_{n=1}^{\tau_a} \mathbb{E}\Big[\frac{1}{(X_{n-1}^a + \lambda)^\alpha}\Big)\Big] \geq \frac{1}{\lambda^\alpha} + \sum_{a \in [d]} \int_0^{\tau_a - 1} \frac{1}{(p_a x + \lambda)^\alpha} \partial x = \frac{1}{\lambda^\alpha} + \frac{1}{p_a}\Big[\psi_\alpha(p_a(\tau_a - 1) + \lambda) - \psi_\alpha(\lambda)\Big]$$

and therefore, for any time allocation $(\tau_a)_{a \in [d]}$, we have:

$$\max_{\pi \in \Pi_{\text{off}}} \mathbb{E}[\mathbb{V}_\alpha(T, \pi)] \geq \frac{d}{\lambda^\alpha} + \sum_{a \in [d]} \frac{1}{p_a}\Big[\psi_\alpha(p_a(\tau_a - 1) + \lambda) - \psi_\alpha(\lambda)\Big]$$

Although the maximum over time allocation is given by Lemma 3.5, we simply use the allocation $(\frac{T}{p_a d_{\text{eff}}})_{a \in [d]}$ to deduce:

$$\max_{\pi \in \Pi_{\text{off}}} \mathbb{E}[\mathbb{V}_\alpha(T, \pi)] \geq \sum_{a \in [d]} \frac{1}{p_a} \psi_\alpha(\frac{T}{d_{\text{eff}}} + \lambda - \frac{1}{p_a}) + \frac{d}{\lambda^\alpha} - \sum_{a \in [d]} \frac{1}{p_a} \psi_\alpha(\lambda)$$

By making the distinction between $\alpha = 1$ and $\alpha < 1$ to obtain the explicit expression of $\psi_\alpha$, we then show that the LHS is equivalent to $d_{\text{eff}} \psi_\alpha(\frac{T}{d_{\text{eff}}} + \lambda)$. The proof of the upper bound is more involved. In proving it, let's first assume that:

**Claim B.3.** *For all $a \in [d]$, $n \geq 1$, there exists a constant $C_2^a$ such that:*

$$\mathbb{E}\Big[\frac{1}{(X_n^a + \lambda)^\alpha)}\Big] \leq (1 + \frac{C_2^a}{(np_a)^{1/4}})\frac{1}{(p_a n + \lambda)^\alpha}$$

Given this result, we deduce:

$$\sum_{n=1}^{\tau_a - 1} \mathbb{E}\Big[\frac{1}{(X_n^a + \lambda)^\alpha)}\Big] \leq \sum_{n=1}^{\tau_a - 1}(1 + \frac{C_2^a}{(np_a)^{1/4}})\frac{1}{(p_a n + \lambda)^\alpha}.$$

Therefore, we have:

$$\max_{(\tau_a)_{a \in [d]}} \sum_{a \in [d]} \sum_{n=1}^{\tau_a} \mathbb{E}\Big[\frac{1}{(X_{n-1}^a + \lambda)^\alpha)}\Big] \leq \frac{d}{\lambda^\alpha} + \max_{(\tau_a)_{a \in [d]}} \sum_{a \in [d]} \sum_{n=1}^{\tau_a} \mathbb{E}\Big[\frac{1}{(X_n^a + \lambda)^\alpha)}\Big]$$

$$\leq \sum_{a \in [d]} \frac{1}{\lambda^\alpha} + \max_{(\tau_a)_{a \in [d]}} \sum_{a \in [d]} \sum_{n=1}^{\tau_a} \frac{1}{(p_a n + \lambda)^\alpha}$$

$$+ (\max_{a \in [d]} C_2^a) \max_{(\tau_a)_{a \in [d]}} \sum_{a \in [d]} \sum_{n=1}^{\tau_a} \frac{1}{(p_a n)^{1/4}} \frac{1}{(p_a n + \lambda)^\alpha}.$$

We first consider the second maximization problem and note that:

$$\max_{(\tau_a)_{a \in [d]}} \sum_{a \in [d]} \sum_{n=1}^{\tau_a} \frac{1}{(p_a n)^{1/4}} \frac{1}{(p_a n + \lambda)^\alpha} \leq \lambda^{1/4} \max_{(\tau_a)_{a \in [d]}} \sum_{a \in [d]} \sum_{n=1}^{\tau_a} \frac{1}{(p_a n + \lambda)^{\alpha + 1/4}}$$

$$= \mathcal{O}(d_{eff} \psi_{\alpha + \frac{1}{4}}(\frac{T}{d_{eff}} + \lambda))$$

$$= o(d_{eff} \psi_\alpha(\frac{T}{d_{eff}} + \lambda))$$

where we used an integral comparison and Lemma 3.5 to deduce the $\mathcal{O}$ scaling and the fact that $\alpha \in ]0, 1]$ the deduce the $o$ scaling. Similarly, we know that the first maximization problem scales as $d_{eff} \psi_\alpha(\frac{T}{d_{eff}} + \lambda)$ through another integral comparison and use of Lemma 3.5. Given this, we conclude that the upper bound is equivalent to $d_{eff} \psi_\alpha(\frac{T}{d_{eff}} + \lambda)$. Thanks to those two results, we finally affirm that:

$$\max_{\pi \in \Pi_{off}} \mathbb{E}[\mathbb{V}_\alpha(T, \pi)] \sim d_{eff} \psi_\alpha(\frac{T}{d_{eff}} + \lambda).$$

The last step needed is to prove Claim. B.3. In doing so, we extend Lemma 5.3 of [5] to more general inverse power function (i.e. $\alpha \neq 1$) with regularization $\lambda > 0$. We first introduce a tuning parameter $u \geq 0$ and write:

$$(\mathbb{E}[X_n^a] + \lambda)^\alpha \mathbb{E}\Big[\frac{1}{(X_n^a + \lambda)^\alpha}\Big] = (np_a + \lambda)^\alpha \mathbb{E}\Big[\frac{1}{(X_n^a + \lambda)^\alpha}\mathbf{1}\{X_n^a \leq u\mathbb{E}[X_n^a]\}\Big]$$

$$+ \mathbb{E}\Big[\frac{1}{(X_n^a + \lambda)^\alpha}\mathbf{1}\{X_n^a > u\mathbb{E}[X_n^a]\}\Big]$$

$$\leq \frac{(np_a + \lambda)^\alpha}{\lambda^\alpha}\mathbb{P}(X_n^a \leq u \cdot np_a) + (\frac{np_a + \lambda}{u \cdot np_a + \lambda})^\alpha.$$

Using a Berstein inequality for Binomiale variable, we have for all $\theta > 0$ and $n \in \mathbb{N}$:

$$\mathbb{P}\left(X_n^a \leq \left(1 - \sqrt{2\theta} - \frac{\theta}{3}\right)np_a\right) \leq e^{-\theta np_a}.$$

Thus, for all $0 < \theta \leq \frac{3(\sqrt{5} - \sqrt{3})^2}{2}$, by setting $u \equiv (1 - \sqrt{2\theta} - \frac{\theta}{3}) \geq 0$, we obtain that:

$$(\mathbb{E}[X_n^a] + \lambda)^\alpha \mathbb{E}\Big[\frac{1}{(X_n^a + \lambda)^\alpha}\Big] \leq \frac{(np_a + \lambda)^\alpha}{\lambda^\alpha}e^{-\theta np_a} + \left(\frac{np_a + \lambda}{(1 - \sqrt{2\theta} - \frac{\theta}{3})np_a + \lambda}\right)^\alpha.$$

By taking $\theta = A\frac{\log(np_a+\lambda)}{np_a}$ for another tunable parameter $A$ and for $n$ large enough to ensure $A\frac{\log(np_a+\lambda)}{np_a} \leq \frac{3(\sqrt{5}-\sqrt{3})^2}{2}$, this yields:

$$(\mathbb{E}[X_n^a]+\lambda)^\alpha \mathbb{E}\Big[\frac{1}{(X_n^a+\lambda)^\alpha}\Big] \leq \frac{(np_a+\lambda)^\alpha}{\lambda^\alpha(np_a+\lambda)^A} + \Big(\frac{np_a+\lambda}{(1-\sqrt{2A\frac{\log(np_a+\lambda)}{np_a}} - A\frac{\log(np_a+\lambda)}{3np_a})np_a+\lambda}\Big)^\alpha$$

$$= \frac{(np_a+\lambda)^\alpha}{\lambda^\alpha(np_a+\lambda)^A} + \Big(\frac{np_a+\lambda}{np_a+\lambda - \sqrt{2Anp_a\log(np_a+\lambda)} - \frac{A\log(np_a+\lambda)}{3}}\Big)^\alpha$$

For $n$ sufficiently large to ensure $\frac{3\sqrt{2Anp_a\log(np_a)}+A\log(np_a)}{3(np_a+\lambda)} \leq 1/2$ and given that $\alpha \in ]0,1]$, we then have:

$$\Big(\frac{1}{1-\frac{2\sqrt{2Anp_a\log(np_a+\lambda)}+A\log(np_a+\lambda)}{3(np_a+\lambda)}}\Big)^\alpha \leq \Big(1+2\frac{3\sqrt{2Anp_a\log(np_a+\lambda)}+A\log(np_a+\lambda)}{3(np_a+\lambda)}\Big)^\alpha$$

$$\leq 1+2\alpha\frac{3\sqrt{2Anp_a\log(np_a+\lambda)}+A\log(np_a+\lambda)}{3(np_a+\lambda)}$$

To conclude, we take take $A \equiv (\alpha+1)$ and this ensure that for $n$ sufficiently large, we obtain:

$$(\mathbb{E}[X_n^a]+\lambda)^\alpha \mathbb{E}\Big[\frac{1}{(X_n^a+\lambda)^\alpha}\Big] \leq 1+2\alpha\frac{3\sqrt{2(\alpha+1)np_a\log(np_a+\lambda)}+(\alpha+1)\log(np_a+\lambda)}{3(np_a+\lambda)}$$

$$+ \frac{1}{\lambda^\alpha(np_a+\lambda)}.$$

The leading order of this quantity is $\sqrt{\frac{\log(n)}{n}} = o(n^{-1/4})$ and therefore, we conclude that there exists a constant $C_2^a$, depending on $\lambda, \alpha$ and $p_a$ such that for all $n \geq 1$:

$$(\mathbb{E}[X_n^a]+\lambda)^\alpha \mathbb{E}\Big[\frac{1}{(X_n^a+\lambda)^\alpha}\Big] \leq 1+\frac{C_2^a}{(np_a)^{1/4}},$$

where $C_2^a$ is artificially increased to remove the two lower bounds conditions on $n$. $\qquad\square$

## B.9 Proof of Prop. 3.7

**Proposition 3.7.** *For $\alpha \in ]0,1]$, by denoting $\gamma_\alpha(\mathbf{p}) \triangleq \frac{\alpha}{2d_{\text{eff}}^{1-\alpha}}\sum_{a\in[d]}\frac{1}{p_a}\Big(\sum_{\tilde{a}\neq a}\frac{1-p_{\tilde{a}}}{p_{\tilde{a}}}\Big)$, we have:*

$$\max_{\pi\in\Pi_{adapt}}\mathbb{E}[\mathbb{V}_\alpha(T,\pi)] - \max_{\pi\in\Pi_{off}}\mathbb{E}[\mathbb{V}_\alpha(T,\pi)] = \gamma_\alpha(\mathbf{p})\frac{1}{T^\alpha} + o(\frac{1}{T^\alpha}). \qquad (\star)$$

*Moreover, if for a given $\beta \in ]0,1[$, we introduce $\Pi_{single}(\beta T)$ the policy class whose censorship information set has a single updating at time $\lfloor\beta T\rfloor$, we have:*

$$\max_{\pi\in\Pi_{single}(\beta T)}\mathbb{E}[\mathbb{V}_\alpha(T,\pi)] - \max_{\pi\in\Pi_{off}}\mathbb{E}[\mathbb{V}_\alpha(T,\pi)] = \gamma_\alpha(\mathbf{p})\frac{\beta}{T^\alpha} + o(\frac{1}{T^\alpha}). \qquad (\star\star)$$

Thus, we find that the power of a single monitoring is sufficient to ensure almost the same gain as adaptivity i.e. constant monitoring. The linear dependency in $T_0$ (due to the linear increase of variance in Binomial models) is also surprising. In non-asymptotic regime, it is still true but for $\beta$ verifying $0 < \beta_- \leq \beta \leq \beta_+ < 1$ for given $(\beta_-,\beta_+)$. We also observe a more general concave property of the single monitoring gain seen as a function of $T_0$, with limits equals to 0 on the borders on the interval. We conjecture that this concavity is likely to turn in a submodular dependency for several monitoring shots.

*Proof.* **Single Monitoring:** We first prove a slightly extended version of ($\star\star$) by considering a monitoring at time $T_0$ and we recover the results of Prop. 3.7 by setting $T_0 \equiv \beta T$, for a given $\beta \in ]0,1[$.

For the first step of the proof, we consider the continuous approximation of $\max\limits_{\pi \in \Pi_{\text{single}}(T_0)} \mathbb{E}[\mathbb{V}_\alpha(T, \pi)]$ given by the optimization problem over continuous variables:

$$\max_{\tau_a(T_0), \tau_a(T)} \mathbb{E}\Big[ \sum_{a \in [d]} \frac{1}{p_a}[\psi_\alpha(N_a(T)) - \psi_\alpha(N_a(T_0))] + \sum_{a \in [d]} \frac{1}{p_a}[\psi_\alpha(N_a(T_0)) - \psi_\alpha(\lambda)]\Big]$$

$$\text{s.t.} \sum_{a \in [d]} \tau_a(T_0) = T_0, \tag{$\mathcal{SM}$}$$

$$\sum_{a \in [d]} \tau_a(T) = T,$$

$$\forall a \in [d], \quad \tau_a(T) \geq \tau_a(T_0).$$

In $\mathcal{SM}$, the single monitoring max player initially commits to an allocation of the $T_0$ first rounds through the policy $(\tau_a(T_0))_{a \in [d]}$, with resulting gain expressed as the second term of the maximization problem. The player then observes the realization $N_a(T_0) \sim \mathcal{B}(\tau_a(T_0), p_a)$ and allocates the rest of the $T - T_0$ budget through the allocation $(\tau_a(T))_{a \in [d]}$, with resulting gain expressed as the first term of the maximization problem. . Therefore, the single monitoring gain assesses the value of observing the deviation of $N_a(T_0)$ from its expectation $\tau_a(T_0)p_a$. In an analogous way, we then introduce the continuous approximation of $\max\limits_{\pi \in \Pi_{\text{off}}} \mathbb{E}[\mathbb{V}_\alpha(T, \pi)]$ given by:

$$\max_{\tau_a(T)} \quad \mathbb{E}\Big[ \sum_{a \in [d]} \frac{1}{p_a}[\psi_\alpha(N_a(T)) - \psi_\alpha(\lambda)]\Big]$$

$$\text{s.t.} \sum_{a \in [d]} \tau_a(T) = T. \tag{$\mathcal{OFF}$}$$

In $\mathcal{OFF}$, $(N_a(T_0))_{a \in [d]}$ is not observed and thus can not be leveraged by the offline player to adapt the second part of the allocation. On what follows, we use $\mathbf{E}[\mathbf{N}]$ and $\mathbf{V}[\mathbf{N}]$ to denote respectively the mean and variance of $\mathbf{N}$, the empirical discrete distribution over $(N_a(T_0))_{a \in [d]}$ with associated weights $(1/p_a d_{\text{eff}})_{a \in [d]}$. Given a realization of $(N_a(T_0))_{a \in [d]}$, we use Lemma 3.5 to deduce the optimal choice of $(\tau_a(T))_{a \in [d]}$ in $\mathcal{SM}$ and resulting expected conditional gain:

$$\sum_{a \in [d]} \frac{1}{p_a}\underbrace{\Big[\psi_\alpha(\frac{T - T_0}{d_{\text{eff}}} + \mathbf{E}[\mathbf{N}]) - \psi_\alpha(N_a(T_0))\Big]}_{\text{Gain between } T_0 \text{ and } T \text{ for arm } a} + \sum_{a \in [d]} \frac{1}{p_a}\underbrace{\Big[\psi_\alpha(N_a(T_0)) - \psi_\alpha(\lambda)\Big]}_{\text{Gain between } 0 \text{ and } T_0 \text{ for arm } a},$$

where the formula is valid under the assumption $\forall a \in [d], T - T_0 \geq d_{\text{eff}}(N_a(T_0) - \mathbf{E}[\mathbf{N}])$. Such assumption encompass the fact that the remaining budget $T - T_0$ should be sufficient to correct the deviation observed. Logically, we know that on expectation $\mathbb{E}[N_a(T_0) - \mathbf{E}[\mathbf{N}]] = 0$, that is no systematic deviation is expected. For for all realization of randomness, the following deterministic crude upper bound hold:

$$d_{\text{eff}}(N_a(T_0) - \mathbf{E}[\mathbf{N}]) \leq d_{\text{eff}}(1 - \frac{1}{d_{\text{eff}}p_a})\frac{T_0}{p_a d_{\text{eff}}}$$

this in turn imposes:

$$\frac{T_0}{T} \leq \min_{a \in [d]} \frac{1}{1 + \frac{d_{\text{eff}} - \frac{1}{p_a}}{p_a d_{\text{eff}}}}.$$

For instance, in the uniform censorship model, this yields condition $\frac{T_0}{T} \leq \frac{dp}{dp + d - 1}$. Nevertheless, this is overly conservative and we can get considerably stronger results by considering high-probability concentration results on $N_a(T_0)$. Indeed, thanks to Chernoff Bounds for Binomial distribution, we have for $\delta \equiv T_0^{-1/4}$, that with probability at least $1 - 2d\exp\{-\delta^2 T_0/3d_{\text{eff}}\}$, for all $a$, $(1 - \delta)T_0/d_{\text{eff}} \leq N_a(T_0) \leq (1 + \delta)T_0/d_{\text{eff}}$. In particular, this yields $(1 - \delta)T_0/d_{\text{eff}} \leq \mathbf{E}[\mathbf{N}] \leq (1 + \delta)T_0/d_{\text{eff}}$. Under this event, we have $d_{\text{eff}}(N_a(T_0) - \mathbf{E}[\mathbf{N}]) \leq 2\delta T_0$, which imposes $T \geq (1 + 2\delta)T_0 = T_0 + 2T_0^{3/4}$. In particular, for $T_0 \equiv \beta T$, where $\beta \in ]0, 1[$, such condition will always be verified for $T$ large enough.

On the other hand, still using Lemma 3.5, we write the conditional expected gain of the offline policy on this same realization of $(N_a(T))_{a \in [d]}$ for $\mathcal{OFF}$ as:

$$\sum_{a \in [d]} \underbrace{\frac{1}{p_a} \left[ \psi_\alpha \left( \frac{T - T_0}{d_{\mathit{eff}}} + N_a(T_0) \right) - \psi_\alpha(N_a(T_0)) \right]}_{\text{Gain between } T_0 \text{ and } T \text{ for arm } a} + \sum_{a \in [d]} \underbrace{\frac{1}{p_a} \left[ \psi_\alpha(N_a(T_0)) - \psi_\alpha(\lambda) \right]}_{\text{Gain between } 0 \text{ and } T_0 \text{ for arm } a}.$$

where $\psi_\alpha(N_a(T_0))$ is artificially introduced. The difference between the two comes from the possibility for the monitoring policy to homogenize the realized $N_a(T_0)$ into a uniform $\mathbf{E}[\mathbf{N}]$. The random difference $\mathcal{G}_{single}(T_0)$, seen as a function of the realization of $(N_a(T_0))$ is then equal to:

$$\mathcal{G}_{single}(T_0) \triangleq \sum_{a \in [d]} \frac{1}{p_a} [\psi_\alpha \left( \frac{T - T_0}{d_{\mathit{eff}}} + \mathbf{E}[\mathbf{N}] \right) - \psi_\alpha \left( \frac{T - T_0}{d_{\mathit{eff}}} + N_a(T_0) \right)]$$

$$= d_{\mathit{eff}} \left[ \bar\psi_\alpha(\mathbf{E}[\mathbf{N}]) - \mathbf{E}[\bar\psi_\alpha(\mathbf{N})] \right],$$

which is exactly the Jensen's gap of the concave function $\bar\psi_\alpha : x \mapsto \psi_a\left( \frac{T - T_0}{d_{\mathit{eff}}} + x \right)$. The main insight is that this gap is then asymptotically equivalent to:

$$\bar\psi_\alpha(\mathbf{E}[\mathbf{N}]) - \mathbf{E}[\bar\psi_\alpha(\mathbf{N})] \sim -\frac{\bar\psi_\alpha^{(2)}(\mathbf{E}[\mathbf{N}])}{2} \mathbf{V}[\mathbf{N}],$$

where the RHS is positive, given that $\bar\psi_\alpha^{(2)}(\mathbf{E}[\mathbf{N}])$ is negative. To show this, we use the original proof of Jensen's inequality and introduce the interval $I \triangleq [\min_a N_a(T_0), \max_a N_a(T_0)]$ to leverage the mean value theorem. This then yields:

$$\frac{\min_{y \in I} -\bar\psi_\alpha^{(2)}(y)}{-\bar\psi_\alpha^{(2)}(\mathbf{E}[\mathbf{N}])} \leq 2 \frac{\bar\psi_\alpha(\mathbf{E}[\mathbf{N}]) - \mathbf{E}[\bar\psi_\alpha(\mathbf{N})]}{-\bar\psi_\alpha^{(2)}(\mathbf{E}[\mathbf{N}]) \mathbf{V}[\mathbf{N}]} \leq \frac{\max_{y \in I} -\bar\psi_\alpha^{(2)}(y)}{-\bar\psi_\alpha^{(2)}(\mathbf{E}[\mathbf{N}])}$$

Whenever $T_0$ is a constant independent of $T$, for $T \to +\infty$ by explicitly writing the definition of the upper and lower bounds, we have almost surely:

$$\frac{\min_{y \in I} -\bar\psi_\alpha^{(2)}(y)}{-\bar\psi_\alpha^{(2)}(\mathbf{E}[\mathbf{N}])} \to 1 \quad \text{and} \quad \frac{\max_{y \in I} -\bar\psi_\alpha^{(2)}(y)}{-\bar\psi_\alpha^{(2)}(\mathbf{E}[\mathbf{N}])} \to 1$$

Difficulties arises when $T_0$ is a function of $T$, as in the statement of the result where $T_0 \equiv \beta T$. By considering the same concentration event as the one introduced above, we have:

$$\frac{\min_{y \in I} -\psi_\alpha^{(2)}(y)}{-\bar\psi_\alpha^{(2)}(\mathbf{E}[\mathbf{N}])} = \min_{y \in I} \left( \frac{T - T_0 + d_{\mathit{eff}} \mathbf{E}[\mathbf{N}]}{T - T_0 + d_{\mathit{eff}} y} \right)^{1+\alpha} \geq \left( \frac{T - T_0 + (1 - \delta) T_0}{T - T_0 + (1 + \delta) T_0} \right)^{1+\alpha}$$

$$\leq \left( \frac{T - \delta T_0}{T + \delta T_0} \right)^{1+\alpha} = \left( \frac{T - T_0^{3/4}}{T + T_0^{3/4}} \right)^{1+\alpha} \to 1.$$

and similarly:

$$\frac{\max_{y \in I} -\psi_\alpha^{(2)}(y)}{-\bar\psi_\alpha^{(2)}(\mathbf{E}[\mathbf{N}])} = \max_{y \in I} \left( \frac{T - T_0 + d_{\mathit{eff}} \mathbf{E}[\mathbf{N}]}{T - T_0 + d_{\mathit{eff}} y} \right)^{1+\alpha} \leq \left( \frac{T - T_0 + (1 + \delta) T_0}{T - T_0 + (1 - \delta) T_0} \right)^{1+\alpha}$$

$$\leq \left( \frac{T + \delta T_0}{T - \delta T_0} \right)^{1+\alpha} = \left( \frac{T + T_0^{3/4}}{T - T_0^{3/4}} \right)^{1+\alpha} \to 1.$$

Thus, thanks to the exponential concentration, we conclude that:

$$\mathbb{E}[\mathcal{G}_{single}(T_0)] = \frac{d_{\mathit{eff}}}{2} \mathbb{E}[\bar\psi_\alpha(\mathbf{E}[\mathbf{N}]) - \mathbf{E}[\bar\psi_\alpha(\mathbf{N})]] \sim -\frac{d_{\mathit{eff}}}{2} \mathbb{E}[\bar\psi_\alpha^{(2)}(\mathbf{E}[\mathbf{N}]) \mathbf{V}[\mathbf{N}]].$$

Next, we affirm that:

$$\mathbb{E}[\bar\psi_\alpha^{(2)}(\mathbf{E}[\mathbf{N}]) \mathbf{V}[\mathbf{N}]] \overset{a)}{\sim} \mathbb{E}\left[ \bar\psi_\alpha^{(2)}(\mathbf{E}[\mathbf{N}]) \right] \mathbb{E}\left[ \mathbf{V}[\mathbf{N}] \right] \overset{b)}{\sim} \bar\psi_\alpha^{(2)}(\mathbb{E}[\mathbf{E}[\mathbf{N}]]) \mathbb{E}\left[ \mathbf{V}[\mathbf{N}] \right],$$

where a) leverages the previous bounds and where we use for b) similar concentration results on inverse of Binomial as done for the proof of Lemma 3.6. We then use the fact that $\mathbb{E}[\mathbf{E}[\mathbf{N}]] = T_0/d_{eff}$ to conclude:

$$\mathbb{E}[\mathcal{G}_{single}(T_0)] \sim -\frac{d_{eff}}{2}\psi_\alpha^{(2)}(\frac{T}{d_{eff}})\mathbb{E}\Big[\mathbf{V}[\mathbf{N}]\Big]. \tag{$\mathcal{V}$}$$

We consider this result to be one of the main insight for adaptivity, as it involves that at first order, the gain grows linearly in the expected value of the empirical variance of the arm allocation process. In opposition to the single monitoring policy, the adaptive policy continuously exploits such variance. Yet, in doing so, it creates a second order induced variance but we then show that this phenomena is negligible at first order. To reach a result with explicit dependency on the censorship probability $(p_a)_{a\in[d]}$, we note that:

$$\mathbb{E}[\mathbf{V}[\mathbf{N}]] = \frac{T_0}{d_{eff}^3}\sum_{a\in[d]}\frac{1}{p_a}\Big[\sum_{b\neq a}\frac{1-p_b}{p_b}\Big],$$

and therefore:

$$\mathbb{E}\Big[\mathcal{G}_{single}(T_0)\Big] \sim \frac{\alpha}{2d_{eff}^2}(\frac{d_{eff}}{T})^{1+\alpha}T_0\sum_{a\in[d]}\frac{1}{p_a}\Big[\sum_{b\neq a}\frac{1-p_b}{p_b}\Big] = \gamma_\alpha(\mathbf{p})\frac{T_0}{T^{1+\alpha}}.$$

In particular, for $T_0 = \beta T$, this yields $\mathbb{E}\Big[\mathcal{G}_{single}(\beta T)\Big] = \gamma_\alpha(\mathbf{p})\frac{\beta}{T^\alpha} + o(\frac{1}{T^\alpha})$.

The second step closely mirrors the proof of Lemma 3.6 and consists in justifying the use of the continuous approximation for the two optimization problems ($\mathcal{OFF}$) and ($\mathcal{SM}$). As in Lemma 3.6, we show that the difference between the continuous and discrete optimization results at most in a second order gain of $o(\frac{1}{T^\alpha})$, even when maximized as a decoupled quantity. By combining those two results, we finally deduce as announced:

$$\max_{\pi\in\Pi_{single}(\beta T)}\mathbb{E}[\mathbb{V}_\alpha(T,\pi)] - \max_{\pi\in\Pi_{off}}\mathbb{E}[\mathbb{V}_\alpha(T,\pi)] \sim \mathbb{E}\Big[\mathcal{G}_{single}(\beta T)\Big] = \gamma_\alpha(\mathbf{p})\frac{\beta}{T^\alpha} + o(\frac{1}{T^\alpha})$$

**Complete Adaptivity**

We next tackle the proof of ($\star$), where the main idea is to show that a formula analogous to ($\mathcal{V}$) holds for the variance of a suited random process. First, using the same proof technique as in Sec. 4 of [19] thanks to the decaying property of the reward in function of the number of realization, we show that the optimal adaptive policy is the greedy policy, that is the policy that picks at time $t$ the action:

$$a_t \triangleq \text{argmax}_{a\in\mathcal{A}_t}(N_a(t-1) + \lambda)^{-\alpha},$$

with arbitrary but consistent tie-breaking. In particular, this ensures that for all actions $a, b$ and time $t$, we have $|N_a(t) - N_b(t)| \leq 1$. We then introduce the offline and adaptive allocations:

$$\tau_a^{off}(T) \triangleq \frac{T}{p_a d_{eff}} \quad \text{and} \quad \tau_a^{on}(T) \triangleq \sum_{i=1}^{N_a(T)}\frac{1}{p_a} + \xi_i^a = \frac{N_a(T)}{p_a} + S^a(N_a(T))$$

where $\frac{1}{p_a} + \xi_i^a$ is the total random number of allocation it takes for action $a$ to be realized in the $i^{th}$ selection, $\xi_i^a$ being equal the centered deviation with respect to the expected value $\frac{1}{p_a}$. Of key importance in our proof is $S^a(N_a(T))$, the cumulative deviation defined as $\sum_{i=1}^{N_a(T)}\xi_i^a$. Note that it is well approximated in large $T$ regime as a random sum of $N_a(T)$ i.i.d. geometric centered variable of parameter $p_a$. Given this and the total budget constraint, we have the simple relation $\tau_a^{on}(T) = \tau_a^{off}(T) + \frac{1}{d_{eff}}\sum_b[\frac{S^a(N_a(T))}{p_b} - \frac{S^b(N_b(T))}{p_a}]$. A relevant quantity to introduce is the random allocation difference $\Delta\tau_{a,b}$ between actions $a$ and $b$ defined by:

$$\Delta\tau_{a,b} \triangleq \frac{1}{d_{eff}}(\frac{S^a(N_a(T))}{p_b} - \frac{S^b(N_b(T))}{p_b})$$

Using this notation, we simply have $\tau_a^{on} = \tau_a^{off} + \sum_b\Delta\tau_{a,b}$. We then introduce the random sets $I^+ \triangleq \{a : \tau_a^{on} \geq \tau_a^{off}\} = \{a : \sum_b\Delta\tau_{a,b} \geq 0\}$ and $I^- \triangleq \{a : \tau_a^{on} < \tau_a^{off}\} = \{a : \sum_b\Delta\tau_{a,b} <$

$0\}$. On the one hand, $I^+$ represents the set of actions that are more sampled by the adaptive policy than by the offline policy i.e. that leads to a gain thanks to the greedy property. One the other hand, $I^-$ is the set of actions under-selected by the adaptive policy, leading to a loss although inferior in absolute value to the resulting gain of $I^+$. As for the proof of the single monitoring case, we condition on the realization $(N_a(T))$ and use a continuous approximation given this conditioning to study the difference of gain. Thus, we have the action gain for $a \in I^+$,:

$$g_a \triangleq \frac{1}{p_a}[\psi_\alpha(N_a(T)) - \psi_\alpha(N_a(T) - p_a \sum_b \Delta\tau_{a,b})]$$

$$\approx \frac{1}{p_a}\Big[p_a\Big(\sum_b \Delta\tau_{a,b}\Big)\psi_\alpha^{(1)}(N_a(T)) - \frac{(p_a \sum_b \Delta\tau_{a,b})^2}{2}\psi_\alpha^{(2)}(N_a(T))\Big].$$

On the other hand, for $a \in I^-$, we have the action loss still under the continuous approximation:

$$l_a \triangleq \frac{1}{p_a}[\psi_\alpha(N_a(T) + p_a \sum_b \Delta\tau_{a,b}) - \psi_\alpha(N_a(T))]$$

$$\approx \frac{1}{p_a}\Big[p_a\Big(\sum_b \Delta\tau_{a,b}\Big)\psi_\alpha^{(1)}(N_a(T)) + \frac{(p_a \sum_b \Delta\tau_{a,b})^2}{2}\psi_\alpha^{(2)}(N_a(T))\Big].$$

By introducing $\mathcal{G}_{\text{adapt}} \triangleq \sum_{a \in I^+} g_a - \sum_{a \in I^-} l_a$, the adaptive equivalent of $\mathcal{G}_{\text{single}}$ and combining previous two results, we deduce:

$$\mathcal{G}_{\text{adapt}} = \sum_{a \in I^+} \frac{1}{p_a}\Big[p_a\Big(\sum_b \Delta\tau_{a,b}\Big)\psi_\alpha^{(1)}(N_a(T)) - \frac{(p_a \sum_b \Delta\tau_{a,b})^2}{2}\psi_\alpha^{(2)}(N_a(T))\Big]$$

$$- \sum_{a \in I^-} \frac{1}{p_a}\Big[p_a\Big(\sum_b \Delta\tau_{a,b}\Big)\psi_\alpha^{(1)}(N_a(T)) + \frac{(p_a \sum_b \Delta\tau_{a,b})^2}{2}\psi_\alpha^{(2)}(N_a(T))\Big]$$

$$= \sum_{a \in I^+} \Big(\sum_b \Delta\tau_{a,b}\Big)\psi_\alpha^{(1)}(N_a(T)) - \sum_{a \in I^-} \Big(\sum_b \Delta\tau_{a,b}\Big)\psi_\alpha^{(1)}(N_a(T))$$

$$- \frac{1}{2} \sum_{a \in [d]} (p_a \sum_b \Delta\tau_{a,b})^2 \psi_\alpha^{(2)}(N_a(T))$$

We then leverage the Taylor expansion $\psi_\alpha^{(1)}(N_a(T)) = \psi_\alpha^{(1)}(\bar{N}(T)) + \psi_\alpha^{(2)}(\bar{N}(T))(\bar{N}(T) - N_a(T))$, where $\bar{N}(T) \triangleq \sum_{a \in [d]} N_a(t)/d$. We know that the second term is asymptotically negligible given that $\alpha > 0$ and that the difference between $\bar{N}(T)$ and $N_a(T)$ is constant with exponential probability, thanks to the greedy policy property. We combine this result with the fact that by definition $\sum_{a \in I^+} \sum_b \Delta\tau_{a,b} - \sum_{a \in I^-} \sum_b \Delta\tau_{a,b} = 0$ to deduce that at first order:

$$\mathcal{G}_{\text{adapt}} = -\frac{1}{2}\psi_\alpha^{(2)}(\bar{N}(T)) \sum_{a \in [d]} (p_a \sum_b \Delta\tau_{a,b})^2 \qquad (\mathcal{L})$$

We see formula ($\mathcal{L}$) as the adaptive analogous of ($\mathcal{V}$). Indeed, it involves the product of the second derivative $\frac{1}{2}\psi_\alpha^{(2)}(\bar{N}(T))$, evaluated on a quantity concentrating at $T/d_{\text{eff}}$ with a variance term associated to the adaptive action allocation process. We remark that for any $a \in [d]$:

$$\mathbb{E}\Big[(\sum_b \Delta\tau_{a,b})^2\Big] = \frac{1}{d_{\text{eff}}^2}\mathbb{E}\Big[([\sum_{b \neq a} \frac{1}{p_b}]S^a(N_a(T)) - \frac{1}{p_a}\sum_{b \neq a} S^b(N_b(T)))^2\Big]$$

$$= \frac{1}{d_{\text{eff}}^2}\Big[(d_{\text{eff}} - \frac{1}{p_a})^2\mathbb{V}[S^a(N_a(T))] + \sum_{b \neq a} \frac{1}{p_a^2}\mathbb{V}[S^b(N_b(T))]\Big]$$

and therefore, by summing:

$$\sum_{a\in[d]} p_a \mathbb{E}\Big[\big(\sum_b \Delta\tau_{a,b}\big)^2\Big] = \frac{1}{d_{\textit{eff}}^2} \sum_{a\in[d]} p_a\Big[\big(\sum_{b\neq a}\frac{1}{p_b}\big)^2 \mathbb{V}[S^a(N_a(T))] + \sum_{b\neq a}\frac{1}{p_a^2}\mathbb{V}[S^b(N_b(T))]\Big]$$

$$= \frac{1}{d_{\textit{eff}}^2}\sum_{a\in[d]}\Big[p_a(d_{\textit{eff}} - \frac{1}{p_a})^2 + d_{\textit{eff}} - \frac{1}{p_a}\Big]\mathbb{V}[S^a(N_a(T))]$$

$$= \sum_{a\in[d]} p_a \frac{d_{\textit{eff}} - \frac{1}{p_a}}{d_{\textit{eff}}}\mathbb{V}[S^a(N_a(T))].$$

To obtain the leading order of $\mathbb{V}[S^a(N_a(T))]$, we use Wald's second equation and the fact that $S^a(N_a(T))$ is approximated by a sum of geometric random variable of parameter $p_a$, modulo a asymptotically negligible summing constraint due to the fixed total budget $T$. This yields $\mathbb{V}[S^a(N_a(T))] \sim \frac{1-p_a}{p_a^2}\mathbb{E}[N_a(T)] \sim \frac{1-p_a}{p_a^2}\mathbb{E}[\bar{N}(T)]$, where the last results leverages again the fact that the difference between the two quantities is constant with exponential probability. We conclude with further algebraic calculation that:

$$\mathbb{E}[\mathcal{G}_{\text{adapt}}] \sim -\mathbb{E}\big[\frac{\psi_\alpha^{(2)}}{2}(\bar{N}(T))\big]\sum_{a\in[d]} p_a \frac{d_{\textit{eff}} - \frac{1}{p_a}}{d_{\textit{eff}}}\frac{1-p_a}{p_a^2}\mathbb{E}[\bar{N}(T)]$$

$$\sim \frac{\alpha}{2}\big(\frac{d_{\textit{eff}}}{T}\big)^{1+\alpha} \sum_{a\in[d]}\frac{1}{p_a}\Big[\sum_{b\neq a}\frac{1-p_b}{p_b}\Big]\frac{T}{d_{\textit{eff}}^2}$$

$$\sim \gamma_\alpha(\mathbf{p})\frac{1}{T^\alpha}.$$

By justifying again that the continuous gain approximation leads to terms of order $o(\frac{1}{T^\alpha})$, as done in the proof of Lemma 3.6, we conclude that:

$$\max_{\pi\in\Pi_{\text{adapt}}}\mathbb{E}[\mathbb{V}_\alpha(T,\pi)] - \max_{\pi\in\Pi_{\text{off}}}\mathbb{E}[\mathbb{V}_\alpha(T,\pi)] = \gamma_\alpha(\mathbf{p})\frac{1}{T^\alpha} + o(\frac{1}{T^\alpha}).$$

$\square$

# C  Proof of Sec. 4 - Contextual Bandits

In this section, we prove Thm. 4.1 of Sec.4, extending the results of MAB to LCB. To do so, we prove Lemmas 4.2, C.1 and Prop. 4.3. Note that the proof of Thm. 4.6 is differed to next section. We conclude the section by discussing the extension of our analysis to Generalized Linear Contextual Bandits.

## C.1  Proof of Lemma 4.2

**Lemma 4.2.** *For all $\delta \in ]0,1]$, there exists a constant $\tilde{\beta}_\delta(T) = \Theta(\sqrt{d\log(T)})$ such that*

$$\mathbb{E}[R(T,\pi_{UCB})] \le 2\tilde{\beta}_\delta(T)\sqrt{T\mathbb{E}[\mathbb{V}_1(T,\pi_{UCB})]} + \delta T\Delta_{max},$$

*where, for $\alpha > 0$ and $\pi \in \Pi$, the linear extension of the cumulative censored potential is given by:*

$$\mathbb{V}_\alpha(T,\pi) \triangleq \sum_{t=1}^T \|a_t\|_{(\mathbb{W}_{t-1}^C)^{-\alpha}}^2 = \sum_{t=1}^T \text{Tr}((\mathbb{W}_{t-1}^C)^{-\alpha} a_t a_t^\top).$$

*Proof.* We have under the event $\neg\mathcal{H}_{\text{UCB}}^{II}(\delta)$ introduced in Lemma C.1 and thanks to Holder inequality:

$$\Delta_t(a) \triangleq \max_{\tilde{a}\in\mathcal{A}_t}\langle\theta^\star,\tilde{a}\rangle - \langle\theta^\star, a_t\rangle \le 2\beta_\delta(t-1)\|a_t\|_{(\mathbb{W}^C(t-1))^{-1}}.$$

Therefore, the conditional regret is upper-bounded by:

$$R(T|\neg\mathcal{H}_{\text{UCB}}^{II}(\delta)) \le \beta_\delta(T)\sum_{t=1}^T\|a_t\|_{(\mathbb{W}^C(t-1))^{-1}} = \beta_\delta(T)\tilde{\mathbb{V}}_{\frac{1}{2}}(T,\pi),$$

where we introduced $\tilde{\mathbb{V}}_{\frac{1}{2}}(T, \pi) \triangleq \sum_{t=1}^{T} \|a_t\|_{\mathbb{W}^C(t-1)^{-1}}$. Cauchy Schwartz inequality then allows to make the junction $\tilde{\mathbb{V}}_{\frac{1}{2}}(T, \pi) \leq \sqrt{T}\sqrt{\mathbb{V}_1(T, \pi)}$. We then introduce $\tilde{\beta}_\delta(T)$ a deterministic upper bound on $\beta_\delta(T)$:

$$
\begin{aligned}
\beta_\delta(T) &= \sqrt{\sigma^2 \log\left(\frac{\det(\mathbb{W}_T^C)}{\det(\lambda\mathbb{I}_d)}\right) + 2\sigma^2 \log(\frac{1}{\delta})} + \sqrt{\lambda}\|\theta^\star\|_2 \\
&\leq \underbrace{\sqrt{\sigma^2 d \log(1 + \frac{T}{d\lambda}) + 2\sigma^2 \log(\frac{1}{\delta})} + \sqrt{\lambda}\|\theta^\star\|_2}_{\triangleq \tilde{\beta}_\delta(T)} \\
&= \Theta(\sqrt{d\log(T)}).
\end{aligned}
$$

Using the concavity of square root and Jensen's inequality, we have $\mathbb{E}[\sqrt{\mathbb{V}_1(T, \pi)}] \leq \sqrt{\mathbb{E}[\mathbb{V}_1(T, \pi)]}$. Finally, thanks to Lemma C.1, we conclude that:

$$
\mathbb{E}[R(T, \pi_{\text{UCB}})] \leq 2\tilde{\beta}_\delta(T)\sqrt{T\mathbb{E}[\mathbb{V}_1(T, \pi_{UCB})]} + \delta T \Delta_{max}.
$$

$\square$

## C.2 Statement and Proof of Lemma C.1

Analogous to Lemma B.1 for the MAB case, one key step in the proof is introduction of the failure of optimism event. Nevertheless, note the difference with the choice of norm.

**Lemma C.1.** *For any $\delta \in ]0, 1]$, uniform regularization $\lambda > 0$ and censored action generating process $(\mathbb{W}_t^C)_{t \leq T}$, let's introduce the event:*

$$
\mathcal{H}_{UCB}^{II}(\delta) \triangleq \left\{ \exists t \geq 0, \|\hat{\theta}_t^\lambda - \theta^\star\|_{\mathbb{W}_t^C} > \underbrace{\sqrt{\sigma^2 \log\left(\frac{\det(\mathbb{W}_t^C)}{\det(\lambda\mathbb{I}_d)}\right) + 2\sigma^2 \log(\frac{1}{\delta})} + \sqrt{\lambda}\|\theta^\star\|_2}_{\triangleq \beta_\delta(t)} \right\}.
$$

*We then have $\mathbb{P}(\mathcal{H}_{UCB}^{II}(\delta)) \leq \delta$.*

*Proof.* The proof closely mirrors the self-normalized bound for vector-valued martingales of Thm.1 from [1]. The main subtlety is to apply the results to the censored measurable vectors $(x_{a_t} a_t)$ instead of classically $(a_t)$. This yields that with probability $1 - \delta$, for all $t \geq 0$:

$$
\|\sum_{n=1}^{t} \epsilon_n x_{a_n} a_n\|_{\mathbb{W}_t^C}^2 \leq \sigma^2 \log \frac{\det(\mathbb{W}_t^C)}{\det(\lambda\mathbb{I}_d)} + 2\log(\frac{1}{\delta}).
$$

Thus, still on this event, for any $t \geq 0$ and action $a \in \mathbb{R}^d$, we have by definition of $\hat{\theta}_t^\lambda$ (Sec.A.1):

$$
\langle a, \hat{\theta}_t^\lambda \rangle - \langle a, \theta^\star \rangle = \langle a, (\mathbb{W}_t^C)^{-1} \sum_{n=1}^{t} \epsilon_n x_{a_n} a_n \rangle - \lambda \langle a, (\mathbb{W}_t^C)^{-1}\theta^\star \rangle,
$$

and therefore, thanks to Cauchy-Schwartz inequality:

$$
|\langle a, \hat{\theta}_t^\lambda \rangle - \langle a, \theta^\star \rangle| \leq \|a\|_{(\mathbb{W}_t^C)^{-1}} \left( \|\sum_{n=1}^{t} \epsilon_t x_{a_t} a_t\|_{\mathbb{W}_t^C} + \lambda^{1/2}\|\theta^\star\|_2 \right)
$$

Using previous result, for all $a \in \mathbb{B}_d, t \geq 0$, with probability $1 - \delta$, we have:

$$
|\langle a, \hat{\theta}_t^\lambda \rangle - \langle a, \theta^\star \rangle| \leq \sigma\sqrt{\log\left(\frac{\det(\mathbb{W}_t^C)}{\det(\lambda\mathbb{I}_d)}\right) + 2\log(\frac{1}{\delta})} + \lambda^{1/2}\|\theta^\star\|_2
$$

To conclude, we classically plug-in the value $a = \mathbb{W}_t^C(\hat{\theta}_t^\lambda - \theta^\star)$ and divide both sides by $\|\hat{\theta}_t^\lambda - \theta^\star\|_{\mathbb{W}_t^C}$ to get that for all $t \geq 0$, with probability $1 - \delta$, we have:

$$
\|\hat{\theta}_t^\lambda - \theta^\star\|_{\mathbb{W}_t^C} \leq \sigma\sqrt{\log\left(\frac{\det(\mathbb{W}_t^C)}{\det(\lambda\mathbb{I}_d)}\right) + 2\log(\frac{1}{\delta})} + \lambda^{1/2}\|\theta^\star\|_2
$$

and therefore, by definition $\mathbb{P}(\mathcal{H}_{\text{UCB}}^{II}(\delta)) \leq \delta$.

$\square$

## C.3 Proof of Prop. 4.3

**Proposition 4.3.** *For any $\delta \in ]0, 1]$, $\lambda > 0$, $\alpha > 0$ and policy $\pi \in \Pi$, we have:*

$$\mathbb{E}[V_\alpha(T, \pi)] \leq \frac{\delta}{\lambda^\alpha} + C(\delta)^\alpha \operatorname{Tr}\Big( \int_0^T \mathbb{W}(t)^{-\alpha} a(t) a(t)^\top \partial t \Big),$$

*where $C(\delta) \triangleq 8(\lambda + 1) \max(\log(d/\delta))/\lambda, 1)/\lambda$.*

*Proof.* First, we use Lemma C.2 to deduce that under $\mathcal{H}_{\text{CEN}}^{II}(\delta)$:

$$\mathbb{V}_\alpha(T, \pi | \mathcal{H}_{\text{CEN}}^{II}(\delta)) = \sum_{t=1}^T \operatorname{Tr}((\mathbb{W}_{t-1}^C)^{-\alpha} a_t a_t^\top) \leq c_\delta^\alpha \sum_{t=1}^T \operatorname{Tr}(\mathbb{W}_{t-1}^{-\alpha} a_t a_t^\top).$$

For all $t \geq 1$, we then use the fact that $W_t \preceq (1 + \frac{1}{\lambda}) W_{t-1}$ to deduce $\operatorname{Tr}(\mathbb{W}_{t-1}^{-\alpha} a_t a_t^\top) \leq (1 + \frac{1}{\lambda})^\alpha \operatorname{Tr}(\mathbb{W}_t^{-\alpha} a_t a_t^\top)$. The last and most important step is the integral comparison:

$$\sum_{t=1}^T \operatorname{Tr}(\mathbb{W}_t^{-\alpha} a_t a_t^\top) \leq \int_0^T \operatorname{Tr}(\mathbb{W}(t)^{-\alpha} a(t) a(t)^\top) \partial t = \operatorname{Tr}\Big( \int_0^T \mathbb{W}(t)^{-\alpha} a(t) a(t)^\top \partial t \Big).$$

In the previous result, the continuous extension $(a(t), \mathbb{W}(t))_{t \leq T}$ of $(a_t, \mathbb{W}_t)_{t \in [T]}$ for a given policy $\pi$ is defined for any time $t \geq 1$ as:

$$a(t) \triangleq a_{\lfloor t \rfloor} \quad \text{and} \quad \mathbb{W}(t) \triangleq \int_{u=1}^t p_{a(u)} a(u) a(u)^\top \partial u = \mathbb{W}_{\lfloor t \rfloor} + (t - \lfloor t \rfloor) p(a_{\lceil t \rceil}) a_{\lceil t \rceil} a_{\lceil t \rceil}^\top.$$

This yields the result:

$$\mathbb{V}_\alpha(T, \pi | \mathcal{H}_{\text{CEN}}^{II}(\delta)) \leq c_\delta^\alpha (1 + \frac{1}{\lambda})^\alpha \operatorname{Tr}\Big( \int_0^T \mathbb{W}(t)^{-\alpha} a(t) a(t)^\top \partial t \Big).$$

Finally, we conclude thanks to Lemma C.2 that:

$$\mathbb{E}[\mathbb{V}_\alpha(T, \pi)] \leq \frac{\delta}{\lambda^\alpha} + C(\delta)^\alpha \operatorname{Tr}\Big( \int_0^T \mathbb{W}(t)^{-\alpha} a(t) a(t)^\top \partial t \Big).$$

$\square$

**Remark 5.** *The main tour de force of the continuous approximation we employ is to relax the maximization problem by considering the class of continuous deterministic integrable policies, which is considerably more tractable from an analysis perspective. On the one hand, it allows to get closed-form solution for the maximization problem whereas the discrete approach can only deal with approximations and upper bounds. On the other hand, it clearly reveals the underlying matrix function the discrete approach is approximating and henceforth allows to leverage powerful integration results. We leverage again this idea in the context of Sec.4 to tackle impact of censorship.*

*To illustrate the abovementioned points, we remark that for the simpler case of classical uncensored environment, we obtain for $\alpha > 0, \alpha \neq 1$:*

$$\sum_{t=1}^T \|a_t\|_{\mathbb{W}_{t-1}^{-\alpha}}^2 \leq \Big( \frac{\lambda + 1}{\lambda} \Big)^\alpha \frac{\operatorname{Tr}\Big( \int_0^T \partial \mathbb{W}(t)^{1-\alpha} \Big)}{1 - \alpha} = \Big( \frac{\lambda + 1}{\lambda} \Big)^\alpha \frac{\operatorname{Tr}(\mathbb{W}_T^{1-\alpha} - \mathbb{W}_0^{1-\alpha})}{1 - \alpha}.$$

*For $\alpha < 1$, we then have thanks to Lemma 3.5 the worst case bound $\operatorname{Tr}(\mathbb{W}_T^{1-\alpha}) \leq d^\alpha (d\lambda + T)^{1-\alpha}$ and henceforth:*

$$\sum_{t=1}^T \|a_t\|_{\mathbb{W}_{t-1}^{-\alpha}}^2 \leq \Big( \frac{\lambda + 1}{\lambda} \Big)^\alpha \frac{d^\alpha (d\lambda + T)^{1-\alpha} - d\lambda^{1-\alpha}}{1 - \alpha}$$

*On the other hand, for $\alpha > 1$, we deduce:*

$$\sum_{t=1}^T \|a_t\|_{\mathbb{W}_{t-1}^{-\alpha}}^2 \leq \Big( \frac{\lambda + 1}{\lambda} \Big)^\alpha \frac{d\lambda^{1-\alpha}}{\alpha - 1}.$$

*Finally, for $\alpha = 1$, we use the formula $\mathrm{Tr}(\log(A)) = \log(\det A)$ to deduce:*

$$\sum_{t=1}^{T} \|a_t\|_{\mathbb{W}_{t-1}^{-1}}^2 \leq \frac{\lambda+1}{\lambda} \int_0^T \frac{\partial \log \det(\mathbb{W}(t))}{\partial t} \partial t = \frac{\lambda+1}{\lambda} \mathrm{Tr}(\log \mathbb{W}_T - \log \mathbb{W}_0) = \frac{\lambda+1}{\lambda} \log \frac{\det \mathbb{W}_T}{\det \mathbb{W}_0}$$

$$\leq \frac{\lambda+1}{\lambda} \log(1 + \frac{T}{\lambda d}),$$

*where we used again Lemma 3.5 to obtain the last (worst-case) upper bound. In doing so, we recover and extend the recent results of [8] in a more natural way.[7] Note that the rank 1 assumption is not needed in the continuous relaxation and therefore our results still hold whenever $a(t)a(t)^T$ is replaced by any positive semi-definite matrix $H(t)$.*

## C.4    Statement of Lemma C.2

In order to prove previous property on $\mathbb{V}_\alpha$, a key step mirroring the MAB case is the use of high confidence lower bound on the censorship process, proven using anytime matrix martingale inequalities:

**Lemma C.2.** *([35]) For any $\delta \in ]0,1]$, $\lambda > 0$ and policy $\pi$, let's introduce the event:*

$$\mathcal{H}_{CEN}^{II}(\delta) \triangleq \left\{ \exists t \geq 0, \mathbb{W}_t^C \prec \frac{1}{c_\delta} \mathbb{W}_t \right\},$$

*where $c_\delta \triangleq 8 \max(\frac{\log(d/\delta))}{\lambda}, 1)$. We then have $\mathbb{P}(\mathcal{H}_{CEN}^{II}(\delta)) \leq \delta$.*

Note that picking as in the MAB case $\delta \sim d/T^2$ would lead to a constant $c_\delta = \Theta(\log(T))$, that is a worsening confidence interval, except if we manage to control the initialization. One interesting technical question for future work would be to allow an initialization condition as in Lemma B.2 ensuring $\mathbb{W}(T_0)$ counterbalance $\log(d/\delta)$.

## C.5    Proof of Thm. 4.1

**Theorem 4.1.** *For a given multi-threshold censorship model $\mathcal{MT}$, there exits $d_{eff}$ such that the UCB algorithm with regularization $\lambda$ has an instance-independent expected regret of:*

$$\mathbb{E}[R(T, \pi_{UCB})] \leq \tilde{\mathcal{O}}(\sigma \sqrt{d \cdot d_{eff}} \sqrt{T}).$$

*Proof.* Analogous to the MAB case, we use Lemma 4.2 to deduce:

$$\mathbb{E}[R(T, \pi_{\mathrm{UCB}})] \leq 2\tilde{\beta}_\delta(T) \sqrt{T \mathbb{E}[\mathbb{V}_1(T, \pi_{UCB})]} + \delta T \Delta_{max},$$

where we have:

$$\tilde{\beta}_\delta(T) = \sqrt{\sigma^2 d \log(1 + \frac{T}{d\lambda}) + 2\sigma^2 \log(\frac{1}{\delta})} + \sqrt{\lambda} \|\theta^\star\|_2.$$

We then pick $\delta = \frac{d}{T^2}$, which yields:

$$\mathbb{E}[R(T, \pi_{\mathrm{UCB}})] \leq 2\left(\sqrt{\sigma^2 d \log(1 + \frac{T}{d\lambda}) + 2\sigma^2 \log(\frac{T^2}{d})} + \sqrt{\lambda} \|\theta^\star\|_2\right) \sqrt{T \mathbb{E}[\mathbb{V}_1(T, \pi_{UCB})]} + \frac{d\Delta_{max}}{T}.$$

We then apply Lemma 4.3 with $\alpha = 1$ and $\delta = \frac{d}{T^2}$ to deduce:

$$\mathbb{E}[\mathbb{V}_\alpha(T, \pi)] \leq \frac{d}{\lambda T^2} + 8 \frac{\lambda+1}{\lambda} \max(\frac{2 \log(T)}{\lambda}, 1) \mathrm{Tr} \left( \int_0^T \mathbb{W}(t)^{-1} a(t) a(t)^\top \partial t \right)$$

$$\leq \frac{d}{\lambda T^2} + 8 \frac{\lambda+1}{\lambda} \max(\frac{2 \log(T)}{\lambda}, 1) \max_{\pi \in \Pi} \mathrm{Tr} \left( \int_0^T \mathbb{W}(t)^{-1} a(t) a(t)^\top \partial t \right).$$

By applying Thm. 4.6, we deduce the two possibilities:

---

[7]Yet, we conjecture that the preliminary use of Cauchy Schwartz inequality in the case $\alpha > 1$ to affirm $\sum_{t=1}^{T} \|a_t\|_{\mathbb{W}_{t-1}^{-\alpha}} \leq \sqrt{T \sum_{t=1}^{T} \|a_t\|_{\mathbb{W}_{t-1}^{-\alpha}}^2}$ is suboptimal in this case as it imposes a $\mathcal{O}(\sqrt{T})$ scaling.

- **Case 1: Single region** $i_l$**.** The effective dimension corresponding to this dynamics is $d/p_{i_l}$, with the following equality for $T \geq t_{l-1}$:

$$\max_{\pi \in \Pi} \mathrm{Tr}\left(\int_0^T \mathbb{W}(t)^{-1}a(t)a(t)^\top \partial t\right) = \frac{1}{p_{i_l}} \log \det(\mathbb{W}(T)) + \sum_{n=1}^{l-1}(\frac{1}{p_{i_n}} - \frac{1}{p_{i_{n+1}}})\log \det \mathbb{W}(t_n),$$

where we have for $T \geq t_{l-1}$ $\mathbb{W}(T) = p_{i_l}(T - t_{l-1})\mathbb{W}_{i_l} + \mathbb{W}(t_{l-1})$. Explicit formula of $(t_n, \mathbb{W}(t_n))$ are given for all $n \leq l$ in Cor. D.1.1. We then note that:

$$\frac{1}{p_{i_l}} \log \det(\mathbb{W}(T)) = \frac{1}{p_{i_l}} \log \det(p_{i_l}(T - t_{l-1})\mathbb{W}_{i_l} + \mathbb{W}(t_{l-1}))$$

$$= d_{\mathit{eff}} \log(T) + \frac{1}{p_{i_l}} \log \det(p_{i_l}(1 - \frac{t_{l-1}}{T})\mathbb{W}_{i_l} + \frac{1}{T}\mathbb{W}(t_{l-1})).$$

For $T \geq t_{l-1}$, we then write this in the form:

$$\max_{\pi \in \Pi} \mathrm{Tr}\left(\int_0^T \mathbb{W}(t)^{-1}a(t)a(t)^\top \partial t\right) = d_{\mathit{eff}} \log(T) + f(T),$$

where $f(T) = o(\log(T))$.

- **Case 2: Bi-region** $(i_{l+1}, i_l)$**.** Similarly, for $T \geq t_l$, we have:

$$\max_{\pi \in \Pi} \mathrm{Tr}\left(\int_0^T \mathbb{W}(t)^{-1}a(t)a(t)^\top \partial t\right) = d_{\mathit{eff}} \log(1 + \frac{T - t_l}{t_l + \lambda^\star}) + \sum_{n=1}^{l}(\frac{1}{p_{i_n}} - \frac{1}{p_{i_{n+1}}})\log \det \mathbb{W}(t_n)$$

$$= d_{\mathit{eff}} \log(T) + d_{\mathit{eff}} \log(\frac{1}{T} + \frac{1 - \frac{t_l}{T}}{t_l + \lambda^\star})$$

$$+ \sum_{n=1}^{l}(\frac{1}{p_{i_n}} - \frac{1}{p_{i_{n+1}}})\log \det \mathbb{W}(t_n)$$

$$= d_{\mathit{eff}} \log(T) + f(T),$$

where $f(T) = o(\log(T))$.

Therefore, for given $d_{\mathit{eff}}$, $f$ and $t_0$, we know that the following holds for all $T \geq t_0$:

$$\mathbb{E}[\mathbb{V}_\alpha(T, \pi)] \leq \frac{d}{\lambda T^2} + 8\frac{\lambda + 1}{\lambda} \max(\frac{2\log(T)}{\lambda}, 1) \mathrm{Tr}\left(\int_0^T \mathbb{W}(t)^{-1}a(t)a(t)^\top \partial t\right)$$

$$\leq \frac{d}{\lambda T^2} + 8\frac{\lambda + 1}{\lambda} \max(\frac{2\log(T)}{\lambda}, 1)(d_{\mathit{eff}} \log(T) + f(T)).$$

Putting the pieces together yields for $T \geq t_0$:

$$\mathbb{E}[R(T, \pi_{\mathrm{UCB}})] \leq 2\Big(\sqrt{\sigma^2 d \log(1 + \frac{T}{d\lambda}) + 2\sigma^2 \log(\frac{T^2}{d})} + \sqrt{\lambda}\|\theta^\star\|_2\Big)\sqrt{T}\Big(\frac{d}{\lambda T^2}$$

$$+ 8\frac{\lambda + 1}{\lambda} \max(\frac{2\log(T)}{\lambda}, 1)(d_{\mathit{eff}} \log(T) + f(T))\Big)^{1/2} + \frac{d\Delta_{max}}{T}.$$

By imposing regularization of order $\lambda = o(\log(T))$ only considering the leading order, this yields:

$$\mathbb{E}[R(T, \pi_{\mathrm{UCB}})] \leq \tilde{\mathcal{O}}(\sqrt{(d + 4)\sigma^2}\sqrt{d_{\mathit{eff}}}\sqrt{T}).$$

Finally, by working in large $d$ regime, we finally conclude that:

$$\mathbb{E}[R(T, \pi_{\mathrm{UCB}})] \leq \tilde{\mathcal{O}}(\sigma\sqrt{d \cdot d_{\mathit{eff}}}\sqrt{T}).$$

Again, we note that our proof easily allows to get high-probability bounds on regret instead of bounds on its expected value.

$\square$

## C.6 Extension to Generalized Linear Contextual Bandits

On what follows, we provide a sketch of the extension our results to Generalized Linear Contextual Bandits (GLCB) but differ the complete treatment to future work. In this model, the reward of a given action $a$ is assumed to be of the form:

$$r(a) = \mu(\langle a, \theta^\star \rangle)$$

for a given function $\mu$ strictly increasing, continuously differentiable and real-valued. Notable instances of such a problem include the Logistic bandit and the Poisson bandit. Of particular importance in the dimensionality study of the problem are the constants:

$$L_\mu = \sup_{a \in \cup \mathcal{A}_t} \mu^{(1)}(\langle a, \theta^\star \rangle) \quad \text{and} \quad \kappa = \inf_{a \in \cup \mathcal{A}_t} \mu^{(1)}(\langle a, \theta^\star \rangle).$$

An important requirement of GLCB is the assumption $\kappa > 0$ needed to ensure identifiability of $\theta^\star$ and asymptotic normality. Given this, the suited definition of pseudo-regret considered is:

$$R(T, \pi) \triangleq \sum_{t=1}^{T} \max_{a \in \mathcal{A}_t} \mu(\langle a, \theta^\star \rangle) - \mu(\langle a_t, \theta^\star \rangle)$$

Note that this regret can be easily mapped to the one studied above thanks to the fact that $L_\mu$ is a Lipschitz constant for $\mu$: for all $a, \tilde{a} \in \cup \mathcal{A}_t$, $|\mu(\langle a, \theta^\star \rangle) - \mu(\langle \tilde{a}, \theta^\star \rangle)| \leq L_\mu |\langle a, \theta^\star \rangle - \langle \tilde{a}, \theta^\star \rangle|$. Mirroring the proof of [28], we use a Maximum Likelihood Estimator (MLE) instead of a Least-Square Estimator for $\theta^\star$. More precisely, we define $\hat{\theta}_t^{MLE}$ as the solution of the equation:

$$\sum_{n=1}^{t} \langle a_n, \epsilon_t + \mu(\langle a_n, \theta^\star \rangle) - \mu(\langle a_n, \theta \rangle) = 0$$

A minor difference between the approach of [28] and what precedes is the use of a period of initial random sampling (e.g. *exploration*) instead of the regularization to ensure inversibility of the design matrix $\mathbb{W}_t^C$. More precisely, the initial sampling ensures that with high-probability, $\lambda_{\min}(\mathbb{W}_t^C) > 0$ in a finite time $T_{\text{init}}$. To be possible, this requires the assumption that there exists $\sigma_0^2 > 0$ such that for all $t \geq 1$, we have $\lambda_{min}\left(\mathbf{E}_{a \in \mathcal{A}_t}\left[aa^\top\right]\right) \geq \sigma_0^2$, where the expectation $\mathbf{E}$ is associated with an uniform sampling of actions. Under the same assumption, the impact of censorship on this initialization step is at worst an increase of the sampling time to $\tilde{T}_{\text{init}} \triangleq T_{\text{init}}/p_{\min}$, which is still constant. Following Lemma 9 of [28], we then consider the censored high-probability confidence set for any $\delta \in [\frac{1}{T}, 1]$:

$$\mathcal{H}_{\text{UCB}}^{III}(\delta) \triangleq \left\{ \exists t \geq 0, \|\hat{\theta}_t^{MLE} - \theta^\star\|_{\mathbb{W}_t^C} > \frac{\sigma}{\kappa}\sqrt{\frac{d}{2}\log(1 + 2\frac{t}{d}) + \log(1/\delta)} \quad \text{and} \quad \lambda_{\min}(\mathbb{W}_t^C) > 1 \right\}.$$

and a direct extension of their results allows us to conclude $\mathbb{P}(\mathcal{H}_{\text{UCB}}^{III}(\delta)) \leq \delta$. Note that the constant $\kappa$ appears when upper bounding in the Loewner order the Fischer Information Matrix of the MLE by the matrix $\mathbb{W}_t^C$. Post-initialization, the conditional regret is then upper bounded by:

$$R(T, \pi_{\text{UCB}} | \neg \mathcal{H}_{\text{UCB}}^{III}(\delta)) \leq \tilde{T}_{\text{init}}\Delta_{max} + \sum_{t=T_{\text{init}}}^{T} L_\mu \frac{\sigma}{\kappa}\sqrt{\frac{d}{2}\log(1 + 2\frac{t}{d}) + \log(1/\delta)} \|a_t\|_{(\mathbb{W}_t^C)^{-1}}$$

$$\leq \tilde{T}_{\text{init}}\Delta_{max} + L_\mu \frac{\sigma}{\kappa}\sqrt{\frac{d}{2}\log(1 + 2T/d) + \log(1/\delta)}\sqrt{T\mathbb{V}_1(\pi_{\text{UCB}}, T)},$$

Combining these elements and taking $\delta = \frac{1}{T}$, we conclude that:

$$\mathbb{E}[R(T, \pi_{\text{UCB}})] \leq \tilde{\mathcal{O}}\left(\frac{L_\mu}{\kappa}\sqrt{d}\sqrt{T\mathbb{E}[\mathbb{V}_1(\pi_{\text{UCB}}, T)]}\right) \leq \tilde{\mathcal{O}}\left(L_\mu \frac{\sqrt{d \cdot d_{eff}}}{\kappa}\sqrt{T}\right),$$

where we used Thm. 4.6 to control $\mathbb{E}[\mathbb{V}_1(\pi_{\text{UCB}}, T)]$ as done in the proof of Th.4.1.

# D Effective Dimension and Temporal Dynamics for Multi-Threshold Models

In this section, we prove Thm. 4.6 and discuss its implications. In doing so, we introduce and prove Lemmas D.1, D.2, D.3 and Cor. D.1.1. We conclude the section by illustrating results for the single-threshold model, through Cor. D.3.1.

### D.1 Supplementary Notations

Without loss of generality (i.e. up to an orthogonal transformation), we can consider that $u \equiv e_d$, the $d^{th}$ basis vector. Given this, for two regions $i < j$, we introduce the notations:

$$l(i,j) \triangleq \frac{\sin^2(\rho_i)}{\sin^2(\rho_j)} \quad and \quad u(i,j) \triangleq \frac{\cos^2(\rho_i)}{\cos^2(\rho_j)}$$

$$r^{\star}(i,j) \triangleq \frac{(d-1)u(i,j) + l(i,j)}{d} \quad and \quad r^{\dagger}(i,j) \triangleq \frac{1}{r^{\star}(j,i)} = \frac{dl(i,j)u(i,j)}{u(i,j) + (d-1)l(i,j)}$$

$$\mathbb{W}_i \triangleq \begin{pmatrix} \frac{\cos^2(\rho_i)}{d-1}\mathbb{I}_{d-1} & (0) \\ (0) & \sin^2(\rho_i) \end{pmatrix} \quad and \quad \mathbb{W}(i,j) = \begin{pmatrix} \cos^2(\rho_j)(u(i,j) - \frac{p_i}{p_j})\mathbb{I}_{d-1} & (0) \\ (0) & \sin^2(\rho_j)(\frac{p_i}{p_j} - l(i,j)) \end{pmatrix}.$$

Whenever $i$ and $j$ are clear from context, we use in $u$ (resp. $l$) as abbreviation for $u(i,j)$ (resp. $l(i,j)$).

### D.2 Proof of Thm. 4.6

**Theorem 4.6.** *For a multi-threshold censorship model $\mathcal{MT}$, we have:*

$$\max_{\pi \in \Pi} \int_0^T \frac{1}{p(a(t))} \frac{\partial \log \det(\mathbb{W}(t))}{\partial t} \partial t = d_{eff} \log(T) + o(\log(T)), \quad (\mathcal{P})$$

*where $d_{eff}$ is the effective dimension. Furthermore, $d_{eff}$ is characterized by two cases:*

- **Case 1:** *Single region $j$ effective dimension $d_{eff} = \frac{d}{p_j}$.*

- **Case 2:** *Bi-region $(i,j)$ effective dimension, with $i < j$:*

$$d_{eff} = \frac{1}{p_j}\left[(d-1)\frac{1-l(i,j)}{\frac{p_i}{p_j} - l(i,j)} + \frac{u(i,j) - 1}{u(i,j) - \frac{p_i}{p_j}}\right] < \frac{d}{p_j}. \quad (\mathcal{D})$$

*where $l(i,j) \triangleq \frac{\sin^2(\phi_i)}{\sin^2(\phi_j)}$ and $u(i,j) \triangleq \frac{\cos^2(\phi_i)}{\cos^2(\phi_j)}$.*

---

**Algorithm 2:** Algorithmic description of the dynamics of $\mathbb{W}(t)$

---

**Initalization:** Set current region $S \leftarrow k$
**while** *a region is reachable from region $S$* **do**            /\* Lemma D.1,Fig.3 \*/
    **play** region $S$ optimal policy **until** first reachable region $i^\star$ is reached;
    **if** *region $i^\star$ is dual reachable from region $S$* **then**        /\* Lemma D.2, Fig.4 \*/
        Bi-region $(i^\star, S)$ effective dimension (case 2);                  /\* Lemma D.3 \*/
        **play** Bi-region $(i^\star, S)$ optimal policy;
        **End**;
    **else**
        Update current region $S \leftarrow i^\star$;                      /\* Lemma D.2, Fig.5 \*/
    **end**
**end**
Single region $S$ effective dimension (case 1);                     /\* Lemma D.1 \*/
**play** region $S$ optimal policy;

---

*Proof.* We first summarize the dynamics of the optimal policy of ($\mathcal{P}$) through an algorithmic description in Alg. 2. Two key notions of our analysis are the concepts of reachability and dual reachability of a region $i$ from a base region $j$, as described in Lemmas D.1 and D.2 and schematized in Fig.3, 4 and 5. Formally, they can be written as two independent necessary constraints on the ratio $p_i/p_j$: $p_i/p_j < r^\star(i,j)$ for reachability and $p_i/p_j > r^\dagger(i,j)$ for dual reachability.

The categorization result provided in the statement of Thm. 4.6 follows from the two possible termination condition of the algorithm. We use as algorithmic invariant to ensure the termination the

fact that the set of reachable regions is strictly decreasing for inclusion and finite. Hence, the **while** loop will terminate either because a dual reachable region is reached or because no more regions are reachable. In order to not overload the presentation, time aspect is not present in the algorithmic description but is extensively covered in Lemmas D.1, D.2, D.3 and Cor. D.1.1, as well as in what follows. One of our main finding is that the dynamics of the optimal policy of ($\mathcal{P}$) are described through $\mathbb{W}(t)$ by two qualitatively different regimes. We emphasize that our continuous approach to analyzing cumulative censored potential is key to obtaining these results.

**Transient Regime:** From the **while** loop in the algorithmic description results a so-called transient regime. More precisely, there exists a decreasing sequence of censorship regions $\{i_1 = k, \ldots, i_l\}$ of length $l \in [k+1]$ and associated time sequence $\{t_0 \triangleq 0, t_1, \ldots, t_l\}$ such that whenever $t_j \leq t \leq t_{j+1}$ for a given index $j \leq l - 1$, the evolution of $\mathbb{W}(t)$ is given by:

$$\mathbb{W}(t) = p_{i_{j+1}}(t - t_j)\mathbb{W}_{i_{j+1}} + \mathbb{W}(t_j) = p_{i_{j+1}}(t - t_j)\mathbb{W}_{i_{j+1}} + \sum_{n=1}^{j} p_{i_n}(t_n - t_{n-1})\mathbb{W}_{i_n} + \lambda \mathbb{I}_d.$$

This result follows from a simple induction with repeated use of Lemma D.1, giving the exact sequence of censorship regions, Moreover, closed-formed formula for the time sequence is provided in Cor. D.1.1. We interpret this transient step as an adversarial self-correction of the initial misspecification of censorship at an extra cost. This characterization of transient regime highlights an important consequence of using classical algorithms in censored environments.

**Steady State Regime:** Post-transient regime, the dynamics of $\mathbb{W}(t)$ enter a steady state regime, where one of the two cases necessarily arise:

- **Case 1: Single region $i_l$.** This case arises when the **while** loop ends because no other regions are reachable. It is equivalent to have the last element of the time sequence $t_l$ is equal to $+\infty$ and we have the single region evolution for all $t \geq t_{l-1}$ thanks to Lemma D.1:

$$\mathbb{W}(t) = p_{i_l}(t - t_{l-1})\mathbb{W}_{i_l} + \mathbb{W}(t_{l-1}) = p_{i_l}(t - t_{l-1})\mathbb{W}_{i_l} + \sum_{n=1}^{l-1} p_{i_n}(t_n - t_{n-1})\mathbb{W}_{i_n} + \lambda \mathbb{I}_d.$$

  The effective dimension corresponding to this dynamic is $d/p_{i_l}$, with the following equality for $T \geq t_{l-1}$:

$$\int_0^T \frac{1}{p(a(t))} \frac{\partial \log \det(\mathbb{W}(t))}{\partial t} \partial t = \frac{1}{p_{i_l}} \log \det(\mathbb{W}(T)) + \sum_{n=1}^{l-1} \left(\frac{1}{p_{i_n}} - \frac{1}{p_{i_{n+1}}}\right) \log \det \mathbb{W}(t_n),$$

  where the closed-form formula for $\mathbb{W}(t_n)$ is provided in Cor. D.1.1 for all $n \leq l - 1$.

- **Case 2: Bi-region $(i_{l+1}, i_l)$.** This case arises when the **while** loop ends because dual reachable region $i_{l+1}$ is reached from region $i_l$, with $i_{l+1} < i_l$. For all $t \geq t_l$, Lemma D.2 yields the evolution:

$$\mathbb{W}(t) \propto p_{i_{l+1}}(t + \lambda^\star) \begin{pmatrix} \cos^2(\phi_{i_l})(u(i_{l+1}, i_l) - \frac{p_{i_{l+1}}}{p_{i_l}})\mathbb{I}_{d-1} & (0) \\ (0) & \sin^2(\phi_{i_l})(\frac{p_{i_{l+1}}}{p_j} - l(i_{l+1}, i_l)) \end{pmatrix}.$$

  where $\lambda^\star$ and the proportionality factor are specified in the proof. The corresponding effective dimension is given by ($\mathcal{D}$) and the following equality holds for all $T \geq t_l$ thanks to Lemma D.3:

$$\int_0^T \frac{1}{p(a(t))} \frac{\partial \log \det(\mathbb{W}(t))}{\partial t} \partial t = d_{eff} \log(1 + \frac{T - t_l}{t_l + \lambda^\star}) + \sum_{n=1}^{l} \left(\frac{1}{p_{i_n}} - \frac{1}{p_{i_{n+1}}}\right) \log \det \mathbb{W}(t_n),$$

  where the closed-form formula for $\mathbb{W}(t_n)$ is provided in Cor. D.1.1 for all $n \leq l$.

$\square$

**Remark 6.** *Fig.3 and 5 provide further insights on formula (D) for $d_{eff}$. Throughout the proof and as illustrated on Fig.3, we see that for (D) to arise, $\frac{p_i}{p_j}$ must belong to a certain interval $J \triangleq ]\max(1, r^\dagger(i,j)), r^\star(i,j)[$. As $r^\star(i,j) < u(i,j)$ and $r^\dagger(i,j) > l(i,j)$, we see (D) as a weighted average of the relative distance of $\frac{p_i}{p_j}$ to $u(i,j)$ and $l(i,j)$. Fig.2 provides a sketch of the variations of $d_{eff}$ as $\frac{p_i}{p_j}$ evolves in this interval.*

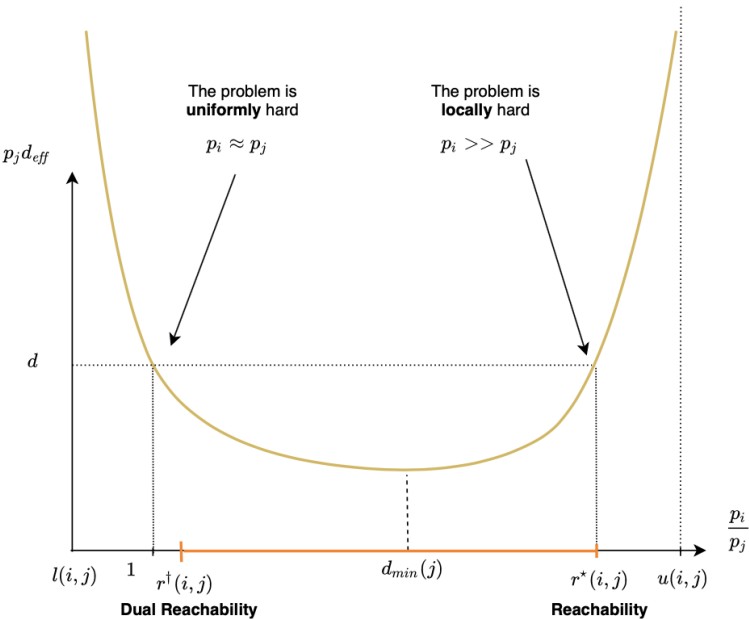

Figure 2: Sketch plot of normalized effective dimension $p_j d_{eff}$ with respect to $\frac{p_i}{p_j}$. We recover the uniform and local hardness conditions mentioned in the discussion of Thm. 4.6, as well as the existence of a *minimum effective dimension* for a certain value of $\frac{p_i}{p_j}$. The necessary conditions of reachability and dual reachability (Lemma D.2 and D.1) verified by $\frac{p_i}{p_j}$ impose that it belongs to the orange segment.

### D.3 Statement and Proof of Lemma D.1

**Lemma D.1** (Reachability Analysis). *Let's assume we start at a given time $t_1$ in transient censored region $j$, with a matrix*

$$\mathbb{W}(t_1) = \begin{pmatrix} \lambda_a \mathbb{I}_{d-1} & (0) \\ (0) & \lambda_b \end{pmatrix},$$

*where $\lambda_a \geq \lambda_b$. We introduce $I_j \triangleq \{i; i < j \quad and \quad \frac{p_i}{p_j} < r^\star(i,j)\}$, the set of reachable regions from region $j$ and affirm that we have the two possible cases:*

- *If $I_j = \varnothing$, i.e. no region is reachable from region $j$, we switch to a steady state regime with single region $j$ effective dimension $d_{eff} = d/p_j$.*

- *Otherwise, next region added to the transient sequence is $i^\star \triangleq \text{argmin}_{i \in I_j} \mu^\star(i, j, \lambda_a, \lambda_b)$, at time $t_2 \triangleq t_1 + \frac{1}{p_j}\mu^\star(i^\star, j, \lambda_a, \lambda_b)$ and we have:*

$$\mathbb{W}(t_2) = \frac{(d-1)\sin^2(\phi_j)\lambda_a - \cos^2(\phi_j)\lambda_b}{d\cos^2(\phi_j)\sin^2(\phi_j)(r^\star(i^\star, j) - \frac{p_i}{p_j})}\mathbb{W}(i^\star, j).$$

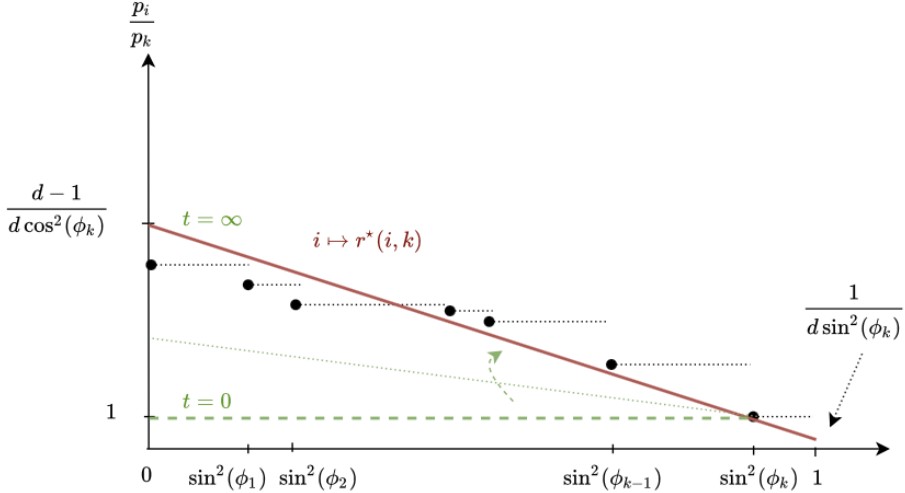

Figure 3: Illustration of the set of reachable regions from a base region $k$, as a function of $\frac{p_i}{p_k}$. Black dots and lines correspond to censorship regions defined by $\mathcal{MT}$. In this figure, we see that a region is reachable if and only if the black dot is below the red reachability line. As time increases, the green line rotates with region $k$ as pivot and asymptotically approaches to the red line. Hence, the first reachable region is the one first *reached* by the green line.

*Proof.* First, we note that the initial starting point is recovered for $t_1 = 0$, base censored state $k$ and $\lambda_a = \lambda_b = \lambda$ but this Lemma allows to go beyond the first step in the study of the behavior of the system. We know the temporal evolution for normalized budget $\mu \triangleq p_1(t - t_1)$ is of the form:

$$\mathbb{W}(t) = \begin{pmatrix} (\mu\frac{\cos^2(\phi_j)}{d-1} + \lambda_a)\mathbb{I}_d & (0) \\ (0) & \mu\sin^2(\phi_j) + \lambda_b \end{pmatrix} = \mu\mathbb{W}_j + \mathbb{W}(t_1).$$

We recall that the set of actions associated with region $j$ is $\{a \in \mathbb{B}_d, \sin(\phi_j) \leq \langle a, e_d \rangle < \sin(\phi_{j+1})\}$. Therefore, the use of Kiefer-Wolfowitz theorem [26] combined with the fact $\lambda_a \geq \lambda_b$ yields that the optimal policy while evolving in region $j$ only plays unit action vector $v_j \equiv (\cos(\phi_j)/(d-1)^{1/2}, \ldots, \cos(\phi_j)/(d-1)^{1/2}, \sin(\phi_j))$. By noting that $v_j v_j^\top = \mathbb{W}_j$, we obtain the formula announced. Reachability of a given state $i < j$ from state $j$ after time $t_1$ is then defined as:

$$\exists t \geq t_t, \quad \frac{1}{p_i}\operatorname{Tr}(\mathbb{W}(t)^{-1}\mathbb{W}_i) = \frac{1}{p_j}\operatorname{Tr}(\mathbb{W}(t)^{-1}\mathbb{W}_j).$$

We interpret this as a classical a first-order optimally condition for convex maximization problems, where the matrix $\mathbb{W}_j$ is weighted by the censorship probability representing the speed of increase in region $j$. We then rewrite this condition as:

$$\exists \mu \geq 0, \quad \frac{1 + f(\mu)\cos^2(\phi_i)}{1 + f(\mu)\cos^2(\phi_j)} = \frac{p_i}{p_j} \quad \text{where} \quad f(\mu) \triangleq \frac{\mu\sin^2(\phi_j) + \lambda_b}{\mu\frac{\cos^2(\phi_j)}{d-1} + \lambda_a} - 1.$$

We know that $f$ is increasing in $\mu$ and the LHS of the equation above is decreasing in $f(\mu)$ as $i < j$. Hence, the reachability condition than stated by looking at the limit of $f$ in $+\infty$. By using the fact that $\lim_{\mu \to +\infty} f(\mu) = \frac{d\sin^2(\phi_j)-1}{\cos^2(\phi_j)}$, we deduce that the reachability condition is equivalent to looking at the position of $\frac{p_i}{p_j}$ with respect to:

$$r^\star(i,j) \triangleq \frac{1 + ud[\sin^2(\phi_j) - \frac{1}{d}]}{d\sin^2(\phi_j)} = \frac{(d-1)u + l}{d} = \frac{1}{d}\operatorname{Tr}(\mathbb{W}_j^{-1}\mathbb{W}_i).$$

On the one hand, if $\frac{p_i}{p_j} \geq r^\star(i,j)$, the state in never reachable in a finite time. On the other hand, whenever $\frac{p_i}{p_j} < r^\star(i,j)$, the state is reachable by investing a budget $\mu^\star(i,j,\lambda_a,\lambda_b)$ such that:

$$f(\mu^\star(i,j,\lambda_a,\lambda_b)) = \frac{1}{\cos^2(\phi_j)} \frac{\frac{p_i}{p_j}-1}{u-\frac{p_i}{p_j}},$$

which in turn involves:

$$\mu^\star(i,j,\lambda_a,\lambda_b) = \frac{d-1}{d\sin^2(\phi_j)\cos^2(\phi_j)} \frac{(\sin^2(\phi_j)\lambda_a + \cos^2(\phi_j)\lambda_b)\frac{p_i}{p_j} - (\sin^2(\phi_i)\lambda_a + \cos^2(\phi_i)\lambda_b)}{r^\star(i,j) - \frac{p_i}{p_j}}.$$

In particular, at $t_1 = 0$ whenever $\lambda_b = \lambda_a = \lambda$ and $j = k$, this gives:

$$\mu^\star(i,k,\lambda,\lambda) = \frac{(d-1)\lambda}{d\sin^2(\phi_k)\cos^2(\phi_k)} \frac{\frac{p_i}{p_k}-1}{r^\star(i,k) - \frac{p_i}{p_k}}.$$

The first reachable region from region $j$ is then defined as $i^\star \triangleq \operatorname{argmin}_{i \in I} \mu^\star(i,j,\lambda_a,\lambda_b)$, where $I \triangleq \{i; i < j \quad and \quad \frac{p_i}{p_j} < r^\star(i,j)\}$. Note that at the moment $t_2 \triangleq t_1 + \frac{1}{p_j}\mu^\star(i^\star,j,\lambda_a,\lambda_b)$ when this region is reached, we have:

$$\mathbb{W}(t_2) = \frac{(d-1)\sin^2(\phi_j)\lambda_a - \cos^2(\phi_j)\lambda_b}{d\cos^2(\phi_j)\sin^2(\phi_j)(r^\star(i^\star,j) - \frac{p_i}{p_j})} \mathbb{W}(i,j).$$

On the other hand, whenever the set $I$ is empty, by definition, the process reaches case 1 steady-state regime and only plays optimal policy of region $j$ for remaining budget. To be fully general, we note that two or more regions can be reached simultaneously. In this case, the optimal policy tie-breaks by taking the region with maximal index i.e. higher censorship, as further described in Lemma D.2. $\qquad\square$

## D.4  Statement and Proof of Cor. D.1.1

More generally, this allows us to deduce the next technical corollary:

**Corollary D.1.1.** *For a sequence of censored regions $\{i_1 = k, \ldots, i_l, i_{l+1}, \ldots\}$, we have for the $l^{th}$ region of the transient sequence, with starting time $t_{l-1}$ and ending time $t_l$:*

$$\mathbb{W}(t_l) = \lambda\mathbb{I}_d + \sum_{n=1}^{l} \mu^\star(i_{n+1}, i_n, \lambda_a^{\mathbb{W}(t_{n-1})}, \lambda_b^{\mathbb{W}(t_{n-1})})\mathbb{W}_{i_n}$$

$$= \frac{\lambda\frac{(d-1)\sin^2(\phi_k)-\cos^2(\phi_k)}{\cos^2(\phi_{i_l})\sin^2(\phi_{i_l})} \prod_{n=1}^{l-1}\left(r^\dagger(i_{n+1}, i_n) - \frac{p_{i_{n+1}}}{p_{i_n}}\right)}{d^l \prod_{n=1}^{l}\left(r^\star(i_{n+1}, i_n) - \frac{p_{i_{n+1}}}{p_{i_n}}\right) \prod_{n=1}^{l-1}\left(u(i_{n+1}, i_n) + dl(i_{n+1}, i_n)\right)} \mathbb{W}(i_{l+1}, i_l),$$

*where $t_l$ is characterized by:*

$$t_l = \sum_{n=1}^{l} \frac{1}{p_{i_n}}\mu^\star(i_{n+1}, i_n, \lambda_a^{\mathbb{W}(t_{n-1})}, \lambda_b^{\mathbb{W}(t_{n-1})}),$$

*and where $\lambda_a^{\mathbb{W}(t_n)}$ and $\lambda_b^{\mathbb{W}(t_n)}$ refer respectively to the upper and lower coefficient of the diagonal matrix $\mathbb{W}(t_n)$.*

*Proof.* We leverage a simple induction reasoning using for $l \geq 1$ the formula given within the proof of lemma D.1:

$$t_l = t_{l-1} + \frac{1}{p_{i_l}}\mu^\star(i_{l+1}, i_l, \lambda_a^{\mathbb{W}(t_{l-1})}, \lambda_b^{\mathbb{W}(t_{l-1})})$$

$$\mathbb{W}(t_l) = \frac{(d-1)\sin^2(\phi_{i_l})\lambda_a^{\mathbb{W}(t_{l-1})} - \cos^2(\phi_{i_l})\lambda_b^{\mathbb{W}(t_{l-1})}}{d\cos^2(\phi_{i_l})\sin^2(\phi_{i_l})(r^\star(i_{l+1}, i_l) - \frac{p_{i_{l+1}}}{p_{i_l}})} \mathbb{W}(i_{l+1}, i_l),$$

and the initialization conditions $t_0 = 0$ and $\mathbb{W}(0) = \lambda\mathbb{I}_d$. $\qquad\square$

## D.5 Statement and Proof of Lemma D.2

**Lemma D.2** (Dual Reachability Analysis). *Let's assume we are currently playing transient region $j$ and we reach the region $i$ at time $t_l$. We then have the following two possible cases:*

- *If $\frac{p_i}{p_j} > r^\dagger(i,j)$, we say that regions $i$ is dual reachable from region $j$, leading to a steady state regime with bi-region $(i,j)$ effective dimension. In such case, for $t \geq t_l$, the potential increase is of the form:*

$$\mathbb{W}(t) = \frac{1}{p_i/p_j + \frac{d}{u+(d-1)l}\frac{r^\star(i,j)-p_i/p_j}{p_i/p_j-r^\dagger(i,j)}}\frac{u-l}{u+(d-1)l}\frac{1}{p_i/p_j - r^\dagger(i,j)}p_i(t+\lambda^\star)\mathbb{W}(i,j).$$

- *Otherwise, we switch from base region $j$ to base region $i$ and continue in the transient regime.*

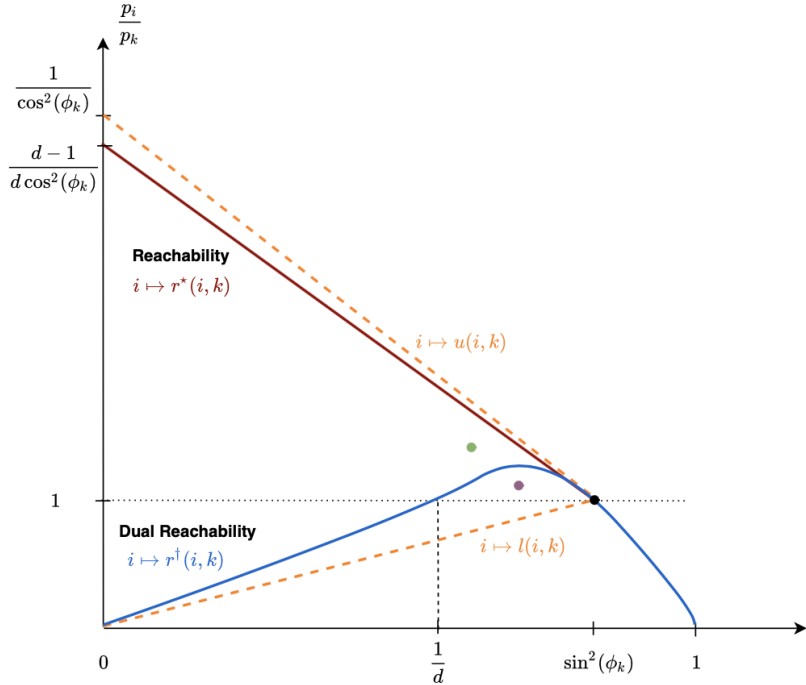

Figure 4: Sketch plot of reachability and dual reachability conditions from base region $k$ associated with the black dot (Lemma D.2 and D.1) as a function of $\frac{p_i}{p_j}$. For a region $i$ to be reachable, $\frac{p_i}{p_j}$ has to be below the red line. For a region $i$ to be dual reachable, $\frac{p_i}{p_j}$ has to be above the blue line. Henceforth, the red dot here is a censorship region that is both reachable and dual reachable whereas the purple dot is a reachable but not dual reachable region. Orange lines represent the functions $u(i,k)$ and $l(i,k)$ introduced above in Sec.D.1.

*Proof.* Using previous section, we know that $\mathbb{W}(t_l) \propto \mathbb{W}(i,j)$ where we recall that the matrix $\mathbb{W}(i,j)$ has the strong property that the gains in regions $i$ and $j$ are equal i.e.:

$$\frac{1}{p_i}\text{Tr}(\mathbb{W}(i,j)^{-1}\mathbb{W}_i) = \frac{1}{p_j}\text{Tr}(\mathbb{W}(i,j)^{-1}\mathbb{W}_j).$$

One of the main result we show in the multi-threshold censorship model is that for $t \geq t_l$, we have:

$$\mathbb{W}(t) - \mathbb{W}(t_l) \propto (t-t_l)\mathbb{W}(i,j),$$

which involves in particular that for $t \geq t_l, \mathbb{W}(t) \propto \mathbb{W}(i,j)$. This is possible thanks to the fact that the optimal policy produces a combination of $p_i\mathbb{W}_i$ and $p_j\mathbb{W}_j$ proportional to $\mathbb{W}(i,j)$ so that

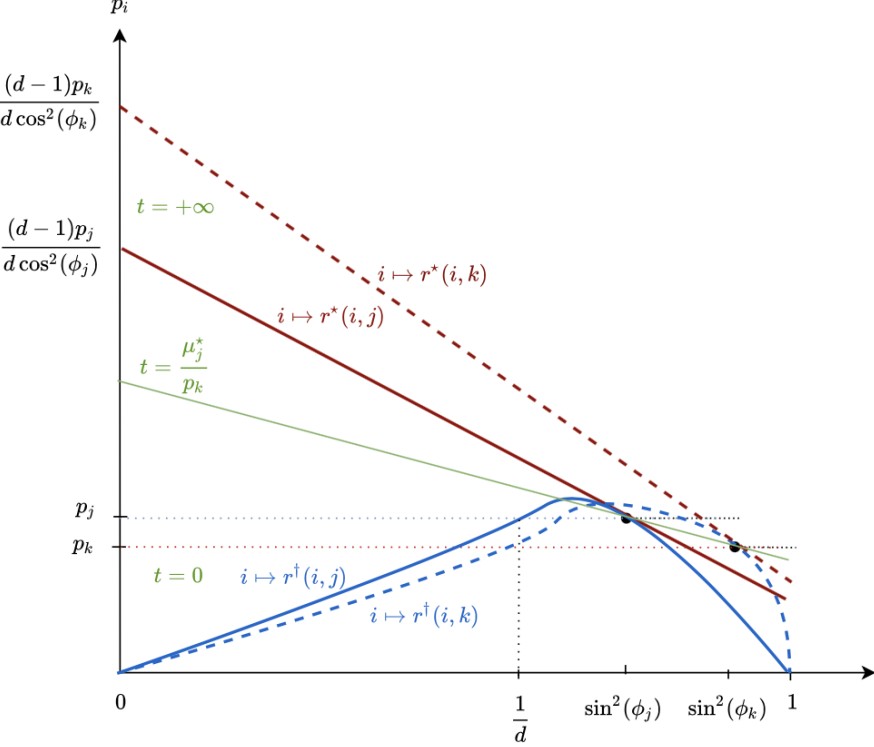

Figure 5: Sketch plot of the evolution of reachability and dual reachability conditions after a region $j$ is reached from region $k$ but is not dual reachable (Else condition in Alg. 2). Doted red (resp. blue) line is reachability (resp. dual reachability) condition for previous region $k$ and full red (resp. blue) lines is reachability (resp. dual reachability) condition for new region $j$. Instead of starting from horizontal line at $t = 0$ to find new reachable state, rotation with region $j$ as pivot is initialized at the green line associated with $t = \frac{\mu_j^\star}{p_k}$. Note that the $y$-axis is not normalized here.

optimally of both regions $i$ and $j$ is maintained while maximal first-order gain is simultaneously ensured. The proportionality condition is then written as the existence of $\mu_i, \mu_j > 0$ such that $p_i \mu_i \mathbb{W}_i + p_j \mu_j \mathbb{W}_j \propto \mathbb{W}(i, j)$ or equivalently as:

$$\exists \mu_i, \mu_j > 0, \quad \frac{\frac{1}{d-1}[p_i \mu_i \cos^2(\phi_i) + p_j \mu_j \cos^2(\phi_j)]}{p_i \mu_i \sin^2(\phi_i) + p_j \mu_j \sin^2(\phi_j)} = \frac{\cos^2(\phi_j)(u(i,j) - \frac{p_i}{p_j})}{\sin^2(\phi_j)(\frac{p_i}{p_j} - l(i,j))} \triangleq R,$$

where $\mu_i$ and $\mu_j$ are the infinitesimal time increase in regions $i$ and $j$. It leads in turn to the ratio equality:

$$\frac{p_i \mu_i}{p_j \mu_j} = \frac{\sin^2(\phi_j)(d-1)R - \cos^2(\phi_j)}{\cos^2(\phi_i) - \sin^2(\phi_i)(d-1)R} = \frac{(d-1)u + l - d\frac{p_i}{p_j}}{(u + (d-1)l)\frac{p_i}{p_j} - dlu} = \frac{d}{u + (d-1)l} \frac{r^\star(i,j) - \frac{p_i}{p_j}}{\frac{p_i}{p_j} - r^\dagger(i,j)}.$$

Thus, we see that bi-region stationarity is possible if and only if $\frac{p_i}{p_j} > r^\dagger(i,j)$ where we introduced the dual reachability condition:

$$r^\dagger(i,j) \triangleq \frac{dl(i,j)u(i,j)}{u(i,j) + (d-1)l(i,j)} = \left(\frac{\frac{d-1}{u(i,j)} + \frac{1}{l(i,j)}}{d}\right)^{-1} = \left(\frac{1}{d} \operatorname{Tr}(\mathbb{W}_i^{-1}\mathbb{W}_j)\right)^{-1} = \frac{1}{r^\star(j,i)}.$$

Hence, the use of the term dual reachability comes from the fact that region $i$ is dual reachable from region $j$ if and only if region $j$ is reachable from region $j$. In such case, further algebraic calculation

then lead to the instantaneous potential increase $\partial W$ for infinitesimal time $\partial t \triangleq \mu_j + \mu_j$:

$$\partial W(\partial t) \triangleq p_j \mu_j \mathbb{W}_j + p_i \mu_i \mathbb{W}_i = \frac{u-l}{u+(d-1)l} \frac{1}{\frac{p_i}{p_j} - r^\dagger(i,j)} p_j \mu_j \mathbb{W}(i,j).$$

We then note that:

$$\frac{\mu_i + \mu_j}{\mu_j} = 1 + \frac{1}{\frac{p_i}{p_j}} \frac{d}{u+(d-1)l} \frac{r^\star(i,j) - \frac{p_i}{p_j}}{\frac{p_i}{p_j} - r^\dagger(i,j)}.$$

Therefore, we conclude that:

$$\partial W(\partial t) = \frac{1}{p_i/p_j + \frac{d}{u+(d-1)l} \frac{r^\star(i,j) - p_i/p_j}{p_i/p_j - r^\dagger(i,j)}} \frac{u-l}{u+(d-1)l} \frac{1}{p_i/p_j - r^\dagger(i,j)} p_i(\mu_j + \mu_i) \mathbb{W}(i,j)$$

$$= \frac{1}{p_i/p_j + \frac{d}{u+(d-1)l} \frac{r^\star(i,j) - p_i/p_j}{p_i/p_j - r^\dagger(i,j)}} \frac{u-l}{u+(d-1)l} \frac{1}{p_i/p_j - r^\dagger(i,j)} p_i \partial t \mathbb{W}(i,j).$$

We then introduce $\lambda^\star$ defined such that:

$$(t_l + \lambda^\star) \mathbb{W}(i,j) \triangleq \frac{1}{p_i} \frac{(u+(d-1)l)(p_i/p_j - r^\dagger(i,j))}{u-l} \left( p_i/p_j + \frac{d}{u+(d-1)l} \frac{r^\star(i,j) - p_i/p_j}{p_i/p_j - r^\dagger(i,j)} \right) \mathbb{W}(t_l).$$

Given the previous two results, we conclude that for all $t \geq t_l$:

$$\mathbb{W}(t) = \frac{1}{p_i/p_j + \frac{d}{u+(d-1)l} \frac{r^\star(i,j) - p_i/p_j}{p_i/p_j - r^\dagger(i,j)}} \frac{u-l}{u+(d-1)l} \frac{1}{p_i/p_j - r^\dagger(i,j)} p_i(t + \lambda^\star) \mathbb{W}(i,j).$$

Note that entering the bi-region stationary regime impedes new regions to be reachable. Indeed, going back to the initial definition of reachability, region $n$ is said to be reachable from region $j$ after time $t_l$ if and only if:

$$\exists t \geq t_l, \quad \frac{1}{p_n} \operatorname{Tr}(\mathbb{W}(t)^{-1} \mathbb{W}_n) = \frac{1}{p_j} \operatorname{Tr}(\mathbb{W}(t)^{-1} \mathbb{W}_j).$$

Yet, using previous result on the evolution of $\mathbb{W}(t)$, we know that the ratio of those two quantities remain equal for any $t \geq t_l$ i.e. no new regions can be reached.

Moreover, using the optimality criterion of Lemma D.1, when several regions are reached simultaneously, the tie-breaking is performed by considering the most censored region, i.e. the one with the highest $i$ index. If the chosen region is not dual reachable, then the next one is considered. In the case where none of them is dual reachable, the base region becomes the maximally censored region and we immediately reiterate the procedure described in Lemma D.2.

$\square$

## D.6 Statement and Proof of Lemma D.3

**Lemma D.3** (Bi-Region Effective Dimension). *Let's assume we reach a bi-region $(i,j)$ steady state regime at time $t_l \leq T$. Then, we have:*

$$\int_{t_l}^T \frac{1}{p(a(t))} \frac{\partial \log \det(\mathbb{W}(t))}{\partial t} \partial t = d_{\textit{eff}} \log(1 + \frac{T - t_l}{t_l + \lambda^\star}) \sim d_{\textit{eff}} \log(T),$$

*where $d_{\textit{eff}} = \frac{1}{p_j} \left[ (d-1) \frac{1 - l(i,j)}{p_i/p_j - l(i,j)} + \frac{u(i,j) - 1}{u(i,j) - p_i/p_j} \right]$ and $\lambda^\star$ is given in the proof of Lemma D.2. Moreover, we have the cumulative transient potential:*

$$\int_0^{t_l} \frac{1}{p(a(t))} \frac{\partial \log \det(\mathbb{W}(t))}{\partial t} \partial t = \sum_{n=1}^l \frac{1}{p_{i_n}} \int_{t_{n-1}}^{t_n} \partial \log \det(\mathbb{W}(t)) = \sum_{n=1}^l \frac{1}{p_{i_n}} \log \frac{\det(\mathbb{W}(t_n))}{\det(\mathbb{W}(t_{n-1}))}$$

$$= \sum_{n=1}^l (\frac{1}{p_{i_n}} - \frac{1}{p_{i_{n+1}}}) \log \det \mathbb{W}(t_n).$$

*Proof.* For $t \geq t_l$, we have the infinitesimal two-step increase $\partial G$ during the infinitesimal time $\partial t \triangleq \mu_i + \mu_j$:

$$
\begin{aligned}
\partial G(\partial t) &\triangleq \mu_i \operatorname{Tr}(\mathbb{W}(t)^{-1}\mathbb{W}_i) + \mu_j \operatorname{Tr}((\mathbb{W}(t) + \mu_i p_i \mathbb{W}_i)^{-1}\mathbb{W}_j) \\
&= \mu_i \operatorname{Tr}(\mathbb{W}(t)^{-1}\mathbb{W}_i) + \mu_j \operatorname{Tr}(\mathbb{W}(t)^{-1}\mathbb{W}_j) + o(\partial t) \\
&= \frac{p_i \mu_i + p_j \mu_j}{p_j} \operatorname{Tr}(\mathbb{W}(t)^{-1}\mathbb{W}_j) + o(\partial t),
\end{aligned}
$$

where we used the property of $\mathbb{W}(i,j)$. Invoking lemma D.2, we know the evolution of $\mathbb{W}(t)$ for $t \geq t_l$:

$$
\mathbb{W}(t) = \frac{1}{1 + \frac{1}{p_i/p_j}\frac{d}{u+(d-1)l}\frac{r^\star(i,j)-p_i/p_j}{p_i/p_j-r^\dagger(i,j)}} \frac{u-l}{u+(d-1)l} \frac{1}{p_i/p_j - r^\dagger(i,j)} p_j(t+\lambda^\star)\mathbb{W}(i,j),
$$

as well as the relations between $\mu_i$ and $\mu_j$:

$$
\begin{cases}
\frac{p_i \mu_i + p_j \mu_j}{p_j} &= \mu_j(1 + \frac{d}{u+(d-1)l}\frac{r^\star(i,j)-p_i/p_j}{p_i/p_j-r^\dagger(i,j)}) \\
\frac{\mu_i + \mu_j}{\mu_j} &= 1 + \frac{1}{p_i/p_j}\frac{d}{u+(d-1)l}\frac{r^\star(i,j)-p_i/p_j}{\frac{p_i}{p_j}-r^\dagger(i,j)}.
\end{cases}
$$

We invoke the fact that $\operatorname{Tr}(\mathbb{W}(i,j)^{-1}\mathbb{W}_j) = \frac{1}{u-p_i/p_j} + \frac{1}{p_i/p_j-l}$ to conclude that:

$$
\begin{aligned}
\partial G(\partial t) &= \frac{1}{p_j} \frac{[(d-1)l+u-d]\frac{p_i}{p_j} - [dlu - ((d-1)u+l)]}{(u-\frac{p_i}{p_j})(\frac{p_i}{p_j}-l)} \frac{(1 + \frac{1}{p_i/p_j}\frac{d}{u+(d-1)l}\frac{r^\star(i,j)-p_i/p_j}{p_i/p_j-r^\dagger(i,j)})\mu_j}{t+\lambda^\star} \\
&= \frac{1}{p_j}\left[(d-1)\frac{1-l}{\frac{p_i}{p_j}-l} + \frac{u-1}{u-\frac{p_i}{p_j}}\right]\frac{\partial t}{t+\lambda^\star} \\
&= d_{\text{eff}}\frac{\partial t}{t+\lambda^\star}.
\end{aligned}
$$

Given that $\partial t$ is an infinitesimal time increase, we have in the steady state regime:

$$
\int_{t_l}^T \partial G = d_{\text{eff}}\int_{t_l}^T \frac{\partial t}{t+\lambda^\star} = d_{\text{eff}}\log(\frac{T+\lambda^\star}{t_l+\lambda^\star}) = d_{\text{eff}}\log(1 + \frac{T-t_l}{t_l+\lambda^\star}).
$$

We finally note that the cumulative potential coming from the transient period is equal to:

$$
\begin{aligned}
\int_0^{t_l} \partial G &= \sum_{n=1}^l \frac{1}{p_{i_n}}\int_{t_{n-1}}^{t_n} \partial \log\det(\mathbb{W}(t)) = \sum_{n=1}^l \frac{1}{p_{i_n}}\log\frac{\det(\mathbb{W}(t_n))}{\det(\mathbb{W}(t_{n-1}))} \\
&= \sum_{n=1}^l (\frac{1}{p_{i_n}} - \frac{1}{p_{i_{n+1}}})\log\det\mathbb{W}(t_n),
\end{aligned}
$$

where the closed-form expression of $\mathbb{W}(t_n)$ is given in Corollary D.1.1. $\square$

### D.7 Special case: Single-threshold model

**Corollary D.3.1.** *For the single threshold model with two regions $0$ and $1$ and associated censorship probabilities $p_0 < p_1$, our main theorem yields:*

- *If $\frac{p_0}{p_1} < \frac{d-1}{d\cos^2(\phi_1)}$, then we reach bi-region steady state regime and have the effective dimension:*

$$
d_{\text{eff}} = \frac{d-1}{p_0} + \frac{1}{p_0}\frac{\sin^2(\phi_1)}{\frac{p_1}{p_0} - \cos^2(\phi_1)} \in [\frac{d}{p_0}, \frac{d}{p_1}].
$$

- *Otherwise, we are from $t = 0$ in single-region steady state regime and have the effective dimension $d_{\text{eff}} = d/p_1$.*

*Proof.* Using Lemma D.2 in the case of the single threshold model, we note that if region $0$ is reachable, it is necessarily dual reachable given that $r^\dagger(0,1) = 0$ and henceforth, we always have $p_0/p_1 > r^\dagger(0,1)$. Thanks to the results of Lemma D.1, we also note that $r^\star(0,1) = \frac{p_0}{p_1} < \frac{d-1}{d\cos^2(\phi_1)}$ and that if region $0$ is reachable, it is done in a time:

$$t_1 = \frac{1}{p_1} \frac{(d-1)\lambda}{d\sin^2(\phi_1)\cos^2(\phi_1)} \frac{\frac{p_0}{p_1} - 1}{\frac{d-1}{d\cos^2(\phi_1)} - \frac{p_0}{p_1}}$$

$\square$