# OpenReview forum: "Effective Dimension in Bandit Problems under Censorship"
_NeurIPS.cc/2022/Conference — NeurIPS 2022 Accept_

### Official Review · Reviewer_R7Ls · 2022-07-11

**Rating:** 3
**Confidence:** 2
**Soundness:** 2 fair
**Presentation:** 3 good
**Contribution:** 2 fair

**Summary:**

The authors study bandits under censorship.
The key contribution is to provide the theoretical performance (the extra risk due to the censorship).

**Questions:**

In the introduction, the paper mentions the difference from the Rubin's missing-completely-at-random.  But more details on the difficulty in online learning are need.  It also help to assess the theoretical contribution of this paper.

The relation to the delayed bandits is not clear. It seems the result is similar to the delayed bandit, although the effective dimension is a new concept from this paper.

In the definition of the censorship model, the parameters are fixed but rely on the action a_t? If so, p_a is fixed given a, although p_{a_t} varies over t.  How much is the difference of missing-completely-at-random?

It seems that a key notation is ignored. \tilde{\Delta}_t^{\lambda} is ignored. It lets reading the paper difficulty.  Because the regret analysis seriously depends on how to address the censorship in the algorithm.  This issue is important for understanding this paper.

**Limitations:**

The Generic UCB algorithm relies on a previous work.   A clear presentation should be needed.

**Strengths And Weaknesses:**

Strengths: the paper first provides a new concept, effective dimension, to measure the effect of the censorship in the bandits.

Weaknesses:
Because the regret analysis should depends on how to handle the censorship, the effective dimension may be specific to algorithm.
As a pure theoretical paper on effective dimension in bandits with censorship, i don't think the contribution is solid.

I know little about the censorship, so i may be ignoring the theoretical difficulty and contribution.

---

> ### Author Response · Authors · 2022-08-02
> **Response to reviewer R7Ls questions 1-2**
>
> We thank the reviewer for their comments and feedback. Please see our answer to questions 1-2 below:
>
> > 4.1) _“Because the regret analysis should depends on how to handle the censorship, the effective dimension may be specific to algorithm.“_
>
> We thank the reviewer for raising this point.
>
> Firstly, we do not see the fact that the effective dimension may be specific to algorithm as a strong limiting constraint of our paper, given that a rigorous analysis UCB class of algorithms under censorship is our main focus; we refer to reviewer to L64-70 and  our response 2.2. to reviewer hhGV.
>
> To the best of our knowledge, our work provides the first dimension analysis in the context of censored bandits from a broad set of bandits and censorship models. Defining a generic dimension measure would be a natural follow-up step instead of a first accomplishment.
>
> We also note that while minimax lower bounds typically allow to prove the tightness of regret guarantees for the order of the budget variable $T$, it is notoriously more challenging to get similar algorithmic agnostic results for the dimensionality aspect d. This remains pretty much an open question, as pointed out for instance in [2], p8.
>
> Nevertheless, we advocate that the effective dimension we introduce goes beyond the simple context of the UCB class. Indeed, it results from the characterization of the cumulative censored potential, which is a quantity that can be related to uniform learning and out of sample generalization in the statistical learning community. Lemma 3.3 and 4.2 reduce the study of the performance of UCB to the control of this learning based quantity, which appears in the study of variants of UCB or other design principles (such as Thompson Sampling or Information Directed Sampling). In particular, for MAB models, the use of similar techniques of Thm. 4 in [3] can allow us to prove a matching lower bound for the effective dimension we provide. On the other hand, for LCB models, new tools should be developed and it is a fascinating open question that would deserve an independent treatment. Our results on LCB are a first step in this direction.
>
>
> > 4.2) _“As a pure theoretical paper on effective dimension in bandits with censorship, i don't think the contribution is solid. I know little about the censorship, so i may be ignoring the theoretical difficulty and contribution. “_
>
> We humbly disagree with the reviewer’s comment. After reading our response below and to other comments, if the reviewer still thinks that our contribution is not solid, we will very much welcome follow-up arguments for further discussion.
> We argue that the results we report are of significant importance to the bandit community and provide a novel approach to tackle  censored and delayed information environments. We do agree about your subsequent comments on the need to clarify minor points – we will be happy to do this in our revised version.  As a brief summary:
>
> - In the context of MAB, we formally introduce the notion of effective dimension, provide the most comprehensive characterization of censored bandits (Sec. 3.1) and instantiate a new and generic proof system that can be applied to more general settings (Sec. 3.2). We also provide new technical insights on the adaptivity gain, initiating a new approach to quantify the relative benefit of adaptive policy and deriving a strong set of results (Sec. 3.3.).
>
> - Furthermore, we introduce and solve the first censored contextual bandit model (Sec. 4.1). We give the parametric description of the effective dimension, discover a newer two phase phenomena of both practical and theoretical interest (Sec. 4.2, 4.3, 4.4 and SI). In order to tackle some significant theoretical difficulties, we introduce new tools drawing from martingale concentration literature, convex optimization and experimental design. We claim that beyond the censored case, these tools can provide new insights for classical settings in the bandit literature (Sec. 4.2).

---

> > ### Comment · Reviewer_R7Ls · 2022-08-09
> > **thanks for the responses**
> >
> > Thanks for taking the time to respond. I read the rebuttal and my opinion has not changed due to that i don't think my concerns are addressed.

---

> ### Author Response · Authors · 2022-08-02
> **Response to reviewer R7Ls questions 3-5**
>
> Please see our answer to questions 3-5 below:
>
> >4.3) _“In the introduction, the paper mentions the difference from the Rubin's missing-completely-at-random. But more details on the difficulty in online learning are need. It also help to assess the theoretical contribution of this paper.”_
>
> **Beyond missing-completely-at-random process:** Although conditional on the choice of a given action the missing data/censorship process is an instance of missing-completely-at-random (MCAR), the online action generating process adds a significant difficulty to the problem and is still an open area of research beyond simple settings. Whereas MCAR is typically studied under a distributional assumption (eg i.i.d. generation of action), this hypothesis does not hold here and is replaced by a weaker adaptive data generation w.r.t. the filtration of past information. Therefore, the structure of the missing data set encompasses strong dependencies with past realization of the  censorship. This strongly couples the dynamics of learning and acting: since censorship mediates current information of the environment, it impacts the outcome of data-driven decision process; this in turn conditions the future decisions and future censored feedback, creating a complex and endogenous joint temporal dependency. In order to tackle this coupling, we use subtle martingale concentration results and stochastic process analysis which are new to the missing data field (Lemma B.1, B.2, 3.6, 3.7, C.1 and C.2). Finally, as briefly discussed in the answer 2.4 to reviewer qkWU, we believe that our potential based framework can be extended beyond MCAR.
>
> > 4.4) _“The relation to the delayed bandits is not clear. It seems the result is similar to the delayed bandit, although the effective dimension is a new concept from this paper.”_
>
> **Beyond delayed bandits:** As stated in the introduction, censorship is partially related to delay as a censored can be interpreted as action with infinite delay. Yet, most of the current results on delayed bandits (mentioned in L59-63) are inapplicable for the censored case as they rely on strong assumptions on the delay that would lead to linear regret (either that the delay is constant, upper bounded, has a finite mean, or simply provide regret guarantees that are linear in the cumulative delays). Under such assumptions on delay, one usually gets a second order additive dependency of the regret in terms of delay parameters, which informally says that delay is benign for bandits.
> On the other hand, we show that censorship leads to a first order multiplicative dependency on regret and we provide a complete characterization of this dependency for a wide range of bandits and censorship models. As mentioned in L46-50 and L54-55, some previous works observed this dependency but only for simple MAB models and failed to deliver a dimensionality interpretation. Finally, most of the building blocks in our proof techniques are new to both the censored and delayed literature.
>
> > 4.5) _“In the definition of the censorship model, the parameters are fixed but rely on the action_ $a_t$_? If so, _$p_a$_ is fixed given_ $a$, _although_ $p_{a_t}$ _varies over t. How much is the difference of missing-completely-at-random?”_
>
> We thank the reviewer for raising awareness on this point and we updated the paper to make this dependency clearer. As explained above, conditional on the choice of a given action the missing data/censorship process is indeed an instance of missing-completely-at-random (MCAR). Yet, difficulty arises from combining the MCAR process with the online nature of the data generating process, as explained in answer 4.3.

---

> ### Author Response · Authors · 2022-08-02
> **Response to reviewer R7Ls questions 6-7**
>
> Please see our answer to questions 6-7 below:
>
> > 4.6) _“It seems that a key notation is ignored. \tilde{\Delta}_t^{\lambda} is ignored. It lets reading the paper difficulty. Because the regret analysis seriously depends on how to address the censorship in the algorithm. This issue is important for understanding this paper.”_
>
> **Minor readability issue:** We agree with the reviewer that the use of the optimistic gap estimator $\tilde{\Delta}_t^{\lambda}$ may have added a layer of complexity in terms of readability (requiring a certain  level of familiarity with the UCB design principle). We have now removed the use of this quantity in favor of a more interpretable reward estimator, further described in L125-127 and in SI A.1. Reduction from the error in estimator to the cumulative censored potential is a key technique in the analysis of both UCB and Thompson Sampling and is provided in Lemma 3.3 and 4.2. As most of our contributions tackle the study of the cumulative censored potential, this is why we favored detailed study of the latter, as the key object of our analysis (Prop. 3.4, 3.7, 4.3 and Lemma 3.6 and 4.4 among others).
>
> > 4.7) _“The Generic UCB algorithm relies on a previous work. A clear presentation should be needed.”_
>
> **Presentation of generic UCB for background knowledge:** As detailed in the answer to reviewer hhGV (1.2), one of the main focus of this paper is in understanding how UCB, used out of the box and without any adaptations, behaves under censorship. he generic UCB was formally introduced by [1] and it has been folklore in the bandit literature. We implicitly followed this style of writing. One can just understand it as a way to present the design principle of the UCB class of algorithms for different operating environments at once (MAB, LCB…) instead of instantiating it again each time. This point can be easily further detailed beyond what is currently done in Sec. A.1 and we’ll be glad to do it.
>
>
> **Biblio: **
>
> [1] “Information Directed Sampling and Bandits with Heteroscedastic Noise”, Kirschner, Johannes and Krause, Andreas, 2018
>
> [2] “Beyond UCB:Optimal and Efficient Contextual Bandits with Regression Oracles”, Dylan J. Foster and Alexander Rakhlin, 2020
>
> [3] “Stochastic multi-armed bandits with unrestricted delay distributions”, Tal Lancewicki, Shahar Segal, Tomer Koren, and Y. Mansour, 2021

---

### Official Review · Reviewer_5N2o · 2022-07-12

**Rating:** 6
**Confidence:** 1
**Soundness:** 3 good
**Presentation:** 2 fair
**Contribution:** 3 good

**Summary:**

This paper analyzes the regret of UCB under different censorship models using the effective dimension of the problem, leading to other insights including the learning process during censorship and the elliptical potential inequality. Both multi-armed and linear contextual bandits are included.

**Questions:**

See above.

**Limitations:**

Yes.

**Strengths And Weaknesses:**

**Strengths**

The theoretical results are thorough and the paper is generally well-written, though dense.


**Weaknesses**

While the main results, e.g. Theorem 3.1 and Prop 3.2, look familiar and reasonable, I could have personally benefitted from more exposition and intuition on some of the more complex expressions introduced. Examples that come to mind include the cumulative potential under censorship (L172) and $H_{CEN}$ (L182).  I.e. what is the intuition of $\mathbb{V}_\alpha$ at different $\alpha$?

Overall, I found sections 3.2-3.3 to be quite dense and difficult to interpret—possibly due to my lack of background on censorship.

---

> ### Author Response · Authors · 2022-08-02
> **Response to reviewer 5N2o questions 1-2**
>
> We thank the reviewer for their comments and feedback. Please see our answer to questions 1-2 below:
>
> > 3.1) _“While the main results, e.g. Theorem 3.1 and Prop 3.2, look familiar and reasonable, I could have personally benefitted from more exposition and intuition on some of the more complex expressions introduced.”_
>
> Thank you for this constructive suggestion about exposition and intuition. We agree with you that the presentation of the paper appears to be  dense and compact, because we elected to present the results under some tight space constraints and favored mathematical rigor to high level insights. We further detail the results and insights of Sec. 3.2 and 3.3 below and we will be happy to include these in our final version (and where appropriate in the Supplementary Information). If you judge it relevant, we’ll be glad to add more figures in the style of fig.2-5 illustrating the technical details of section 3.
>
> > 3.2) _“Examples that come to mind include the cumulative potential under censorship (L172) and_ $H_{CEN}$ _(L182). I.e. what is the intuition of_ $\mathbb{V}_{\alpha}$ _at different_ $\alpha$ ?”
>
> $H_{\text{CEN}}$ is a “bad” event associated with the non-concentration of censorship over all timestep and actions. More precisely, it encompasses the event where there is a significant gap between the realized and expected number of observed rewards. Under this event, the UCB algorithm fails to perform well due to a lack of received feedback. Since this event happens with a small probability (Lemma B.2 for MAB and C.2 for LCB), we focus on its complement for our analysis of the leading order of the regret (L553-554). This allows us to lower bound for each action the realized number of  reward observations by a multiple of the number of times this action was selected and thus removing the randomness induced by the censorship.
>
> **Cumulative potential:** Without censorship, the cumulative potential (L173) translates the average rate of decay of the uncertainty on the reward of the different arms as a function of the number of observations. It is intimately related to a measure of the divergence between the true distribution of the reward and the empirical distribution of the observed reward for each arm [1]. Introducing censorship transforms a deterministic decay rate into a stochastic decay where learning happens if and only if the reward is observed. For a typical distribution of rewards, the pace of decay is proportional to a term in $n^{-\alpha}$ or can be upper bounded by such a term (see for e.g. [1]), where $n$ is the number of observed rewards. Therefore, a higher $\alpha$ corresponds to faster learning.

---

> ### Author Response · Authors · 2022-08-02
> **Response to reviewer 5N2o question 3**
>
> Please see our answer to question 3 below:
>
> > 3.3) _“Overall, I found sections 3.2-3.3 to be quite dense and difficult to interpret—possibly due to my lack of background on censorship.“_
>
> **Discussion on Section 3.2:** The goal of section 3.2 is to instantiate our potential-based proof approach in the context of MAB in order to prove Thm. 3.1 and Prop. 3.2. As stated in L164-171, our proof contribution can be understood as follows:
>
> - in Lemma 3.3, we first control the performance of the UCB algorithm by upper bounding its regret by the cumulative potential under censorship, a quantity with a learning interpretation which is further detailed in answer 3.2 above. In other words, our approach can be interpreted as a way to map the complexity of decision making under uncertainty to the complexity of learning this uncertainty in a censored environment.
>
> - Prop. 3.4 then provides a tight worst-case upper bound on this complexity of learning i.e. the expected cumulative censored potential. As detailed in L198-199, the bound is mainly driven by the first factor for large $T$ but features additional terms induced by the non-asymptotic guarantees. It can be interpreted as a quantification of the learning difficulty of the problem: it increases with $d_{eff}$, the dimensionality and decreases with alpha the rate of learning. Moreover, it is sublinear in $T$: on average learning happens.
>
> - The last part of the section provides intuition to prove Prop. 3.4 by breaking the proof in two steps as explained in L181-183. As a high level interpretation, we first map the censored learning problem to an uncensored but harder one. and then assess the worst-case complexity of learning for the latter.
>
> **Discussion on Section 3.3:** To begin with, it is well known that adaptability is the key feature of sequential decision problems: optimal policies use feedback from previous decisions to decide the next action to take based on the data, and in comparison non-adaptive policies are suboptimal. Somewhat interestingly,, the main result of this section is that adaptability in the context of censoring does not provide a significant advantage to the decision maker. Precisely, being able to observe which decisions have been censored and reacting to this does not bring more than a second order gain. In proving this, we quantify and gain insight into the expected performance of policies that are adaptive to the realization of the censoring process compared to a class of non-adaptive (i.e., offline) policies. This allows us to complete the characterization of the learning complexity introduced in Prop. 3.4. While we believe that this  is an important contribution to the literature, it is one of the more complex sections of the paper and may be overlooked on first reading.
>
> **Biblio:**
> [1] “Adaptive Sampling for Estimating Probability Distributions”, Shekhar, Shubhanshu and Javidi, Tara and Ghavamzadeh, Mohammad, 2020

---

### Official Review · Reviewer_qkWU · 2022-07-13

**Rating:** 7
**Confidence:** 4
**Soundness:** 4 excellent
**Presentation:** 3 good
**Contribution:** 4 excellent

**Summary:**

The authors provide analysis on UCB-like algorithms for the case where rewards are censored with action-dependent probability. They show that regret bounds are, loosely-speaking, worsened in the same way as would happen for uncensored problems where the dimensionality of the problem is increased.

**Questions:**

In the "Information Structure" paragraph 120-124 it took me a moment to understand whether the Bernoulli parameter p_{a_t} depends on the value of a_t. In the non-contextual/non-conditional case, is there a different censoring probability for each action value? Or is there just one Bernoulli parameter.

If there are action-specific Bernoulli parameters: Alg 1 only updates the gap estimator when a reward is observed. Are there variants that also take into account with which probability each action's reward's are observed? (using an estimate of the p's)? Are there any advantages to such algorithms? I'm thinking of re-weighting estimators in causal inference and survival analysis. What analysis is possible w.r.t. your dimensionality and regret results for such algorithms, if they exist / are relevant?

For the contextual bandits, it seems that the censoring model is such that p (the observation probability) depends on the action a. What about cases where it also depends on the context? (by the way the context seems not to be instantiated as a variable in 4.1?) If censoring depends on the context also, which of your analysis would be possible and which wouldn't?

**Limitations:**

Yes. But please clarify which censoring your analysis applies to and which kind of censoring would be hard to analyze using your framework.




**Strengths And Weaknesses:**

In general it is valuable work and I support the acceptance of this work. It is both novel and well-written.

I will only ask a few questions and make a few suggestions.

general writing/grammar/typos/acronyms/etc

you don't fully ever define "regularization" in the UCB algorithms. Doing so will help the reader.

UCB first appears without its full name in line 56

"leveraging Kaplan-Meier estimator" in line 57 should be "leveraging the Kaplan-Meier estimator"

when you say "a notable exception includes [38]" in lines 64-65, it would be good to mention
what is distinct about it and your relationship to that work. Otherwise you are asking to the reader to do extra work to figure out why you are mentioning it.

MAB is defined in lines 85 and in 106

---

> ### Author Response · Authors · 2022-08-02
> **Response to reviewer qkWU questions 1-2**
>
> First, we thank the reviewer regarding general writing/grammar/typos/acronyms, all of them being corrected in the updated version. Please see our answer to questions 1-2 below:
>
> > 2.1) _“In the "Information Structure" paragraph 120-124 it took me a moment to understand whether the Bernoulli parameter_ $p_{a_t}$ _depends on the value of_ $a_t$. _In the non-contextual/non-conditional case, is there a different censoring probability for each action value? Or is there just one Bernoulli parameter."_
>
> **Information structure:** We agree with the reviewer that this point is a bit confusing in the current version. In both finite and linear settings, the Bernoulli parameter corresponding to the censorship probability depends on the action chosen i.e. our model allows the censorship to be heterogeneous across actions. In the finite case, we use the notation $p_{a}$ instead of $p(a)$ for a given action a to denote this relation and operate under full generality on the values of $p_{a}$.
> Similarly, for the linear model, $p$ is action-dependent but we further assume that it follows a Multi-Threshold model, as detailed in Sec. 4.1. Given that the action chosen at time t is a random variable, $p(a_{t})$ refers to a random variable as well. This is the key challenge in addressing by censoring models and one of the main contributions of our work is to develop the tools to fully address it. By assuming only homogeneous models, previous literature has conveniently sidestepped the problem.
>
> > 2.2) _“If there are action-specific Bernoulli parameters: Alg 1 only updates the gap estimator when a reward is observed. Are there variants that also take into account with which probability each action's reward's are observed? (using an estimate of the p's)? Are there any advantages to such algorithms? I'm thinking of re-weighting estimators in causal inference and survival analysis. What analysis is possible w.r.t. your dimensionality and regret results for such algorithms, if they exist / are relevant?“_
>
> We thank the reviewer for this excellent question, which is one of the fascinating open problems raised by our work and a question we would love to tackle in future developments.
>
> **UCB variants:** In order to compare such UCB “variants”, we find it useful to reason in terms of information sources: our current work leverages the noisy reward signal, conditional on the fact that it is realized to learn the latent state. To illustrate it in the context of e-commerce, the UCB algorithm described in this paper uses the feedback/reward of users conditioned on the fact that users followed the recommendation in order to learn its preferences (setting A). Another variant (setting B) could only focus on the reaction of the user to learn its preferences by assuming a parametric model for the decision (typically a Multinomial Logit model in the literature, briefly mentioned in C.6). A last UCB variant, as you have suggested, would use both the reward and the decision to learn and act (setting C).
> We view the notion of effective dimension as the natural measure of the information complexity for each of these settings (i.e., information sources) and hence a natural way of comparing their difficulty. In our paper, we define and compute this measure for setting A and briefly pinpoint the equivalent quantity for setting B in C.6. The setting C is still an open question with no algorithm or theoretical insights to the best of our knowledge. A satisfactory answer likely involves a trade-off between the relative information gain of the observation decision and of the reward: learning the p for uninformative decisions models will likely maintain the original effective dimension of setting A whereas rational and deterministic ones should allow to reduce the effective dimension to a logarithmic factor of the original dimension (by using a binary search of best action).
>
> Finally, given the finite action set in the MAB case, it is indeed possible to use re-weighting estimators to guide the decision making process without assuming that the censorship conveys information on the latent state. However, it does not lead to significant gain in MAB beyond second order effect. Indeed, it allows to reduce the variance of the estimate but each arm still has to be queried to gain information on the reward and henceforth our effective dimension result still holds.

---

> ### Author Response · Authors · 2022-08-02
> **Response to reviewer qkWU questions 3-4**
>
> We thank the reviewer for their comments and feedback. Please see our answer to questions 3-4 below:
>
> > 2.3) _“For the contextual bandits, it seems that the censoring model is such that p (the observation probability) depends on the action a. What about cases where it also depends on the context? (by the way the context seems not to be instantiated as a variable in 4.1?) If censoring depends on the context also, which of your analysis would be possible and which wouldn't?“_
>
> **Contextual bandits:** It is true that the context related part of the contextual bandits section is very briefly mentioned and we thank the reviewer for calling our attention to this point. As commonly done in the literature, we assume that the reward is a linear function of a feature map of both the context and action characteristics. For sake of modeling generality, we do not assume that the context is drawn iid from a given distribution. In fact, our setting even includes adversarial context generation processes. As mentioned in L116-117, this is equivalent to allowing the action set to be generated by an oblivious adversary and this is why context variables are not directly instantiated in the paper. Given this, our censoring model does have a dependency on the context although through the feature map. On the one hand, adding a supplementary dependency on an exogenous element of context would likely lead to considering the empirical or population average of the censorship probability in the definition of the effective dimension and would nicely fit in our proof framework using supplementary concentration results. One the other hand, a dependency on an endogenous contextual element is more likely to require a case-by-case analysis in function of the relation between this element and the original feature map.
>
> > _2.4) “Please clarify which censoring your analysis applies to and which kind of censoring would be hard to analyze using your framework.“_
>
> **Censoring in MAB:** Regarding MAB, our censorship models apply to any type of action based censorship model verifying the Bernoulli generation process described in L121-125 (Missing Completely At Random, see review R7Ls, question 3). To the best of our knowledge, it is the most comprehensive result of current censorship literature on MAB. Removing the Missing Completely At Random property will create a bias that would need to be accounted for in the UCB design. Yet, we conjecture that our potential based analysis will still hold to derive guarantees for this censorship. Another potential feature of censorship is a time dependent character (e.g. Markov chain or more general stochastic processes): again, our framework easily extends if the change is driven by exogenous variables and requires more complex tuning in the context of endogenous variables.
>
> **Censoring in Contextual Bandits:** Here we operate under a restrictive yet fairly generic censorship model (Sec. 4.1.). The two main constraints of the Multi-Threshold model are the radial aspect (the censorship probability depends on the action through a scalar product with a given vector) and the monotonic property (the censorship is monotone in the value of this scalar product). We conjecture that our results still hold if we remove the monotone aspect although it will require some modification in the proofs of section D. We also believe that the radial property is a key constraint, guided by an intuition from the literature on generalized linear models (references in L71-75 and C.6.). Indeed, for these very related models, a characterisation of the scenarios where it is possible to remove this property is still an open question to the best of our knowledge. As for the MAB case, we conjecture that our framework extends beyond Missing Completely At Random or time independent censorship processes.

---

### Official Review · Reviewer_hhGV · 2022-07-19

**Rating:** 4
**Confidence:** 3
**Soundness:** 2 fair
**Presentation:** 3 good
**Contribution:** 2 fair

**Summary:**

This work considers bandit with censorship where the reward of the pulled arm may not be observed. Authors analyzed the regret upper bounds of existing algorithms under the new proposed setting.

**Questions:**

See my comments above.

**Limitations:**

No obvious limitation.

**Strengths And Weaknesses:**

Strengths:
The setting is novel, and the analysis is solid.

Weaknesses:
1. As a new setting, the author does not provide motivating examples. While some previous works may relate to the setting, the author should answer why do we need a general setting.
2. No new algorithm is proposed. If presenting new algorithm is challenging, I think authors should discuss why it is challenging and the drawbacks of existing algorithm when applied to the new setting.

---

> ### Author Response · Authors · 2022-08-02
> **Response to reviewer hhGV question 1**
>
> We thank the reviewer for their comments and feedback. Please see our answer to question 1 below:
>
> > 1.1) _“As a new setting, the author does not provide motivating examples. While some previous works may relate to the setting, the author should answer why we need a general setting. ”_
>
> **Motivation:** As stated in the introduction, our work is motivated by settings in which the feedback received by the decision-maker in each round of decision is censored by a stochastic process that depends on the current action as well as past history of feedbacks and actions. Examples are many – in the context of an e-commerce platform,  an operator (principal) aims to maximize a cumulative reward metric (e.g. benefits or user satisfaction) by recommending products to customers (agents). At a given time (stage), the principal can only revise the quality estimates on specific products based on the data from agents who follow its recommendation to buy those products. Thus, the choice model of the agents endogenizes the censorship process. Additionally, unreliable communication between principal and agents in many platform markets can be modeled as stochastic censorship. Our paper is motivated by censored feedback settings in which the data generating process is mediated by agents’ behavior and/or the data available is incomplete due to reliability issues. We provide several motivating references in the introduction of the paper (L25-33), and will be happy to elaborate more in our final submission.
>
> **Another motivation:** It may be of interest to note that somewhat similar questions were raised in the recent value alignment literature to study the extent to which an AI agent can infer agents’ preferences and beliefs from her choices or actions i.e. the question of learning from human feedback. Here a variety of settings, ranging from pure learning [1,2] to sequential and/or cooperative games under partial information [3,4] are of interest. We believe that the ideas presented in our paper are useful to study these settings.
>
> **Need for general setting:** The “optimality” in all such dynamic decision-making problems depend on how fast the unknown latent parameters – which affect the agent preferences and/or product quality and hence the stage-wise rewards – can be learned. The question then is to develop efficient algorithms that account for the censorship and estimate the performance loss, relative to the “no censorship” benchmark.
> Our work establishes first theoretical results for evaluating the statistical value of received feedback under a broad class of censorship models. To our knowledge, our work is the first normative inquiry of how censorship impacts the statistical complexity of bandit problems. We considered a generic yet well motivated censorship model and chose not to fine tune known algorithms for some specific censored environments as done in the previous literature, but rather focused on providing useful insights on the performance of UCB class of algorithms that are commonly used  in many practical settings. In particular, we provided a complete characterisation for the finite arm case and to the best of our knowledge, developed a  first set of results for the linear case.
>
>
> **Biblio:**
>
> [1] “Inverse Reward Design”, Dylan Hadfield-Menell and Smitha Milli and Pieter Abbeel and Stuart Russell and Anca Dragan, 2017
>
> [2] “Deep reinforcement learning from human preferences”, Paul Christiano and Jan Leike and Tom B. Brown and Miljan Martic and Shane Legg and Dario Amodei, 2017
>
> [3] Cooperative Inverse Reinforcement Learning, Dylan Hadfield-Menell and Anca Dragan and Pieter Abbeel and Stuart Russell, 2016
>
> [4] “The Assistive Multi-Armed Bandit”, Chan, Lawrence and Hadfield-Menell, Dylan and Srinivasa, Siddhartha and Dragan, Anca, 2019

---

> ### Author Response · Authors · 2022-08-02
> **Response to reviewer hhGV question 2**
>
> We thank the reviewer for their comments and feedback. Please see our answer to question 2 below:
>
> > 1.2) _“No new algorithm is proposed. If presenting new algorithm is challenging, I think authors should discuss why it is challenging and the drawbacks of existing algorithm when applied to the new setting.”_
>
> **Performance of UCB class under censorship:** We focus on understanding how UCB – used out-of-the-box and without any adaptations – behaves under censorship, as mentioned in L64-70. Our humble viewpoint is that the significant use of this class of algorithms by practitioners in various settings and absence of systematic tools to study censored feedback provides a sufficiently strong basis to study this question. Our paper initiates this line of work.
>
> Fortunately, our results (Thm. 3.1 and Prop. 3.2 for the MAB case and Thm. 4.1 for LCB) suggest that the UCB class of algorithms is indeed a robust and reliable method for stochastic MAB and LCB problems under censorship.
>
> **Limitations of UCB:** That being said, the limitation lies in the initial transient phase during which initial misspecification of censorship is self-corrected at an extra cost (Sec. 4.4, L313-319) - we are the first ones to identify this phenomenon. Such correction can be interpreted as a consequence of misspecification of the prior (the principal believing that it operates in an uncensored environment) and can likely be improved by either using a more suited censored-aware prior or learning the characteristics of censorship on the fly.
>
> **New algorithms:** To your point on the design of new algorithms, indeed most (if not all) previous papers require either modifying familiar MAB algorithms to account for specific censorship models, or propose new, delay-robust algorithms. From a practitioner viewpoint though, it is not clear that these refinements can be implemented easily. Thus, we need a generic and well-grounded design approach to tackle censorship and avoid the pitfalls of  an algorithmic zoo of fine-tuned solutions. Moreover, the analysis approach we develop is quite useful for designing newer censorship-robust algorithms, either in the style of UCB or leveraging related design principles such as Thompson Sampling or Information Directed Sampling. Our proof techniques and in particular the reduction and optimization of a potential-based quantity can be crucial for providing guarantees for these new algorithms.

---

### Meta-Review · Area_Chair_tAKU · 2022-08-27

**Recommendation:** Accept
**Confidence:** Less certain

**Metareview:**

The reviews for this paper have a high variance (ratings 3,4,6,7). The reviewer who gave a 3 seems unfamiliar with the basic bandit literature and I don't find the review particularly insightful. The other three reviewers mention that the theory is useful and solid, and the setting is interesting to them. On the negative side the reviewers mention that the paper is quite dense and some motivating examples would be useful. These are both minor points and, in my opinion, the positive aspects outweigh the negatives.

**Award:**

No

---

### Decision · Program_Chairs · 2022-09-14

Accept